# Purifying Approximate Differential Privacy with Randomized Post-processing

**Yingyu Lin,**\* **Erchi Wang,**\* **Yi-An Ma, Yu-Xiang Wang**
University of California, San Diego
{yil208, erw011, yianma, yuxiangw}@ucsd.edu

## Abstract

We propose a framework to convert $(\varepsilon, \delta)$-approximate Differential Privacy (DP) mechanisms into $(\varepsilon', 0)$-pure DP mechanisms under certain conditions, a process we call "purification." This algorithmic technique leverages randomized post-processing with calibrated noise to eliminate the $\delta$ parameter while achieving near-optimal privacy-utility tradeoff for pure DP. It enables a new design strategy for pure DP algorithms: first run an approximate DP algorithm with certain conditions, and then purify. This approach allows one to leverage techniques such as strong composition and propose-test-release that require $\delta > 0$ in designing pure-DP methods with $\delta = 0$. We apply this framework in various settings, including Differentially Private Empirical Risk Minimization (DP-ERM), stability-based release, and query release tasks. To the best of our knowledge, this is the first work with a statistically and computationally efficient reduction from approximate DP to pure DP. Finally, we illustrate the use of this reduction for proving lower bounds under approximate DP constraints with explicit dependence in $\delta$, avoiding the sophisticated fingerprinting code construction.

## 1 Introduction

Differential privacy (DP), in its original form [DMNS06, Definition 1], has only one privacy parameter $\varepsilon$. Over the two decades of research in DP, many have advocated that DP is too stringent to be practical and have proposed several relaxations. Among them, the most popular is arguably the *approximate DP* [DKM+06], which introduces a second parameter $\delta$.

**Definition 1 (Differential privacy [DMNS06, DR+14])** *A mechanism $\mathcal{M}$ satisfies $(\varepsilon, \delta)$-differential privacy if, for all neighboring datasets $D \simeq D'$ (datasets differing in at most one entry) and for any measurable set $S \subseteq Range(\mathcal{M})$, it holds that:*

$$\mathbb{P}[\mathcal{M}(D) \in S] \le e^\varepsilon \mathbb{P}[\mathcal{M}(D') \in S] + \delta.$$

When $\delta = 0$, the definition is now fondly referred to as $\varepsilon$-*pure* DP. Choosing $\delta > 0$ significantly weakens the protection, as it could leave any event with probability smaller than $\delta$ completely unprotected.

Two common reasons why DP researchers adopt this relaxation are: **1. Utility gain:** approximate DP is perceived to be more practical, as it allows for larger utility; **2. Flexible algorithm design:** Many algorithmic tools (such as advanced composition and Propose-Test-Release) support approximate DP but not pure DP, which enables more flexible (and often more efficient) algorithm design when $\delta > 0$ is permitted. For these two reasons, it is widely believed that the relaxation is a *necessary evil*.

We argue that the claim of "worse utility for pure-DP" is oftentimes a *myth*. In some applications of DP, a pure-DP mechanism has both stronger utility and stronger privacy. For example, in low-dimensional private histogram release, the Laplace mechanism enjoys smaller variance in almost all

---

\* Equal Contribution. Alphabetical order.

regimes except when $\delta$ is too large to be meaningful (see Figure 1). In other cases, the poor utility is sometimes not caused by an information-theoretic barrier, but rather due to the second issue — it is often much harder to design optimal pure-DP mechanisms.

In the problem of privately releasing $k$ linear queries of the dataset in $\{0,1\}^d$, it may appear that pure-DP mechanisms are much worse if we only consider composition-based methods. The composition of the Gaussian mechanism enjoys an expected worst-case error of $\mathcal{O}_p(\frac{\sqrt{k \log(1/\delta)}}{n\varepsilon})$. On the other hand, if we wish to achieve pure DP, the composition of the Laplace mechanism's error bound becomes $\mathcal{O}_p(\frac{k}{n\varepsilon})$. However, the bound can be improved to $\mathcal{O}_p(\frac{\sqrt{kd}}{n\varepsilon})$ when we use a more advanced pure-DP algorithm known as the K-norm mechanism [HT10] that avoids composition.

In the problem of private empirical risk minimization (DP-ERM) [BST14], the optimal excess empirical risk of $\Theta\left(\frac{\sqrt{d \log(1/\delta)}}{n\varepsilon}\right)$ under approximate DP is achieved by the noisy-stochastic gradient descent (NoisySGD) mechanism and its full-batch gradient counterpart. These methods are natural and are by far the strongest approaches for differentially private deep learning as well [ACG+16, DBH+22]. The optimal rate of $\Theta(\frac{d}{n\varepsilon})$ for

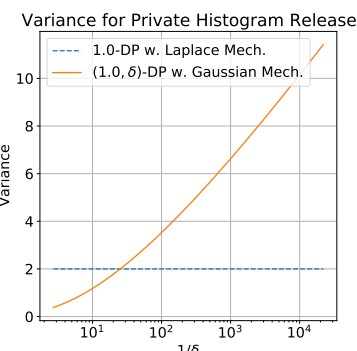

Figure 1: Per-coordinate variance of private histogram release under pure-DP and approximate DP with the same $\varepsilon$ parameter. The dashed line is given by Laplace mechanism with pure 1.0-DP, regardless of $\delta$. The solid line is generated using the analytic Gaussian mechanism in [BW18].

pure DP, on the contrary, cannot be achieved by a Laplace noise version of NoisySGD, as advanced composition is not available for pure DP. Instead, it requires an exponential mechanism that demands a delicate method to deal with the mixing rate and sampling error of certain Markov chain Monte Carlo (MCMC) sampler [LMW+24].

The issue with the lack of algorithmic tools for pure DP becomes more severe in data-adaptive DP mechanisms. For example, all Propose-Test-Release (PTR) style methods [DL09] involve privately testing certain properties of the input dataset, which inevitably incurs a small failure probability that requires choosing $\delta > 0$. While smooth-sensitivity-based methods [NRS07] can achieve pure DP, they require adding heavy-tailed noise, which causes the utility to deteriorate exponentially as the dimension $d$ increases.

## 1.1 Summary of the Results

In this paper, we develop a new algorithmic technique called "purification" that takes any $(\varepsilon, \delta)$-approximate DP mechanism and converts it into an $(\varepsilon + \varepsilon')$-pure DP mechanism, while still enjoying similar utility guarantees of the original algorithm.

The contributions of our new technique for achieving pure DP are as follows.

1. It simplifies the design of the near-optimal pure DP mechanism by allowing the use of tools reserved for approximate DP.

2. It enables an $O(\sqrt{k})$-type composition without compromising the DP guarantee with $\delta > 0$, where $k$ is the number of composition.

3. It shows that PTR-like mechanisms with pure DP are possible! To the best of our knowledge, our method is the only pure DP method for such purposes.

4. In DP-ERM and private selection problems, we show that, up to a logarithmic factor, the resulting pure DP mechanism enjoys an error rate that matches the optimal rate for pure DP, i.e., replacing the $\log(1/\delta)$ term with the dimension $d$ or $\log(|\text{OutputSpace}|)$.

**Technical summary.** The main idea of the "purification" technique is to use a randomized post-processing approach to "smooth" out the $\delta$ part of $(\varepsilon, \delta)$-DP. To accomplish this, we proceed as follows. We leverage an equivalent definition of $(\varepsilon, \delta)$-DP that interprets $\delta$ as the total variation distance from a pair of hypothetical distributions that are $\varepsilon$-indistinguishable. Next, we develop a method to convert the total variation distance to the $\infty$-Wasserstein distance. Finally, we leverage

Table 1: Summary of applications of our purification technique for constructing new pure-DP mechanisms from existing approximate DP mechanisms. The resulting pure-DP mechanisms are either *information-theoretically optimal or match the results of the best-known pure-DP mechanisms for the task* (we include more discussion in Appendix G). DP-ERM (SC) refers to DP-ERM with a strongly convex objective function, where $\lambda$ denotes the strong convexity parameter of the individual loss function. DP-ERM ($\ell_1$) refers to DP-ERM with an $\ell_1$ constraint. $\lambda_{\min}$ in linear regression is the smallest eigenvalue of the sample covariance matrix, $X^T X/n$. All results are presented up to a logarithmic factor (in $n$ and other parameters, but not in $1/\delta$. We note that in the table $\log(1/\delta)$ is proportional to $d$, the effective dimension.)

| Problem | $(\varepsilon,\delta)$-DP Mechanism | Utility (before purification) | Utility (after) |
|---|---|---|---|
| DP-ERM | Noisy-SGD [BST14] | $\frac{\sqrt{d\log(1/\delta)}L\|\mathcal{C}\|_2}{n\varepsilon}$ | $\frac{dL\|\mathcal{C}\|_2}{n\varepsilon}$ (Thm 3) |
| DP-ERM (SC) | Noisy-SGD [BST14] | $\frac{d\log(1/\delta)L^2}{\lambda n^2\varepsilon^2}$ | $\frac{d^2L^2}{\lambda n^2\varepsilon^2}$ (Thm 3) |
| DP-ERM ($\ell_1$) | DP-Frank-Wolfe [TTZ14] | $\frac{(\log d)^{2/3}(\log(1/\delta))^{1/3}}{(n\varepsilon)^{2/3}}$ | $\sqrt{\frac{\log d}{n\varepsilon}}$ (Thm 4) |
| Bounding $\Delta_{\text{local}}$ | PTR-type [KNRS13, DRE+20] | $\frac{d^{1/2}(\Delta_{\text{local}}+\log(1/\delta)/\varepsilon)}{\varepsilon}$ | $\frac{d^{1/2}\Delta_{\text{local}}}{\varepsilon}+\frac{d^{3/2}}{\varepsilon^2}$ (Thm 5) |
| Mode Release | Distance to Instability [TS13] | $\frac{\log(1/\delta)}{\varepsilon}$ | $\frac{\log|\mathcal{X}|}{\varepsilon}$ (Thm 6) |
| Linear Regression | AdaSSP [Wan18] | $\min\{\frac{\sqrt{d\log(1/\delta)}}{n\varepsilon},\frac{d\log(1/\delta)}{\lambda_{\min}n^2\varepsilon^2}\}$ | $\min\{\frac{d}{n\varepsilon},\frac{d^2}{\lambda_{\min}n^2\varepsilon^2}\}$ (Thm 7) |
| Query Release | MWEM [HLM12] | $\frac{(\log k)^{1/2}(d\log(1/\delta))^{1/4}}{\sqrt{n\varepsilon}}$ | $\frac{(\log k)^{1/3}d^{1/3}}{(n\varepsilon)^{1/3}}$ (Thm 8) |

the Approximate Sample Perturbation (ASAP) technique from [LMW+24] that achieves pure DP by adding Laplace noise proportional to the $\infty$-Wasserstein distance. A challenge arises because the output distribution of a generic $(\varepsilon,\delta)$-DP mechanism is not guaranteed to be supported on the entire output space, which may invalidate a tight TV distance to $W_\infty$ conversion. We address this by interlacing a very small uniform distribution over a constraint set. Another challenge is that, unlike in [LMW+24], where the $\varepsilon$-indistinguishable distributions correspond to pure DP mechanisms on neighboring datasets, here the hypothetical distribution may depend on both neighboring datasets rather than just one. This prevents the direct application of the standard DP analysis of the Laplace mechanism and the composition theorem. To address this, we formulate our analysis in terms of the indistinguishability of general distributions rather than DP-specific language. We use a Laplace perturbation lemma and the weak triangle inequality, leading to a clean and effective analysis. Moreover, we show how the dimension-reduction technique can be used so the purification technique can be applied to discrete outputs and to sparse outputs, which ensures only logarithmic dependence in the output-space cardinality or dimension.

## 1.2  Related Work

The idea of using randomized post-processing to enhance privacy guarantees has been explored in prior work [FMTT18, MV22, LMW+24]. [FMTT18] focus on Rényi differential privacy, while [MV22, LMW+24] study the implementation of the exponential mechanism, specifically perturbing Markov chain Monte Carlo (MCMC) samples to obtain pure DP guarantees. Our work builds on [LMW+24], where we generalize the domain assumption of [LMW+24, Lemma 8] and extend the approach to more general upstream approximate DP mechanisms, such as DP-SGD [ACG+16, DBH+22]. We also discuss the connection to the classical statistics literature on randomized post-processing given by [B+51, Theorem 10] in Appendix B.

Previous works have investigated the purification of approximate differential privacy into pure DP, but limited to settings with a finite output space. A straightforward uniform mixing method can purify approximate DP mechanisms with finite output spaces, as summarized in [HC22], and we further discuss it in Appendix C. [BGH+23] first transform an approximate DP mechanism into a replicable one, and then apply a pure DP selection procedure to obtain a pure DP mechanism[2], at the cost of increased sample complexity.

## 2  Algorithm: Converting Approximate DP to Pure DP

In this section, we propose the purification algorithm (Algorithm 1) which converts $(\varepsilon,\delta)$-approximate DP mechanisms with continuous output spaces into $\varepsilon'$-pure DP mechanisms under certain conditions. The algorithm consists of two steps: (1) mixing the approximate DP output with a uniform distribution

---

[2] Presented in [BGH+23, Algorithm 1], the algorithm can achieve a pure DP guarantee by replacing the approximate DP selection step with a pure DP alternative.

(Line 3), and (2) adding Laplace noise calibrated to $\delta^{\frac{1}{d}}$ (Line 4.) Intuitively, the first step is to enforce a bound on the $\infty$-Wasserstein distance (Definition 5), which can be loosely interpreted as a randomized analogue of $\ell_1$ sensitivity. The second step adds Laplace noise proportional to this Wasserstein bound to guarantee pure DP. This step is based on techniques from [SWC17, LMW$^+$24], and can be viewed as the generalization of the Laplace mechanism. The privacy and utility guarantee of Algorithm 1 are provided in Theorem 1.

---

**Algorithm 1:** $\mathcal{A}_{\text{pure}}(x_{\text{apx}}, \Theta, \varepsilon', \delta, \omega)$: Purification of Approximate Differential Privacy

**1 Input:** Privacy parameters $\varepsilon, \delta$, additional privacy budget $\varepsilon'$, an output $x_{\text{apx}}$ of an $(\varepsilon, \delta)$-DP algorithm $\mathcal{M}$ satisfying Assumption 1, an $\ell_q$ ball $\Theta$ as Assumption 1, the mixture level $\omega$

**2** Set $\Delta \leftarrow 2d^{1-\frac{1}{q}} R \left(\frac{\delta}{2\omega}\right)^{\frac{1}{d}}$               ▷ Lemma 6.

**3** With probability $1 - \omega$, set $x \leftarrow x_{\text{apx}}$; otherwise, sample $x \sim \text{Unif}(\Theta)$.    ▷ Uniform Mixing

**4** $x_{\text{pure}} \leftarrow x + \text{Laplace}^{\otimes d} \left(2\Delta/\varepsilon'\right)$

**5 Output:** $x_{\text{pure}}$

---

**Notations.** Let $\mathcal{X}$ be the space of data points, $\mathcal{X}^* := \cup_{n=0}^{\infty} \mathcal{X}^n$ be the space of the data set. For a vector $v = (v_1, \ldots, v_d) \in \mathbb{R}^d$ and $q \geq 1$, we define its $\ell_q$-norm as $\|v\|_q := \left(\sum_{i=1}^d |v_i|^q\right)^{1/q}$. For a set $S \subseteq \mathbb{R}^d$, we denote its $\ell_q$-norm diameter $\text{Diam}_q(S) := \sup_{x,y \in S} \|x - y\|_q$. For a (randomized) function $\mathcal{M} : \mathcal{X} \to \mathbb{R}^d$, we denote its range as $\text{Range}(\mathcal{M}) = \{\mathcal{M}(D) : D \in \mathcal{X}^*\}$.

**Assumption 1.** The $(\varepsilon, \delta)$-DP algorithm $\mathcal{M}$ satisfies $\text{Diam}_q(\text{Range}(\mathcal{M})) \leq R$. Specifically, it lies in an $\ell_q$ ball of radius $R$, denoted by $\Theta$.

**Theorem 1** *Define $x_{pure}, x_{apx}$ as in Algorithm 1 under Assumption 1. The output of Algorithm 1 satisfies $(\varepsilon + \varepsilon')$-DP with utility guarantee*

$$\mathbb{E} \|x_{pure} - x_{apx}\|_1 \leq \omega R + \frac{4Rd}{\varepsilon'} \left(\frac{\delta}{2\omega}\right)^{\frac{1}{d}}.$$

The detailed proofs are deferred to Appendix E and Appendix F. For clarity, Theorem 1 presents the utility guarantee in the $\ell_1$ norm. Extensions to general $\ell_p$ norms follow directly from bounding the expected $\ell_q$ norm of the Laplace noise (see Equation (5)).

**Remark 2** *When applying Algorithm 1 to various settings as shown in Table 1, the utility bounds either match the known information-theoretic lower bounds for pure DP or the best-known pure-DP mechanisms for the task. By the parameter setting given in Line 2 of Algorithm 1, the $\log(1/\delta)$ factor in the utility bounds can be replaced by $d$, omitting the logarithmic factors. In Section 4.2, we further show how dimension-reduction techniques can be used when applying purification to settings with sparsity conditions.*

**Remark 3** *Parameter choices of Algorithm 1 for different settings are provided in later sections. For example, parameters for the purified DP-SGD are given in Corollary 4. Note that in Algorithm 1, we only require the range of $\mathcal{M}$ to be a subset of the $\ell_q$-ball $\Theta$, where $\Theta$ can be selected as $\ell_1$ balls ($q = 1$), $\ell_2$ balls ($q = 2$), or hypercubes ($q = \infty$), which admits simple $\mathcal{O}(d)$-runtime uniform sampling oracles.*

**Corollary 4 (Parameters of Algorithm 1 for DP-SGD)** *Let $\mathcal{M} : \mathcal{X}^* \to \Theta$ be an $(\varepsilon, \delta)$-DP mechanism, where $\Theta \subset \mathbb{R}^d$ is an $\ell_2$ ball with $\ell_2$-diameter $C$. Let $x_{apx}$ be the output of $\mathcal{M}$. Set the mixture level parameter as $\omega = \frac{1}{n^2}$, and set $\delta = \frac{2\omega}{(16Cdn^2)^d}$. Then, $\mathcal{A}_{pure}(x_{apx}, \Theta, \varepsilon' = \varepsilon, \delta, \omega)$ satisfies $2\varepsilon$-DP guarantee with utility bound $\mathbb{E}[\|x_{pure} - x_{apx}\|_2] \leq \frac{1}{n^2\varepsilon} + \frac{C}{n^2}$.*

The purification algorithm can be applied to finite output spaces. The key idea is to embed the elements in the finite output space into a hypercube using the binary representation. Given a finite output space $\mathcal{Y} = \{1, 2, 3, \ldots, 2^d\} \doteq [|2^d|]$, we first map each element to its binary representation in $\{0, 1\}^d$. Then, we apply Algorithm 1 on the cube $[0, 1]^d$ in the Euclidean Space $\mathbb{R}^d$. The procedure is outlined in Algorithm 2 with DP and utility guarantees provided in Theorem 2, and the proof is deferred to Appendix F.

---

**Algorithm 2:** $\mathcal{A}_{\text{pure-discrete}}(\varepsilon, \delta, u_{\text{apx}}, \mathcal{Y})$: Binary Embedding Purification for Finite Spaces

---

1 **Input:** privacy parameters $\varepsilon, \delta$, binary representation mapping $\mathsf{BinMap} : [2^d] \to \{0,1\}^d$,
   output $u_{\text{apx}}$ from $(\varepsilon, \delta)$-DP mechanism $\widetilde{M} : \mathcal{X}^* \to \mathcal{Y} = [2^d]$,
2 $z_{\text{bin}} \leftarrow \mathsf{BinMap}(u_{\text{apx}})$                            ▷ Binary embedding
3 $z_{\text{pure}} \leftarrow \mathcal{A}_{\text{pure}}(z_{\text{bin}}, \Theta = [0,1]^d, \varepsilon' = \varepsilon, \delta, \omega = 2^{-d})$ ▷ Purify the embedding by Algorithm 1
4 $z_{\text{round}} \leftarrow \mathsf{Round}_{\{0,1\}^d}(z_{\text{pure}})$                ▷ $\mathsf{Round}_{\{0,1\}^d}(\boldsymbol{x}) = (\mathbf{1}(x_i \geq 0.5))_{i=1}^d$
5 $u_{\text{pure}} \leftarrow \mathsf{BinMap}^{-1}(z_{\text{round}})$         ▷ Decode index back to decimal integer index
6 **Output:** $u_{\text{pure}}$

---

**Theorem 2** *If* $\delta < \frac{\varepsilon^d}{(2d)^{3d}}$, *then Algorithm 2 satisfies* $(2\varepsilon, 0)$-*pure DP with utility guarantee* $\mathbb{P}[u_{apx} = u_{pure}] > 1 - 2^{-d} - \frac{d}{2}e^{-d}$.

## 3 Technical Lemma: from TV distance to $\infty$-Wasserstein distance

In this section, we present a technical lemma that proofs the uniform mixing step in Algorithm 1 (Line 3) can enforce an $\infty$-Wasserstein distance bound, as mentioned in Section 2. This $\infty$-Wasserstein bound is a key step in our analysis, enabling the subsequent addition of Laplace noise calibrated to this bound to ensure pure DP. See Figure 4 for a summary of the privacy analysis. We define the $\infty$-Wasserstein distance below. Additional discussion can be found in Appendix A.

**Definition 5** *The* $\infty$-*Wasserstein distance between distributions* $\mu$ *and* $\nu$ *is on a separable Banach space* $(\Theta, \|\cdot\|_q)$ *is defined as*

$$W_\infty^{\ell_q}(\mu, \nu) \doteq \inf_{\gamma \in \Gamma_c(\mu,\nu)} \operatorname*{ess\,sup}_{(x,y)\sim\gamma} \|x-y\|_q = \inf_{\gamma \in \Gamma_c(\mu,\nu)} \{\alpha \mid \mathbb{P}_{(x,y)\sim\gamma}[\|x-y\|_q \leq \alpha] = 1\},$$

*where* $\Gamma_c(\mu, \nu)$ *is the set of all couplings of* $\mu$ *and* $\nu$. *The expression* $\operatorname{ess\,sup}_{(x,y)\sim\gamma}$ *denotes the essential supremum with respect to the measure* $\gamma$.

By the equivalent characterization of approximate DP (Lemma 14), we can derive a TV distance bound between the output distributions of the $(\varepsilon, \delta)$-DP mechanism on neighboring datasets and a pair of distributions that are $\varepsilon$-indistinguishable. Our goal is to translate this TV distance bound into a $W_\infty$ distance bound. In general, the total variation distance bound does not imply a bound for the $W_\infty$ distance. However, when the domain is bounded, we have the following result.

**Lemma 6 (Converting** $d_{\text{TV}}$ **to** $W_\infty$**)** *Let* $q \geq 1$, *and let* $\Theta \subseteq \mathbb{R}^d$ *be a convex set with* $\ell_q$-*norm diameter* $R$ *and containing an* $\ell_q$-*ball of radius* $r$. *Let* $\mu$ *and* $\nu$ *be two probability measures on* $\Theta$. *Suppose* $\nu$ *is the sum of two measures,* $\nu = \nu_0 + \nu_1$, *where* $\nu_1$ *is absolutely continuous with respect to the Lebesgue measure and has density lower-bounded by a constant* $p_{\min}$ *over* $\Theta$, *while* $\nu_0$ *is an arbitrary measure. Define the* $W_\infty$ *distance with respect to* $\ell_q$ (*Definition 5.*) *The following holds.*

$$\text{If } d_{\text{TV}}(\mu, \nu) < p_{\min} \cdot \text{Vol}(\mathbb{B}_{\ell_q}^d(1)) \cdot \left(\frac{r}{4R}\right)^d \cdot \Delta^d, \text{ then } W_\infty^{\ell_q}(\mu, \nu) \leq \Delta, \tag{1}$$

*where* $\text{Vol}(\mathbb{B}_{\ell_q}^d(1)) = \frac{2^d}{\Gamma(1+\frac{d}{q})} \prod_{i=1}^d \frac{\Gamma(1+\frac{1}{q})}{\Gamma(1+\frac{i}{q})}$ *is the Lebesgue measure of the* $\ell_q$-*norm unit ball, with* $\Gamma$ *being the Gamma function. E.g.,* $\text{Vol}(\mathbb{B}_{\ell_2}^d(1)) = \frac{\pi^{d/2}}{\Gamma(\frac{d}{2}+1)}$, *and* $\text{Vol}(\mathbb{B}_{\ell_1}^d(1)) = \frac{2^d}{\Gamma(d+1)}$. *In particular, if the domain* $\Theta$ *is an* $\ell_q$-*ball, then the term* $\left(\frac{r}{4R}\right)$ *Eq. (1) can be improved to* $\frac{1}{4}$.

This result builds on [LMW+24] while generalizing its domain assumption from the $\ell_2$-balls to more general convex sets. In the proof provided in Appendix F, instead of relying on [LMW+24, Lemma 24], which applies only to $\ell_2$-balls, we construct a convex hull that extends to more general convex sets. We provide the proof sketch as follows.

**Proof sketch of Lemma 6** To prove Lemma 6, we use an equivalent, non-coupling-based definition of the infinity-Wasserstein distance:

**Lemma 7 ([GS84], Proposition 5)** *Define* $\mu, \nu$ *and* $W_\infty$ *as Definition 5. Then,*

$$W_\infty^{\ell_q}(\mu, \nu) = \inf\{\alpha > 0 : \mu(U) \leq \nu(U^\alpha), \text{ for all open subsets } U \subset \Theta\},$$

*where the* $\alpha$-*expansion of* $U$ *is denoted by* $U^\alpha := \{x \in \Theta : \|x - U\|_q \leq \alpha\}$.

This definition provides a geometric interpretation of $W_\infty^{\ell_q}(\mu, \nu)$ by comparing the measure of a set $U$ to the measure of its $\alpha$-expansion $U^\alpha$.

We summarize the key idea of the proof. Suppose $TV(\mu, \nu) \leq \xi$. By the definition of total variation distance, this implies $\mu(U) \leq \nu(U) + \xi$ for any measurable set $U$. To prove that $W_\infty^{\ell_q}(\mu, \nu) \leq \Delta$ (for the $\Delta$ given in the lemma), it suffices, by Lemma 7, to show that $\mu(U) \leq \nu(U^\Delta)$ for any open set $U$.

Given that $\mu(U) \leq \nu(U) + \xi$, our goal thus reduces to proving $\nu(U) + \xi \leq \nu(U^\Delta)$. Rewriting this inequality, it suffices to prove:

$$\nu(U^\Delta \setminus U) \geq \xi.$$

To establish this, we show that the "expansion band" $U^\Delta \setminus U$ must contain sufficient mass. We use the convexity of $\Theta$ to argue that this band must contain a small $\ell_q$ ball of a specific radius (related to $\Delta$). We then use the minimum density $p_{\min}$ and the Lebesgue measure of this $\ell_q$ ball to lower-bound its $\nu$-measure. The value of $\Delta$ in the lemma statement is chosen precisely so that this lower bound (and thus $\nu(U^\Delta \setminus U)$) is at least $\xi$, which completes the sketch. $\blacksquare$

The conversion lemma requires one of the distributions to satisfy a minimum density condition (the "$p_{\min}$".) This motivates the uniform mixing step in Algorithm 1, which ensures that the mixed distribution meets this requirement. Combining the TV bound implied by $(\varepsilon, \delta)$-DP, the effect of uniform mixing, and the conversion lemma from TV distance to $W_\infty$ distance, we obtain a bound on the $\infty$-Wasserstein distance after the uniform mixing step in Algorithm 1.

**Example 8 (Tightness of the Conversion)** *Let* $\tilde{\Delta} \in (0, 1)$. *Consider the probability distributions* $\mu = \tilde{\Delta}^d \delta_{\mathbf{0}}^{\mathrm{Dirac}} + (1 - \tilde{\Delta}^d)\mathrm{Unif}\left(\mathbb{B}_{\ell_q}^d(1) \setminus \mathbb{B}_{\ell_q}^d(\tilde{\Delta})\right)$, *and* $\nu = \mathrm{Unif}\left(\mathbb{B}_{\ell_q}^d(1)\right)$, *where* $\delta_{\mathbf{0}}^{\mathrm{Dirac}}$ *is the Dirac measure at* $\mathbf{0}$, *and* $\mathrm{Unif}(S)$ *denotes the uniform distribution over the set $S$. Then, by Definition 12 and Lemma 7, we have* $d_{\mathrm{TV}}(\mu, \nu) = \tilde{\Delta}^d$, *and* $W_\infty^{\ell_q}(\mu, \nu) = \tilde{\Delta}$.

This shows that the bound in Lemma 6 is tight up to a constant. In this case, we have $p_{\min} = \left(\mathrm{Vol}(\mathbb{B}_{\ell_q}^d(1))\right)^{-1}$ and $R = 2r$, so Lemma 6 gives the bound $W_\infty^{\ell_q} \leq 8\tilde{\Delta}$, which matches the exact value $W_\infty^{\ell_q} = \tilde{\Delta}$ up to a constant.

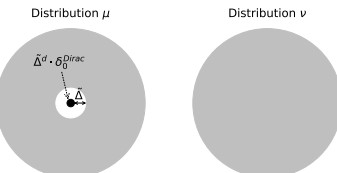

Figure 2: Illustration of Example 8 on the metric space $(\mathbb{R}^2, \|\cdot\|_2)$

# 4 Empirical Risk Minimization with Pure Differential Privacy

In this section, we apply our purification technique to develop pure differentially private algorithms for the Differential Private Empirical Risk Minimization (DP-ERM) problem, which has been extensively studied by the differential privacy community [CMS11, BST14, KJ16, WYX17, FKT20, KLL21, GHSGT23].

We consider the *convex* formulation of the DP-ERM problem, where the objective is to design a differentially private algorithm that minimizes the empirical risk $\mathcal{L}(\theta; D) = \frac{1}{n}\sum_{i=1}^{n} f(\theta; x_i)$ given a dataset $D = \{x_1, \ldots, x_n\} \subseteq \mathcal{X}^n$, a convex feasible set $\mathcal{C} \subseteq \mathbb{R}^d$, and a convex loss function $f : \mathcal{C} \times \mathcal{X} \rightarrow \mathbb{R}$. Algorithm performance is measured by the expected excess empirical risk $\mathbb{E}_{\mathcal{A}}[\mathcal{L}(\theta)] - \mathcal{L}^*$, where $\mathcal{L}^* = \min_{\theta \in \mathcal{C}} \mathcal{L}(\theta)$.

## 4.1 Purified DP Stochastic Gradient Descent

The Differential Private Stochastic Gradient Descent (DP-SGD) mechanisms [BST14, ACG$^+$16] are the most popular algorithms for DP-ERM. These mechanisms are inherently iterative and heavily rely on (1) advanced privacy accounting/composition techniques [BS16, Mir17, DRS22] and (2) amplification by subsampling for Gaussian mechanisms [BBG18, BDRS18, WBK19, ZW19, KJH20].

Either of these two techniques results in $(\varepsilon, \delta)$-DP guarantee with $\delta > 0$. In contrast, directly using the Laplace mechanism to release gradients fails to achieve a competitive utility rate.

While the exponential mechanism [MT07, BST14] achieves optimal utility rates under $(\varepsilon, 0)$-pure DP, this comes at the expense of increased computational complexity. Specifically, [BST14, Algorithm 2] implements the exponential mechanism via a random walk over the grid points of a cube that contains $\mathcal{C}$, ensuring convergence in terms of max-divergence. This approach constitutes a zero-order method that does not leverage gradient information. To design a fast, pure DP algorithm with nearly optimal utility, we propose a pure DP-SGD method that transforms the output of an $(\varepsilon, \delta)$-DP SGD algorithm into a $(\varepsilon, 0)$-pure DP solution using Algorithm 1. Implementation details are provided in Algorithm 4 in Appendix H. Theoretical guarantees on utility, privacy, and computational efficiency are stated in Theorem 3.

**Theorem 3 (Utility, privacy, and runtime for purified DP-SGD)** *Let $\mathcal{C} \subset \mathbb{R}^d$ be a convex set with $\ell_2$ diameter $C$, and suppose that $f(\cdot; x)$ is L-Lipschitz for every $x \in \mathcal{X}$. Set the parameters as specified in Corollary 4. Algorithm 4 guarantees $2\varepsilon$-pure differential privacy. Furthermore, using $\tilde{\mathcal{O}}(n^2 \varepsilon^{3/2} d^{-1})$ incremental gradient calls, the resulting output $\theta_{\mathrm{pure}}$ satisfies:*

1. *If $f(\cdot; x)$ is convex for every $x \in \mathcal{X}$, then $\mathbb{E}_{\mathcal{A}}[\mathcal{L}(\theta_{\mathrm{pure}})] - \mathcal{L}^* \leq \tilde{\mathcal{O}}(CLd/n\varepsilon)$.*

2. *If $f(\cdot; x)$ is $\lambda$-strongly convex for every $x \in \mathcal{X}$, then $\mathbb{E}_{\mathcal{A}}[\mathcal{L}(\theta_{\mathrm{pure}})] - \mathcal{L}^* \leq \tilde{\mathcal{O}}(d^2 L^2/n^2 \lambda \varepsilon^2)$.*

The total runtime of Algorithm 4 is $\tilde{\mathcal{O}}(n^2 \varepsilon^{3/2} + d)$, where each incremental gradient computation incurs a cost of $\mathcal{O}(d)$ gradient operations, and the purification (Algorithm 1) executes in $\mathcal{O}(d)$ time. Table 2 presents a comparison of utility and computational efficiency. Notably, the purified DP-SGD attains a near-optimal utility rate, matching that of the exponential mechanism [BST14], while substantially reducing computation complexity by improving the dependence on both $n$ and $d$.

Table 2: Comparison of utility and computational complexity in the $\ell_2$ Lipschitz and convex setting under $\varepsilon$-pure differential privacy. For simplicity, we assume the data domain $\mathcal{C}$ has $\ell_2$ diameter 1 and the Lipschitz constant $L = 1$. All results are stated up to logarithmic factors.

| Mechanism | Reference | Utility | Runtime |
|---|---|---|---|
| Laplace Noisy GD | Lemma 19 | $d^{1/2}/(n\varepsilon)^{1/2}$ | $n^2 \varepsilon$ |
| Purified DP-SGD (Algo. 4) | Theorem 3 | $d/n\varepsilon$ | $n^2 \varepsilon^{3/2} + d$ |
| Exponential mechanism | [BST14, Theorem 3.4] | $d/n\varepsilon$ | $d^4 n^3 \vee d^3 n^4 \varepsilon$ |

## 4.2 Purified DP Frank-Wolfe Algorithm

As noted in Section 2, applying Algorithm 1 yields the relation $\log(1/\delta) \sim d \log(\varepsilon/\Delta)$, introducing an extra $\tilde{\mathcal{O}}(d)$ factor due to the Laplace noise scale $\Delta/\varepsilon$. While optimal in low dimensions, this factor can degrade utility for algorithms with dimension-independent convergence, such as the Frank-Wolfe algorithm and its DP variants [FW+56, TTZ14, AFKT21, BGN21]. To address this, we integrate dimension-reduction and sparse recovery techniques (Algorithm 8) into our purification method (Algorithm 1). Applied to the $(\varepsilon, \delta)$-DP Frank-Wolfe algorithm from [TGTZ15, Algorithm 2], which uses the exponential mechanism and advanced composition, our approach preserves dimension-independent convergence rates.

To obtain a pure-DP estimator from the approximate DP output $\theta_{\mathrm{FW}}$, we first apply dimension reduction, exploiting the problem's sparsity to project $\theta_{\mathrm{FW}}$ into a lower-dimensional space. We then apply the purification Algorithm 1 in this space and recover the estimate in the original ambient space. The full procedure is given in Algorithm 7, with the proof of Theorem 4 deferred to Appendix I.3. Our method achieves pure differential privacy while matching the best known utility rate $\tilde{\mathcal{O}}(n^{-1/2} \varepsilon^{-1/2})$, as in [AFKT21, Theorem 6].

**Theorem 4** *Let the domain $\mathcal{C}$ be an $\ell_1$-ball centered at $\mathbf{0}$. Let $\varepsilon$ be the pure differential privacy parameter. Assume that the function $f(\cdot; \mathrm{x})$ is convex, $L_1$-Lipschitz, and $\beta$-smooth with respect to the $\ell_1$ norm for all $\mathrm{x} \in \mathcal{X}$. Algorithm 7 satisfies $2\varepsilon$-pure differential privacy and achieves the following utility bound:*

$$\mathbb{E}_{\mathcal{A}}\left[\mathcal{L}(\theta_{\mathrm{pure}})\right] - \mathcal{L}^* \leq \tilde{\mathcal{O}}\left(\frac{L_1^{1/2}\beta^{1/2}\|\mathcal{C}\|_1^{3/2}}{(n\varepsilon)^{1/2}}\right).$$

*The runtime is $\mathcal{O}(dn^{3/2})$, plus a single call to a LASSO solver.*

The upper bound is tight with respect to $n$ and $\varepsilon$. We provide a lower bound in Lemma 30, which is derived using a packing argument. We defer proof details to Appendix I.4.

## 5 Pure DP Data-dependent Mechanisms

This section shows that our purification technique offers a systematic approach for designing data-dependent pure DP mechanisms. We present three examples: (1) the Propose-Test-Release (PTR) mechanism and its variant with privately released local sensitivity [DL09, KNRS13, DRE+20]; (2) stable value release methods, such as frequent item identification [TS13, Vad17]; and (3) a linear regression algorithm using adaptive perturbation of sufficient statistics [Wan18].

### 5.1 Propose-Test-Release with Pure Differential Privacy

Given a dataset $D \in \mathcal{X}^*$ and a query function $q : \mathcal{X} \to \mathbb{R}$, the Propose-Test-Release (PTR) framework proceeds in three steps: (1) Propose an upper bound $b$ on $\Delta_{\mathrm{Local}}^q(D)$, the local sensitivity of $q(D)$ (as defined in Definition 33); (2) Privately test whether $D$ is sufficiently distant from any dataset that violates this bound; (3) If the test succeeds, assume the sensitivity is bounded by $b$, and use a differentially private mechanism, such as the Laplace mechanism with scale parameter $b/\varepsilon$, to release a slightly perturbed query response. However, due to the failure probability of the testing step, PTR provides only approximate differential privacy. By applying the purification technique outlined in Algorithm 10 in Appendix J.1, PTR can be transformed into a pure DP mechanism.

Next, we examine a variant of the PTR framework, presented in Algorithm 11, where the output space of the query function has dimension $d$, and the local sensitivity $\Delta_{\mathrm{Local}}^q(D)$ is assumed to have bounded global sensitivity. In this approach, Algorithm 11 first privately constructs a high-probability upper bound on the local sensitivity. The query is then released with additive noise proportional to this bound, and the purification technique is applied to ensure a pure DP release. This method enables an adaptive utility upper bound that depends on the local sensitivity, as established in Theorem 5. The proof is provided in Appendix J.2.

**Theorem 5** *Algorithm 11 satisfies $3\varepsilon$-DP. Moreover, the output $q_{\mathrm{pure}}$ from Algorithm 11 satisfies*

$$\mathbb{E}[\|q_{\mathrm{pure}} - q(D)\|_2] \leq \tilde{\mathcal{O}}\left(\frac{d^{1/2}\Delta_{\mathrm{Local}}^q(D)}{\varepsilon} + \frac{d^{3/2}}{\varepsilon^2}\right).$$

### 5.2 Pure DP Mode Release

We address the problem of privately releasing the most frequent item, commonly referred to as mode release, argmax release, or voting [DR+14]. Our approach follows the distance-to-instability algorithm, $\mathcal{A}_{\mathrm{dist}}$, introduced in [TS13] and further outlined in Section 3.3 of [Vad17]. Let $\mathcal{X}$ denote a finite data universe, and let $D \in \mathcal{X}^n$ be a dataset. We define the mode function $f : \mathcal{X}^n \to \mathcal{X}$, where $f(D)$ returns the most frequently occurring element in the dataset $D$.

**Theorem 6** *Let $D \in \mathcal{X}^n$ be a dataset. Algorithm 12 satisfies $2\varepsilon$-DP and runs in time $\mathcal{O}(n + \log|\mathcal{X}|)$. Furthermore, if the gap between the frequencies of the two most frequent items exceeds $\Omega\left(\log|\mathcal{X}|\log(\log|\mathcal{X}|/\varepsilon)/\varepsilon\right)$, Algorithm 12 returns the mode of $D$ with probability at least $1 - \mathcal{O}(1/|\mathcal{X}|)$.*

The proof is provided in Appendix J.3. Consider the standard Laplace histogram approach, which releases the element with the highest noisy count. To ensure that the correct mode is returned with high probability, the gap between the largest and second-largest counts must exceed $\Theta\left(\log|\mathcal{X}|/\varepsilon\right)$ [Vad17, Proposition 3.4]. Our result matches this bound up to a $\log\log|\mathcal{X}|$ factor.

### 5.3 Pure DP Linear Regression

We conclude this section by presenting a pure DP algorithm for the linear regression problem. Given a fixed design matrix $X \in \mathcal{X} \subset \mathbb{R}^{n \times d}$ and a response variable $Y \in \mathcal{Y} \subset \mathbb{R}^n$, we assume the existence of $\theta^* \in \Theta$ such that $Y = X\theta^*$. The non-private ordinary least squares estimator $(X^\top X)^{-1}X^\top Y$, requires computing the two sufficient statistics: $X^\top X$ and $X^\top Y$. These can be privatized using Sufficient Statistics Perturbation (SSP) [VS09, FGWC16] or its adaptive variant, AdaSSP [Wan18], to achieve improved utility.

To facilitate the purification procedure, we first localize the output of AdaSSP by deriving a high-probability upper bound. This bound is then used to clip the output of AdaSSP, after which the purification technique is applied. Implementation details are provided in Algorithm 13, and the corresponding utility guarantee is stated in Theorem 7. The proof is deferred to Appendix J.4.

**Theorem 7** *Assume $X^\top X$ is positive definite and $\|\mathcal{Y}\|_2 \lesssim \|\mathcal{X}\|_2 \|\theta^*\|_2$. Then, with high probability, the output $\theta_{\text{pure}}$ of Algorithm 13 satisfies:*

$$\text{MSE}(\theta_{\text{pure}}) \leq \tilde{\mathcal{O}} \left( \frac{d\|\mathcal{X}\|_2^2 \|\theta^*\|_2^2}{n\varepsilon} \wedge \frac{d^2 \|\mathcal{X}\|_2^4 \|\theta^*\|_2^2}{\varepsilon^2 n^2 \lambda_{\min}} \right), \tag{2}$$

*Here, $\lambda_{\min}$ denotes the normalized minimum eigenvalue, defined as $\lambda_{\min}(X^\top X/n)$, and MSE denotes the mean squared error, given by $\text{MSE}(\theta) = \frac{1}{2n}\|Y - X\theta\|_2^2$.*

We note that [AD20] also proposes an $\varepsilon$-DP mechanism for linear regression using the approximate inverse sensitivity mechanism, which has an excess mean squared error of $\mathcal{O}\left(dL^2 \text{tr}(X^\top X/n)/n^2\varepsilon^2\right)$, where $L$ is the Lipschitz constant of the individual mean squared error loss function. We make two comparisons: (1) compared to the utility bound in [AD20, Proposition 4.3], Theorem 7 avoids dependence on the Lipschitz constant, which in turn relies on the potentially unbounded quantity $\|\Theta\|_2$; (2) both algorithms exhibit a computational complexity of $\mathcal{O}(nd^2 + d^3)$. Nonetheless, the inverse sensitivity mechanism entails further computational overhead due to the rejection sampling procedure ([AD20, Algorithm 3]).

## 6 Pure DP Query Release

We study the private query release problem, where the data universe is defined as $\mathcal{X} = \{0,1\}^d$, and the dataset is represented as a histogram $D \in \mathbb{N}^{|\mathcal{X}|}$, satisfying $\|D\|_1 = n$. We consider linear query functions $q : \mathcal{X} \to [0,1]$, where $q \in Q$, and the workload $Q$ consists of $K$ distinct queries. For convenience, we define $Q(D) := \left(\frac{1}{n}\sum_{i=1}^n q_1(d_i), \ldots, \frac{1}{n}\sum_{i=1}^n q_K(d_i)\right)$. Our goal is to release a privatized version of $Q(D)$ that minimizes the $\ell_\infty$ error.

Our algorithm, described in Algorithm 16, proceeds in three steps: (1) the multiplicative weights exponential mechanism [HLM12] is used to release a synthetic dataset; (2) subsampling is applied to this synthetic dataset to further reduce its size; (3) the subsampled dataset is encoded into a binary representation and then passed through the purification procedure (Algorithm 2). The subsampling step is necessary to balance the additional error introduced by purification, which depends on the size of the output space. The privacy and utility guarantees are formalized in Theorem 8, with the proof deferred to Appendix K.2.

**Theorem 8** *Algorithm 16 satisfies $2\varepsilon$-DP, and the output $D_{\text{pure}}$ of the following utility guarantee:*

$$\mathbb{E}[\|Q(D) - Q(D_{\text{pure}})\|_\infty] \leq \tilde{\mathcal{O}} \left( \frac{d^{1/3}}{n^{1/3}\varepsilon^{1/3}} \right).$$

*Moreover, the runtime of the algorithm is $\tilde{\mathcal{O}}(nK + |\mathcal{X}| + \varepsilon^{2/3}n^{2/3}d^{1/3}|\mathcal{X}|K)$.*

We emphasize that our algorithmic framework (Algorithm 16) is flexible. The upstream approximate differential privacy query release algorithm—currently instantiated using MWEM [HLM12]—can be replaced with other $(\varepsilon, \delta)$-DP algorithms, such as the Private Multiplicative Weights method [HR10] or the Gaussian mechanism. Our approach achieves the tightest known utility upper bound for pure DP, as provided by the Small Database mechanism [BLR13], while offering improved runtime.

## 7 Purification as a Tool for Proving Lower Bounds

In this section, we demonstrate that the purification technique can be leveraged to establish utility lower bounds for $(\varepsilon, \delta)$-differentially private algorithms. Lower bounds for pure differential privacy mechanisms are often more straightforward to derive, commonly relying on packing arguments [HT10, BBKN14]. In contrast, establishing lower bounds for approximate DP mechanisms typically requires more intricate constructions, such as those based on fingerprinting codes [BUV14, DSS+15, BSU17, LLL21, SU16].

We begin by providing the intuition behind how the purification technique can be used to prove lower bounds for approximate DP. Suppose there exists an $(\varepsilon, \delta)$-DP mechanism with $\delta$ within appropriate

range such that $\log(1/\delta) \sim \tilde{\mathcal{O}}(d)$ and an error bound of $\mathcal{O}\left(d^c \log^{1-c}(1/\delta)/n\varepsilon\right)$ for some constant $c \in (0,1)$, and we assume the output space is a bounded set in $\mathbb{R}^d$. Then, Theorem 1 and Remark 2 imply that it is possible to construct a $2\varepsilon$-DP mechanism with an error bound of $\tilde{\mathcal{O}}\left(d/n\varepsilon\right)$, where $\tilde{\mathcal{O}}$ omits logarithmic factors. This provides a new approach to proving lower bounds for $(\varepsilon, \delta)$-DP algorithms via a *contrapositive argument*:

*If no $\mathcal{O}(\varepsilon)$-pure-DP mechanism exists with an error bound $\tilde{\mathcal{O}}\left(\frac{d}{n\varepsilon}\right)$, then no $(\varepsilon, \delta)$-DP mechanism can achieve an error bound of $\mathcal{O}\left(d^c \log^{1-c}(1/\delta)/n\varepsilon\right)$ for all appropriate $\delta$, for any $c \in (0,1)$.*

We apply this approach to derive the lower bound for $(\varepsilon, \delta)$-DP mean estimation task. Specifically, let the data universe be $\mathcal{D} := \{-1/\sqrt{d}, 1/\sqrt{d}\}^d$, and consider a dataset $D = \{x_1, \ldots, x_n\} \subseteq \{-1/\sqrt{d}, 1/\sqrt{d}\}^d$. The objective is to privately release the column-wise empirical mean $\bar{D} = \frac{1}{n}\sum_{i=1}^n x_i \in [-1/\sqrt{d}, 1/\sqrt{d}]^d$. For pure DP we have the following lower bound:

**Lemma 9 (Lemma 5.1 in [BST14], simplified)** *For any $\varepsilon$-pure DP mechanism $\mathcal{M}$, there exists a dataset $D \in \{-1/\sqrt{d}, 1/\sqrt{d}\}^{n \times d}$ such that with probability at least $1/2$ over the randomness of $\mathcal{M}$, we have $\|\mathcal{M}(D) - \bar{D}\|_2 = \Omega\left(\frac{d}{n\varepsilon}\right)$.*

By applying the purification technique with an appropriately chosen $\delta$ and a contrapositive argument based on Lemma 9, we obtain the following $(\varepsilon, \delta)$-differential privacy lower bound, as stated in Theorem 9. We defer proof to Appendix L.

**Theorem 9** *Let $\varepsilon \leq \mathcal{O}(1)$, and $\delta \in \left(\exp(-4d\log(d)\log^2(nd)), \ 4n^{-d}\log^{-2d}(8d)\right)$. For any $(\varepsilon, \delta)$-DP mechanism $\mathcal{M}$, there exist a dataset $D \in \{-1/\sqrt{d}, 1/\sqrt{d}\}^{n \times d}$ such that with probability at least $1/4$ over the randomness of $\mathcal{M}$, we have*

$$\|\mathcal{M}(D) - \bar{D}\|_2 = \tilde{\Omega}\left(\frac{\sqrt{d\log(1/\delta)}}{\varepsilon n}\right).$$

*Here, $\tilde{\Omega}(\cdot)$ hides all polylogarithmic factors, except those with respect to $\delta$.*

**Remark 10** *For mean estimation under $(\varepsilon, \delta)$-DP with exponentially small $\delta$, we can also establish a $\tilde{\Omega}(d)$ lower bound via a packing argument, as detailed in Appendix L.3.*

Additional examples illustrating the use of the purification trick to derive lower bounds is provided in Appendix L.2, which includes bounds for one-way marginal release and private selection.

# 8   Conclusion and Limitation

In this paper, we introduced a novel purification framework that systematically converts approximate DP mechanisms into pure DP mechanisms via randomized post-processing. Our approach bridges the flexibility of approximate DP and the stronger privacy guarantees of pure DP. Our purification technique has broad applicability across several fundamental DP problems. In particular, we propose a faster pure DP algorithm for the DP-ERM problem that achieves near-optimal utility rates for pure DP. We also show that our method can be applied to design pure-DP PTR algorithms. A limitation of our approach is that when applying this continuous purification framework to mechanisms with a finite output space, the utility bound incurs an additional $\log\log|\mathcal{Y}|$ factor compared to prior work, where $|\mathcal{Y}|$ is the output space cardinality. Nonetheless, to our knowledge, this is the first systematic purification result applicable to continuous domains. Future work includes extending our framework to more adaptive settings and refining its applicability to high-dimensional problems.

**Acknowledgments**

The research is partially supported by NSF Awards #2048091, CCF-2112665 (TILOS), DARPA AIE program, the U.S. Department of Energy, Office of Science, and CDC-RFA-FT-23-0069 from the CDC's Center for Forecasting and Outbreak Analytics. We thank helpful discussion with Abrahdeep Thakurta, who pointed us towards proving the packing lower bound pure-DP DP-ERM under the L1 constraints (Lemma 30). We also thank anonymous reviewers from TPDP'25 for the reference to the *folklore* that mixing a uniform distribution alone suffices for "purifying" approximate DP in the discrete output space regime, as well as the anonymous reviewer from Neurips 2025 for a simple purification approach for Euclidean spaces that extends this discrete-set purification folklore (Appendix C and D). Finally, we thank Xin Lyu for pointing out a simple alternative proof of Theorem 9 based on a packing argument (Remark 10).

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

# Contents

# A Preliminaries

This section provides definitions of max divergence, total variation distance, and $\infty$-Wasserstein distance, and the lemmas that will be used in the privacy analysis of Theorem 1.

**Notations.** Let $\mathcal{X}$ be the space of data points, $\mathcal{X}^* := \cup_{n=0}^{\infty} \mathcal{X}^n$ be the space of the data set. For an integer $n$, let $[n] = \{1, \ldots, n\}$. A subgradient of a convex function $f$ at $x$, denoted $\partial f(x)$, is the set of vectors $\mathbf{g}$ such that $f(y) \geq f(x) + \langle \mathbf{g}, y - x \rangle$, for all $y$ in the domain. For simplicity, we assume the functions are differentiable in this paper and consider the gradient $\nabla f$. The operators $\cdot \vee \cdot$ and $\cdot \wedge \cdot$ denote the maximum and minimum of the two inputs, respectively. We use $\| \cdot \|_q$ to denote $\ell_q$ norm. For a set $A$, $\|A\|_q := \sup_{x \in A} \|A\|_q$ represents the $\ell_q$ radius of set $A$ and $\text{Diam}_q(A) := \sum_{x,y \in A} \|x - y\|_q$ represent the $\ell_q$ diameter of set $A$. For a finite set $S$, we denote its cardinality by $|S|$. Throughout this paper, we use $\mathbb{E}_{\mathcal{A}}[\cdot]$ to denote taking expectation over the randomness of the algorithm.

## A.1 Definitions on Distributional Discrepancy

**Definition 11 ([VEH14], Theorem 6; [DTTZ14], Definition 3.6)** *The Rényi divergence of order $\infty$ (also known as Max Divergence) between two probability measures $\mu$ and $\nu$ on a measurable space $(\Theta, \mathcal{F})$ is defined as:*

$$D_\infty(\mu \| \nu) = \ln \sup_{S \in \mathcal{F}, \ \mu(S) > 0} \left[ \frac{\mu(S)}{\nu(S)} \right].$$

*In this paper, we say that $\mu$ and $\nu$ are $\varepsilon$-indistinguishable if $D_\infty(\mu \| \nu) \leq \varepsilon$, and $D_\infty(\nu \| \mu) \leq \varepsilon$.*

A mechanism $\mathcal{M}$ is $\varepsilon$-differentially private if and only if for every two neighboring datasets $D$ and $D'$, we have $D_\infty(\mathcal{M}(D) \| \mathcal{M}(D')) \leq \varepsilon$, and $D_\infty(\mathcal{M}(D') \| \mathcal{M}(D)) \leq \varepsilon$.

**Definition 12** *The total variation (TV) distance between two probability measures $\mu$ and $\nu$ on a measurable space $(\Theta, \mathcal{F})$ is defined as:*

$$d_{\text{TV}}(\mu, \nu) = \sup_{S \in \mathcal{F}} |\mu(S) - \nu(S)|.$$

The $\infty$-Wasserstein distance between distributions captures the largest discrepancy between samples with probability 1 under the optimal coupling. Unlike other Wasserstein distances that consider expected transport costs, it offers a worst-case perspective. Though the $\infty$-Wasserstein distance can be defined with a general metric, for clarity, we focus on its $\ell_q$-norm version in this paper.

**Definition 13 (Restatement of Definition 5)** *The $\infty$-Wasserstein distance between distributions $\mu$ and $\nu$ on a separable Banach space $(\Theta, \| \cdot \|_q)$ is defined as*

$$W_\infty^{\ell_q}(\mu, \nu) \doteq \inf_{\gamma \in \Gamma_c(\mu, \nu)} \operatorname{ess\,sup}_{(x,y) \sim \gamma} \|x - y\|_q = \inf_{\gamma \in \Gamma_c(\mu, \nu)} \{\alpha \mid \mathbb{P}_{(x,y) \sim \gamma} [\|x - y\|_q \leq \alpha] = 1\},$$

*where $\Gamma_c(\mu, \nu)$ is the set of all couplings of $\mu$ and $\nu$, i.e., the set of all joint probability distributions $\gamma$ with marginals $\mu$ and $\nu$ respectively. The expression $\operatorname{ess\,sup}_{(x,y) \sim \gamma}$ denotes the essential supremum with respect to measure $\gamma$. By [GS84, Proposition 1], the infimum in this definition is attainable, i.e., there exists $\gamma^* \in \Gamma_c(\mu, \nu)$ such that $W_\infty^{\ell_q}(\mu, \nu) = \operatorname{ess\,sup}_{(x,y) \sim \gamma^*} \|x - y\|_q$.*

For Laplace perturbation (Lemma 15), we require a bound on $W_\infty^{\ell_1}$, the Wasserstein distance defined by the $\ell_1$-norm, i.e., for $q = 1$. A bound for $W_\infty^{\ell_1}$ can be derived from $W_\infty^{\ell_q}$ using the inequality $\| \cdot \|_1 \leq d^{1 - \frac{1}{q}} \| \cdot \|_q$, which gives $W_\infty^{\ell_1}(\cdot, \cdot) \leq d^{1 - \frac{1}{q}} W_\infty^{\ell_q}(\cdot, \cdot)$.

## A.2 Lemmas for DP Analysis of Algorithm 1

We now introduce the three lemmas for the privacy analysis: the equivalence definition of $(\varepsilon, \delta)$-DP (Lemma 14), the Laplace perturbation (Lemma 15), and the weak triangle inequality for $\infty$-Rényi divergence (Lemma 16).

**Lemma 14 (Lemma 3.17 of [DR+14])** *A randomized mechanism $\mathcal{M}$ satisfies $(\varepsilon, \delta)$-DP if and only if for all neighboring datasets $D \simeq D'$, there exist probability measures $P, P'$ such that $d_{\text{TV}}(\mathcal{M}(D), P) \leq \frac{\delta}{e^\varepsilon + 1}$, $d_{\text{TV}}(\mathcal{M}(D'), P') \leq \frac{\delta}{e^\varepsilon + 1}$, $D_\infty(P \| P') \leq \varepsilon$, and $D_\infty(P' \| P) \leq \varepsilon$.*

**Lemma 15 (Laplace perturbation, adapted from Theorem 3.2 of [SWC17])** *Let $\mu$ and $\nu$ be probability distributions on $\mathbb{R}^d$. Let $\text{Laplace}^{\otimes d}(b)$ denote the distribution of $\mathbf{z} \in \mathbb{R}^d$, where $z_i \overset{i.i.d.}{\sim} \text{Laplace}(b)$. If $W_\infty^{\ell_1}(\mu, \nu) \leq \Delta$, then we have*

$$D_\infty\left(\mu * \text{Laplace}^{\otimes d}\left(\tfrac{\Delta}{\varepsilon}\right) \,\|\, \nu * \text{Laplace}^{\otimes d}\left(\tfrac{\Delta}{\varepsilon}\right)\right) \leq \varepsilon, \text{ and}$$
$$D_\infty\left(\nu * \text{Laplace}^{\otimes d}\left(\tfrac{\Delta}{\varepsilon}\right) \,\|\, \mu * \text{Laplace}^{\otimes d}\left(\tfrac{\Delta}{\varepsilon}\right)\right) \leq \varepsilon.$$

The proof is provided in Appendix F for completeness. The Laplace mechanism is a special case of Lemma 15 by setting $\mu$ and $\nu$ to Dirac distributions at $f(D)$ and $f(D')$, respectively. Lemma 15 can also be derived from the limit case of [FMTT18, Lemma 20] as the Rényi order approaches infinity.

**Lemma 16 (Weak Triangle Inequality, adapted from [LSS14], Lemma 4.1)** *Let $\mu, \nu, \pi$ be probability measures on a measurable space $(\Theta, \mathcal{F})$. If $D_\infty(\mu\|\pi) < \infty$ and $D_\infty(\pi\|\nu) < \infty$, then $D_\infty(\mu\|\nu) \leq D_\infty(\mu\|\pi) + D_\infty(\pi\|\nu)$.*

The proof is deferred to Appendix F. This lemma generalizes [LSS14, Lemma 4.1], which focuses on discrete distributions. Additionally, Lemma 16 corresponds to the infinite Rényi order limit of [Mir17, Proposition 11] and [MMR12, lemma 12].

## B   Supplementary Discussion on Randomized Post-Processing

In this section, we clarify the distinction between our "purification" process and the randomized post-processing in [B$^+$51, Theorem 10]. Define the following function

$$f_{\varepsilon,\delta}(\alpha) = \max\left\{0, 1 - \delta - e^\varepsilon \alpha, e^{-\varepsilon}(1 - \delta - \alpha)\right\}.$$

By [WZ10], a mechanism $\mathcal{M}$ is $(\varepsilon, \delta)$-DP if and only if $\mathcal{M}$ is $f_\varepsilon$-DP.

[B$^+$51, Theorem 10] [DRS22, Theorem 2.10] establish the existence of a (randomized) post-processing method that transforms a pair of distributions into another pair with a dominating trade-off function. Specifically, they show that if $T(P, Q) \leq T(P', Q')$, then there exists a randomized algorithm Proc such that $\text{Proc}(P) = P'$ and $\text{Proc}(Q) = Q'$, where $T$ denotes the trade-off function [DRS22, Definition 2.1]. Their proof constructs a sequence of transformations and takes the limit. In contrast, our "purification" process provides a computationally efficient post-processing method for a different problem. Given that $f_{\varepsilon,\delta} \leq f_{(\varepsilon+\varepsilon'),0}$ (see Figure 3) and that $T(\mathcal{M}(D), \mathcal{M}(D')) \geq f_{\varepsilon,\delta}$ for all neighboring datasets $D \simeq D'$, we seek a randomized post-processing procedure $\mathcal{A}_{\text{pure}}$ such that $T(\mathcal{A}_{\text{pure}} \circ \mathcal{M}(D), \mathcal{A}_{\text{pure}} \circ \mathcal{M}(D')) \geq f_{(\varepsilon+\varepsilon'),0}$ while maintaining utility guarantee.

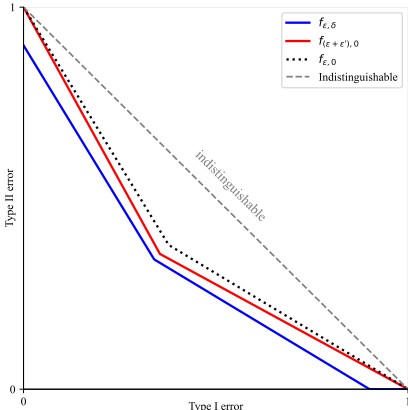

Figure 3: Trade-off functions for $(\varepsilon, \delta)$-DP, $(\varepsilon, 0)$-DP, and $(\varepsilon + \varepsilon', 0)$-DP. Our method provides a solution to post-process the $(\varepsilon, \delta)$-DP distribution pair (in blue) to the $(\varepsilon + \varepsilon', 0)$-DP pair (in red).

## C   Discussion: Purification on Finite Output Spaces

This section discusses a "folklore" method for converting approximate DP mechanisms into pure DP when the output space is *finite*, and explains why this method fails in the *continuous* case. Given an

$(\varepsilon, \delta)$-DP mechanism $\mathcal{M} : \mathcal{X}^* \to \mathcal{Y}$ with finite output space $\mathcal{Y}$, one can construct a new mechanism by mixing $\mathcal{M}$ with uniform distribution over $\mathcal{Y}$: $\mathcal{A}_{\mathrm{mix}}(\mathcal{M}) = (1-\omega)\mathcal{M} + \omega \mathrm{Unif}(\mathcal{Y})$. The resulting mechanism $\mathcal{A}_{\mathrm{mix}}(\mathcal{M})$ satisfies $(\varepsilon + \ln(1 + \frac{\delta|\mathcal{Y}|}{\omega}e^{-\varepsilon}))$-pure DP [HC22, Lemma 3.2]. However, this strategy does not yield pure DP when the output space is continuous, as the following counterexample shows.

**Example 17** *Let $f : D \to [0,1]$ be a statistic computed on a dataset $D \in \mathcal{X}^*$. Consider the following mechanism: with probability $\delta$, output the true value $f(D)$; with probability $1 - \delta$, output a value uniformly from $[0,1]$. That is, $\mathcal{M}(D) = \delta \cdot \boldsymbol{\delta}_{f(D)}^{\mathrm{Dirac}} + (1-\delta)\mathrm{Unif}([0,1])$, where $\boldsymbol{\delta}_{f(D)}^{\mathrm{Dirac}}$ denotes the Dirac measure at $f(D)$. Then $\mathcal{M}$ satisfies $(0, \delta)$-DP. Now, consider applying the uniform mixture strategy:*

$$\mathcal{A}_{\mathrm{mix}}(\mathcal{M}(D)) = (1-\omega)\mathcal{M}(D) + \omega \mathrm{Unif}([0,1])$$
$$= (1-\omega)\delta \cdot \boldsymbol{\delta}_{f(D)}^{\mathrm{Dirac}} + (1 - \delta - \delta\omega)\mathrm{Unif}([0,1]).$$

*This new mechanism satisfies $(0, (1-\omega)\delta)$-DP, but not pure DP unless $\omega = 1$, which is the trivial mixture.*

## D  Discussion: Extending the Purification on Finite Output Spaces to the Euclidean Space

A natural approach for purification on Euclidean space is to quantize the continuous output space into a finite discrete set, and apply the folklore purification for finite sets. Specifically, to purify the $(\varepsilon, \delta)$-DP mechanism $\mathcal{M} : \mathcal{X}^* \to \Theta \subseteq \mathbb{R}^d$, one could first cover the range $\Theta$ by a $\Delta$-net, round the output of $\mathcal{M}$ to the nearest grid point, and then purify it using the finite range folklore method. We thank an anonymous NeurIPS 2025 reviewer for suggesting this approach. The main challenge of this approach is the explicit construction of the $\Delta$-net and the efficient uniform sampling from it. When constructing the $\Delta$-net by the cubes (as detailed below), this natural method achieves a similar utility bound as Algorithm 1, as the bottleneck of both bounds is the term $\left(\frac{\delta}{\omega}\right)^{\frac{1}{d}}$. The bound is different in the *non-dominating* terms: if $\Theta$ is an $\ell_\infty$ ball, this quantization-based method has tighter bound; if $\Theta$ is an $\ell_1$ ball, it yields a worse bound.

**Construction of $\Delta$-net.**  We first note that a naive approach, sampling from an $\ell_1$ ball (or $\ell_2$ ball) and then rounding to the nearest grid point, does not yield a uniform distribution on the grid points. This non-uniformity arises from boundary effects. To illustrate, consider the $\ell_1$ ball of radius $R$ and take the point $x = (R, 0, \ldots, 0)$ on its surface. That the volume of the intersection:

$$\{z : \|z\|_1 \leq R, \ \|z - x\|_1 \leq r\},$$

is only a $\frac{1}{2^d}$ fraction of the ball $\{z : \|z - x\|_1 \leq r\}$.

We now provide the derivation. Writing the intersection in coordinates:

$$\sum_{i=1}^{d} |z_i| \leq R, \ |z_1 - R| + \sum_{i=2}^{d} |z_i| \leq r,$$

we get $R - r \leq z_1 \leq R$, and for such $z_1$, the other coordinates must satisfy

$$\sum_{i=2}^{d} |z_i| \leq \min\{R - z_1, r - R + z_1\}$$

Therefore, the intersection volume is $\int_{R-r \leq z_1 \leq R} \int_{\sum_{i=2}^{d} |z_i| \leq \min\{R-z_1, r-R+z_1\}} \mathbf{1} d(z_2, \ldots, z_d) dz_1$, $= \int_{R-r \leq z_1 \leq R} \frac{2^{d-1}}{(d-1)!} (\min\{R - z_1, r - R + z_1\})^{d-1} dz_1 = \frac{2^{d-1}}{(d-1)!} (\int_{R-r \leq z_1 \leq R-r/2} (r - R + z_1)^{d-1} dz_1 + \int_{R-r/2 \leq z_1 \leq R} (R - z_1)^{d-1}) dz_1 = \frac{2^{d-1}}{(d-1)!} \cdot 2\frac{1}{d}(\frac{r}{2})^d = \frac{1}{d!}r^d$, which is $\frac{1}{2^d}$ portion of the volume of the ball $\{z : \|z - x\|_1 \leq r\}$.

We also note that the above intersection volume can be lower bounded for $\ell_2$ norm, see Lemma 24 in [LMW$^+$24].

Given these challenges with non-uniformity, we instead consider constructing the $\Delta$-net by tiling a large cube $\Theta' \supseteq \Theta$ with smaller cubes.

**Cube-Based $\Delta$-net Construction and Analysis.** Without loss of generality, assume $\Theta' = [0, R]^d$. Take $\Delta = 2R\left(\frac{\delta}{2\omega}\right)^{\frac{1}{d}}\log d$, and $K = 2\lfloor\frac{R}{\Delta}\rfloor$. Construct the $\Delta$-net using cubes of the form $\prod_{k=1}^{d}\left[\frac{i_k R}{K}, \frac{(i_k+1)R}{K}\right]$. By the folklore purification, this algorithm satisfies $\left(\varepsilon + \ln(1 + \frac{\delta K^d}{\omega}e^{-\varepsilon})\right)$-pure DP. We have

$$\varepsilon + \ln(1 + \frac{\delta K^d}{\omega}e^{-\varepsilon}) \leq \varepsilon + \frac{\delta K^d}{\omega}e^{-\varepsilon} \leq \varepsilon + \frac{\delta\left(\frac{R}{R\left(\frac{\delta}{2\omega}\right)^{\frac{1}{d}}\log d}\right)^d}{\omega}e^{-\varepsilon} \leq 2\varepsilon.$$

When $q = \infty$, this method achieves the utility bound of $\mathbb{E}\left\|x_{\text{pure}} - x_{\text{apx}}\right\|_\infty \leq \omega R + \frac{4R\log d}{\varepsilon'}\left(\frac{\delta}{2\omega}\right)^{\frac{1}{d}}$.

This bound is tighter than the bound by directly applying Algorithm 1 in the dependence in $d$. For comparison, directly extending the proof of Theorem 1 (Appendix F.4) to the $\ell_\infty$ norm yields: $\mathbb{E}\left\|x_{\text{pure}} - x_{\text{apx}}\right\|_\infty \leq \omega R + \frac{4Rd(\log d+1)}{\varepsilon'}\left(\frac{\delta}{2\omega}\right)^{\frac{1}{d}}$, which is derived as follows:

$$\mathbb{E}\left\|x_{\text{pure}} - x_{\text{apx}}\right\|_\infty = \mathbb{P}(u > \omega)\mathbb{E}\left[\left\|x_{\text{pure}} - x_{\text{apx}}\right\|_\infty \mid u > \omega\right] + \mathbb{P}(u \leq \omega)\mathbb{E}\left[\left\|x_{\text{pure}} - x_{\text{apx}}\right\|_\infty \mid u \leq \omega\right]$$

$$\leq \mathbb{E}\left[\max_i |z_i|\right] + \omega R \qquad\qquad (z_i \sim \text{Laplace}\left(\frac{2\Delta}{\varepsilon'}\right))$$

$$= \left(\sum_{k=1}^{d}\frac{1}{k}\right)\frac{2\Delta}{\varepsilon'} + \omega R$$

$$\leq (\log d + 1)\frac{2\Delta}{\varepsilon'} + \omega R$$

$$\leq \frac{4Rd(\log d + 1)}{\varepsilon'}\left(\frac{\delta}{2\omega}\right)^{\frac{1}{d}} + \omega R.$$

When $q = 1$, applying the standard conversion $\|\cdot\|_1 \leq d\|\cdot\|_\infty$, this method achieves the utility bound of $\mathbb{E}\left\|x_{\text{pure}} - x_{\text{apx}}\right\|_1 \leq \omega dR + \frac{4Rd\log d}{\varepsilon'}\left(\frac{\delta}{2\omega}\right)^{\frac{1}{d}}$, which is worse than Theorem 1 by a factor of $d$.

# E   Privacy Analysis: Proof Sketch of Theorem 1

The proof sketch of Theorem 1 is illustrated in Figure 4. Let $D$ and $D'$ be neighboring datasets, and let $\mathcal{M}$ be an $(\varepsilon, \delta)$-DP mechanism. We aim to show that $\mathcal{A}_{\text{pure}}(\mathcal{M}(D))$ and $\mathcal{A}_{\text{pure}}(\mathcal{M}(D'))$—the post-processed outputs of $\mathcal{M}(D)$ and $\mathcal{M}(D')$ after applying Algorithm 1—are $\varepsilon$-indistinguishable. By sketching the proof, we also provide the intuition of the design of Algorithm 1.

First, by the equivalent definition of approximate DP (Lemma 14), there exists a hypothetical $\varepsilon$-indistinguishable distribution pair $P$ and $P'$, such that $\mathcal{M}(D)$ and $\mathcal{M}(D')$ are $\mathcal{O}(\delta)$-close to $P$ and $P'$, respectively, in total variation distance. Note that $P$ and $P'$ can both depend on $D$ and $D'$, i.e., $P = P(D, D')$ and $P' = P'(D, D')$, rather than simply $P = P(D)$ and $P' = P'(D')$. This dependence complicates the direct application of standard DP analysis. To address this, we transition to a distributional perspective.

To show that $\mathcal{A}_{\text{pure}}(\mathcal{M}(D))$ and $\mathcal{A}_{\text{pure}}(\mathcal{M}(D'))$ are $\varepsilon$-indistinguishable distributions, by the weak triangle inequality of $\infty$-Rényi divergence (Lemma 16), it suffices to show that $\mathcal{A}_{\text{pure}}(\mathcal{M}(D))$ and $\mathcal{A}_{\text{pure}}(P)$ are $\varepsilon$-indistinguishable (in terms of $\infty$-Rényi divergence), and that $\mathcal{A}_{\text{pure}}(\mathcal{M}(D'))$ and $\mathcal{A}_{\text{pure}}(P')$ are $\varepsilon$-indistinguishable as well. Now the problem reduces to: given two distributions with total variation distance bound, how to post-process them (with randomness) to obtain the $\infty$-Rényi divergence bound?

A natural way to establish the $\infty$-Renyi bound is via the Laplace mechanism, which perturbs *deterministic* variables with Laplace noise according to the $\ell_1$-sensitivity. To generalize the Laplace mechanism to perturbing *random* variables, we develop the Laplace perturbation (Lemma 15), which achieves the $\infty$-Rényi bound by convolving Laplace noise according to the $W_\infty^{\ell_1}$ distance. The $W_\infty^{\ell_1}$ distance can be viewed as a randomized analog of the $\ell_1$-distance. The remaining step is to derive a $W_\infty^{\ell_1}$ bound from the TV distance bound, and this motivates Lemma 6, a $d_{\text{TV}}$ to $W_\infty^{\ell_q}$ conversion

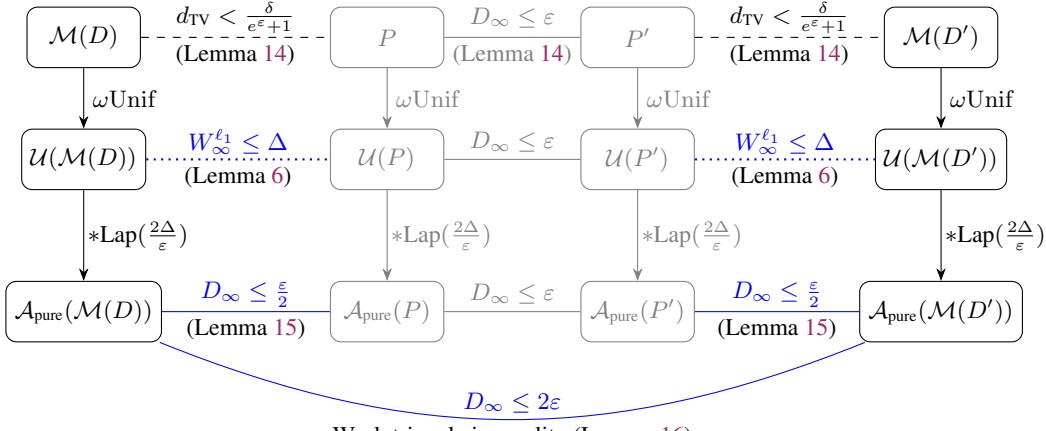

Figure 4: Flowchart illustrating the proof sketch of Theorem 1 and the intuition behind Algorithm 1. The notation $D_\infty \leq \varepsilon$ is an abbreviation for the pair of inequalities $D_\infty(\mu\|\nu) \leq \varepsilon$ and $D_\infty(\nu\|\mu) \leq \varepsilon$, where $\mu$ and $\nu$ correspond to the two end nodes of the respective edges (e.g., $P$ and $P'$). The symbol $\omega$Unif represents a mixture with the uniform distribution (Algorithm 1, Line 3), where $\mathcal{U}(\cdot) = (1 - \omega) \cdot + \omega\mathrm{Unif}(\Theta)$. The notation $*\mathrm{Lap}$ refers to the convolution with the Laplace distribution, as in Algorithm 1, Line 4.

lemma that generalizes [LMW$^+$24, Lemma 8]. A small mixture of a uniform distribution ensures that the conditions of Lemma 6 hold. Therefore, Algorithm 1 consists of two key steps: mixture with the uniform distribution (Line 3), and the Laplace perturbation calibrated to $\delta$ (Line 4.) The formal proof of Theorem 1, along with proofs of the key lemmas, is deferred to Appendix F.

## F  Deferred Proofs in Section 2 and Appendix A

### F.1  Proof of Lemma 6

**Proof of Lemma 6** Denote $\mathrm{dist}(x, y) = \|x - y\|_q$. Denote $\mathcal{B}(c, r)$ the ball with center $c$, and $\ell_q$-radius $r$. Assume $\mathcal{B}(c, r) \in \Theta$. Without loss of generality, assume $\Delta < R$.

Fix $\Delta$ and set $\xi = p_{\min} \cdot \mathrm{Vol}\left(\mathbb{B}_{\ell_q}^d(1)\right) \cdot \left(\frac{r}{4R}\right)^d \cdot \Delta^d$, so that $d_{\mathrm{TV}}(\mu, \nu) < \xi$.

To prove the result that $W_\infty \leq \Delta$, we use the equivalent definition of $W_\infty$ in Lemma 7. By this definition, to prove $W_\infty(\mu, \nu) \leq \Delta$, it suffices to show that

$$\mu(A) \leq \nu(A^\Delta), \text{ for all open set } A \subseteq \Theta,$$

where we re-define $A^\Delta := \{x \in \mathbb{R}^d \mid \mathrm{dist}(x, A) \leq \Delta\}$. Note that by this definition, $A^\Delta$ might extend beyond $\Theta$. However, we still have $\nu(A^\Delta) = \nu(A^\Delta \cap \Theta)$, since $\nu$ is supported on $\Theta$.

Note that if $\nu(A^\Delta) = 1$, it is obvious that $\mu(A) \leq \nu(A^\Delta)$. When $A$ is an empty set, the proof is trivial. So we only consider nonempty open set $A \subseteq \Theta$ with $\nu(A^\Delta) < 1$.

Note that for an arbitrary open set $A \subseteq \Theta$, we have

$$\mu(A) \leq \nu(A) + d_{TV}(\mu, \nu) < \nu(A) + \xi.$$

Thus to prove $\mu(A) \leq \nu(A^\Delta)$, it suffices to prove $\nu(A) + \xi \leq \nu(A^\Delta)$, i.e., $\nu(A^\Delta \setminus A) \geq \xi$.

To prove $\nu(A^\Delta \setminus A) \geq \xi$, we construct a ball $K \subset \mathbb{R}^d$ of radius $\frac{r\Delta}{4R}$ satisfying the properties:

Property 1  $K$ is contained in the set $A^\Delta \setminus A$, i.e., $K \subseteq A^\Delta \setminus A$, and

Property 2  $K$ is contained in $\Theta$, i.e., $K \subseteq \Theta$. This guarantees that $\nu(K) = \nu(K \cap \Theta) \geq p_{\min} \cdot \mathrm{Vol}(K) = \xi$.

To construct the above ball $K$, we adopt the following strategy:

Step 1  First, select a point $y \in \Theta$ such that $\mathrm{dist}(y, A) = \Delta/2$.

Step 2 Then, construct the ball $K$ with center $c_1 = \omega c + (1 - \omega)y$, radius $\omega r$, where $\omega = \frac{\Delta}{4R}$ [3], i.e., $K = \mathcal{B}(c_1, \omega r)$. We will later show that this ball is contained in the convex hull of $\mathcal{B}(c, r) \cup \{y\}$. This construction is inspired by [MV22].

We first prove such point $y$ in Step 1 exists, that is, the set $\Theta \cap \{x \in \mathbb{R}^d | \text{dist}(x, A) = \Delta/2\}$ is nonempty. (Note that we only consider nonempty open set $A \subseteq \Theta$ with $\nu(A^\Delta) < 1$.)

We prove it by contradiction. If instead $\Theta \cap \{x \in \mathbb{R}^d | \text{dist}(x, A) = \Delta/2\} = \emptyset$, then for all $x \in \Theta$, $\text{dist}(x, A) \neq \Delta/2$. Due to the continuity of dist and the convexity of $\Theta$, we know that only one of these two statements holds:

- $\text{dist}(x, A) < \Delta/2$, for all $x \in \Theta$.

- $\text{dist}(x, A) > \Delta/2$, for all $x \in \Theta$.

Since $\emptyset \neq A \subseteq \Theta$, there exist $x' \in A \in \Theta$, such that $\text{dist}(x', A) = 0$. Therefore, the first statement holds. Thus $\Theta \subseteq A^{\Delta/2} \subseteq A^\Delta$, which contradicts $\nu(A^\Delta) < 1$.

Therefore, there exist $y \in \Theta$, such that $\text{dist}(y, A) = \Delta/2$, making Step 1 valid.

Next, we show that the $K$ we construct in Step 2 satisfies Property 1 and Property 2.

To prove $K \subseteq A^\Delta \setminus A$, let $x \in K = \mathcal{B}(c_1, \omega r)$. We show that $x \in A^\Delta$ and $x \notin A$. We have

$$
\begin{aligned}
\text{dist}(x, y) &\leq \text{dist}(x, c_1) + \text{dist}(c_1, y) \\
&= \|x - c_1\|_q + \|\omega c + (1 - \omega)y - y\|_q \\
&= \|x - c_1\|_q + \omega\|c - y\|_q \\
&\leq \omega r + \omega R \\
&= \frac{\Delta}{4R}(r + R) \\
&\leq \Delta/2
\end{aligned}
\tag{3}
$$

- ($x \notin A$). If $x \in A$, since $A$ is an open set and $\text{dist}(y, A) = \Delta/2$, we have that $\text{dist}(x, y) > \Delta/2$, which contradicts to (3). Therefore $x \notin A$.

- ($x \in A^\Delta$). We have $\text{dist}(x, A) \leq \text{dist}(x, y) + \text{dist}(y, A) \overset{(3)}{\leq} \Delta$, implying that $x \in A^\Delta$.

To prove $K \subseteq \Theta$, take any $x \in K = \mathcal{B}(c_1, \omega r)$, we show that $x \in \Theta$. Write $x = c_1 + \omega r\mathbf{v}$, where $\|\mathbf{v}\|_q \leq 1$. We have

$$
x = \omega c + (1 - \omega)y + \omega r\mathbf{v} = \omega(c + r\mathbf{v}) + (1 - \omega)y.
$$

Since $c + r\mathbf{v} \in \mathcal{B}(c, r) \subseteq \Theta$, $y \in \Theta$, and $0 < \omega = \frac{\Delta}{4R} < 1$, by the convexity of $\Theta$, we have $x \in \Theta$, which completes the proof.

In particular, if $\Theta$ is an $\ell_q$-ball centered at $c$, say $\Theta = c + \mathbb{B}_{\ell_q}^d(r)$, then the $\omega$ in Step 2 can be chosen as $\omega = \frac{\Delta}{4r}$, which similarly follows, since $\|c - x\|_p \leq r$ for any $x \in \Theta$. This improves the conversion by:

$$
\text{If} \quad d_{\text{TV}}(\mu, \nu) < p_{\min} \cdot \text{Vol}(\mathbb{B}_{\ell_q}^d(1)) \cdot \left(\frac{1}{4}\right)^d \cdot \Delta^d, \quad \text{then} \quad W_\infty(\mu, \nu) \leq \Delta.
\tag{4}
$$

∎

## F.2 Proof of Lemma 15

**Proof of Lemma 15** denote $P = \mu * \text{Laplace}^{\otimes d}\left(\frac{\Delta}{\varepsilon}\right)$, $Q = \nu * \text{Laplace}^{\otimes d}\left(\frac{\Delta}{\varepsilon}\right)$. Then, $P$ and $Q$ are absolutely continuous with respect to the Lebesgue measure. This is because for any Lebesgue zero measure set $S$, $P(S) = \int_x \mathbb{P}_{\text{Laplace}}(S - x)\text{d}\mu(x)$, where $S - x \doteq \{y | x + y \in S\}$, and $\mathbb{P}_{\text{Laplace}}$ denotes the probability measure of $\text{Laplace}^{\otimes d}\left(\frac{\Delta}{\varepsilon}\right)$. Since $S - x$ is zero measure for all $x \in \mathbb{R}^d$, and $\mathbb{P}_{\text{Laplace}}$ is absolutely continuous w.r.t. the Lebesgue measure, we have $\mathbb{P}_{\text{Laplace}}(S - x)$ for all

---

[3]We note that the symbol $\omega$ in Section F.1 is distinct from the $\omega$ used in other parts of the paper. This distinction is an intentional abuse of notation for clarity within specific contexts.

$x \in \mathbb{R}^d$. Therefore, $P(S) = 0$. Thus $P$ is absolutely continuous w.r.t. the Lebesgue measure, and $Q$ similarly follows.

Denote $p$ and $q$ the probability density function of $P$ and $Q$ respectively. Since $W_\infty^{\ell_1}(\mu, \nu) \leq \Delta$, by [GS84, Proposition 1], there exists $\gamma^* \in \Gamma_c(\mu, \nu)$, such that $\mathbb{P}_{(u,v) \sim \gamma^*}[\|u - v\|_1 > \Delta] = \gamma^*\{\|u - v\|_1 > \Delta\} = 0$. By the definition of the max divergence (Theorem 6 of [VEH14]), we have

$$e^{D_\infty(P\|Q)} = \operatorname*{ess\,sup}_{x \sim P} \frac{p(x)}{q(x)} \hspace{3cm} \text{([VEH14, Theorem 6])}$$

$$\leq \operatorname*{ess\,sup}_{x \sim P} \frac{\int_{\mathbb{R}^d} e^{-\frac{\|x-u\|_1}{b}} \mathrm{d}\mu(u)}{\int_{\mathbb{R}^d} e^{-\frac{\|x-v\|_1}{b}} \mathrm{d}\nu(v)} \hspace{2cm} \text{(Convolution theorem, PDF of Laplace)}$$

$$\leq \operatorname*{ess\,sup}_{x \sim P} \frac{\int_{\mathbb{R}^d \times \mathbb{R}^d} e^{-\frac{\|x-u\|_1}{b}} \mathrm{d}\gamma^*(u,v)}{\int_{\mathbb{R}^d \times \mathbb{R}^d} e^{-\frac{\|x-v\|_1}{b}} \mathrm{d}\gamma^*(u,v)} \hspace{2cm} (\gamma^* \in \Gamma_c(\mu, \nu))$$

$$\leq \operatorname*{ess\,sup}_{x \sim P} \frac{\int_{\mathbb{R}^d \times \mathbb{R}^d} e^{-\frac{\|x-v\|_1}{b} + \frac{\|u-v\|_1}{b}} \mathrm{d}\gamma^*(u,v)}{\int_{\mathbb{R}^d \times \mathbb{R}^d} e^{-\frac{\|x-v\|_1}{b}} \mathrm{d}\gamma^*(u,v)} \hspace{2cm} \text{(triangle's inequality)}$$

$$\leq \operatorname*{ess\,sup}_{x \sim P} \frac{\int_{\|u-v\|_1 \leq \Delta} e^{-\frac{\|x-v\|_1}{b} + \frac{\|u-v\|_1}{b}} \mathrm{d}\gamma^*(u,v) + \int_{\|u-v\|_1 > \Delta} e^{-\frac{\|x-v\|_1}{b} + \frac{\|u-v\|_1}{b}} \mathrm{d}\gamma^*(u,v)}{\int_{\mathbb{R}^d \times \mathbb{R}^d} e^{-\frac{\|x-v\|_1}{b}} \mathrm{d}\gamma^*(u,v)}$$

$$\leq \operatorname*{ess\,sup}_{x \sim P} \frac{\int_{\|u-v\|_1 \leq \Delta} e^{-\frac{\|x-v\|_1}{b} + \frac{\|u-v\|_1}{b}} \mathrm{d}\gamma^*(u,v)}{\int_{\mathbb{R}^d \times \mathbb{R}^d} e^{-\frac{\|x-v\|_1}{b}} \mathrm{d}\gamma^*(u,v)} \hspace{2cm} (\gamma^*\{\|u - v\|_1 > \Delta\} = 0)$$

$$\leq \operatorname*{ess\,sup}_{x \sim P} \frac{e^{\frac{\Delta}{b}} \int_{\|u-v\|_1 \leq \Delta} e^{-\frac{\|x-v\|_1}{b}} \mathrm{d}\gamma^*(u,v)}{\int_{\mathbb{R}^d \times \mathbb{R}^d} e^{-\frac{\|x-v\|_1}{b}} \mathrm{d}\gamma^*(u,v)}$$

$$\leq \operatorname*{ess\,sup}_{x \sim P} \frac{e^{\frac{\Delta}{b}} \int_{\mathbb{R}^d \times \mathbb{R}^d} e^{-\frac{\|x-v\|_1}{b}} \mathrm{d}\gamma^*(u,v)}{\int_{\mathbb{R}^d \times \mathbb{R}^d} e^{-\frac{\|x-v\|_1}{b}} \mathrm{d}\gamma^*(u,v)}$$

$$= e^{\frac{\Delta}{b}}$$

∎

### F.3 Proof of Lemma 16

**Proof of Lemma 16** Since $D_\infty(\mu\|\pi) < \infty$, for any measurable set $S \in \mathcal{F}$, such that $\mu(S) > 0$, we have $\pi(S) > 0$. That is, $\{S \in \mathcal{F} \mid \mu(S) > 0\} \subseteq \{S \in \mathcal{F} \mid \pi(S) > 0\}$. By Definition 11,

$$D_\infty(\mu, \nu) = \ln \sup_{S \in \mathcal{F}, \, \mu(S) > 0} \left[\frac{\mu(S)}{\nu(S)}\right]$$

$$= \ln \sup_{S \in \mathcal{F}, \, \mu(S) > 0, \, \pi(S) > 0} \left[\frac{\mu(S)\pi(S)}{\pi(S)\nu(S)}\right]$$

$$\leq \ln \sup_{S \in \mathcal{F}, \, \mu(S) > 0} \left[\frac{\mu(S)}{\pi(S)}\right] + \ln \sup_{S \in \mathcal{F}, \, \pi(S) > 0} \left[\frac{\pi(S)}{\nu(S)}\right]$$

$$= D_\infty(\mu\|\pi) + D_\infty(\pi\|\nu).$$

∎

### F.4 Proof of Theorem 1 and Corollary 4

We first provide proof of the privacy guarantee by reorganizing the proof sketch in Section E, as illustrated in Figure 4.

Let $D$ and $D'$ be neighboring datasets, and let $\mathcal{M}$ be an $(\varepsilon, \delta)$-DP mechanism as in Section E. By the equivalence definition of approximate DP (Lemma 14), there exists a hypothetical distribution pair

$P, P'$ such that

$$d_{\mathrm{TV}}(\mathcal{M}(D), P) \leq \frac{\delta}{e^\varepsilon + 1}, d_{\mathrm{TV}}(\mathcal{M}(D'), P') \leq \frac{\delta}{e^\varepsilon + 1}, D_\infty(P'\|P) \leq \varepsilon, \text{ and } D_\infty(P\|P') \leq \varepsilon$$

Note that $P$ and $P'$ can both depend on $D$ and $D'$, i.e., $P = P(D, D')$ and $P' = P'(D, D')$, rather than simply $P = P(D)$ and $P' = P'(D')$.

Denote $\mathcal{U}(\cdot) = (1 - \omega) \cdot + \omega \mathrm{Unif}(\Theta)$. After the uniform mixture step (Line 3), the distributions $\mathcal{U}(\mathcal{M}(D)), \mathcal{U}(\mathcal{M}(D')), \mathcal{U}(P), \mathcal{U}(P')$ all satisfy the assumption in Lemma 6 with

$$p_{\min} \geq \frac{\omega}{\mathrm{Vol}(\Theta)} \geq \frac{\omega}{(R/2)^d \mathrm{Vol}\left(\mathbb{B}_{\ell_q}^d(1)\right)}.$$

Therefore, by Lemma 6 (with the case of $\Theta$ is an $\ell_q$ ball) and the fact that $\|\cdot\|_1 \leq d^{1-\frac{1}{q}} \|\cdot\|_q$, we have

$$W_\infty^{\ell_1}(\mathcal{U}(\mathcal{M}(D)), \mathcal{U}(P)) \leq \Delta, \text{ and } W_\infty^{\ell_1}(\mathcal{U}(\mathcal{M}(D')), \mathcal{U}(P')) \leq \Delta,$$

where $\Delta$ is defined in Line 2 in Algorithm 1. Therefore, by Lemma 15, we have

$$D_\infty\left(\mathcal{U}(\mathcal{M}(D)) * \mathrm{Laplace}^{\otimes d}\left(\tfrac{2\Delta}{\varepsilon'}\right) \| \mathcal{U}(P) * \mathrm{Laplace}^{\otimes d}\left(\tfrac{2\Delta}{\varepsilon'}\right)\right) \leq \varepsilon'/2,$$

$$D_\infty\left(\mathcal{U}(P) * \mathrm{Laplace}^{\otimes d}\left(\tfrac{2\Delta}{\varepsilon'}\right) \| \mathcal{U}(\mathcal{M}(D)) * \mathrm{Laplace}^{\otimes d}\left(\tfrac{2\Delta}{\varepsilon'}\right)\right) \leq \varepsilon'/2,$$

$$D_\infty\left(\mathcal{U}(\mathcal{M}(D')) * \mathrm{Laplace}^{\otimes d}\left(\tfrac{2\Delta}{\varepsilon'}\right) \| \mathcal{U}(P') * \mathrm{Laplace}^{\otimes d}\left(\tfrac{2\Delta}{\varepsilon'}\right)\right) \leq \varepsilon'/2,$$

$$D_\infty\left(\mathcal{U}(P') * \mathrm{Laplace}^{\otimes d}\left(\tfrac{2\Delta}{\varepsilon'}\right) \| \mathcal{U}(\mathcal{M}(D')) * \mathrm{Laplace}^{\otimes d}\left(\tfrac{2\Delta}{\varepsilon'}\right)\right) \leq \varepsilon'/2.$$

Finally, with the weak triangle inequality (Lemma 16), we conclude that the output of Algorithm 1 satisfies $(\varepsilon + \varepsilon')$-DP.

Next, we provide the utility bound. Let $u \sim \mathrm{Unif}(0, 1)$ in Line 3 of Algorithm 1.

$$\mathbb{E}\|x_{\mathrm{pure}} - x_{\mathrm{apx}}\|_q = \mathbb{P}(u > \omega)\mathbb{E}\left[\|x_{\mathrm{pure}} - x_{\mathrm{apx}}\|_q \mid u > \omega\right] + \mathbb{P}(u \leq \omega)\mathbb{E}\left[\|x_{\mathrm{pure}} - x_{\mathrm{apx}}\|_q \mid u \leq \omega\right]$$

$$(5)$$

$$\leq \mathbb{E}\left[\left(\sum_{i=1}^d |z_i|^q\right)^{\frac{1}{q}}\right] + \omega R \qquad\qquad (z_i \sim \mathrm{Laplace}\left(\tfrac{2\Delta}{\varepsilon'}\right))$$

$$\leq \left(\mathbb{E}\left[\sum_{i=1}^d |z_i|^q\right]\right)^{\frac{1}{q}} + \omega R \qquad\qquad \text{(Jensen's inequality)}$$

$$= (d\Gamma(q+1))^{\frac{1}{q}} \frac{2\Delta}{\varepsilon'} + \omega R \qquad\qquad (6)$$

$$\leq \frac{4dqR}{\varepsilon'}\left(\frac{\delta}{2\omega}\right)^{\frac{1}{d}} + \omega R \qquad\qquad (7)$$

**Remark 18** *We can remove the dependence on $q$ in the above bound by controlling $\|x_{\mathrm{pure}} - x_{\mathrm{apx}}\|_1$ via concentration inequalities for sub-exponential random vectors.*

**Proof of Corollary 4** Since $\Theta$ is an $\ell_q$ ball, we have $R = 2r = C$. The proof follows directly from Theorem 1 by taking $q = 2$. ∎

### F.5   Proof of Theorem 2

**Proof of Theorem 2** Notice that BinMap is a data-independent deterministic function, thus by post-processing, $z_{\mathrm{bin}} = \mathrm{BinMap}(u_{\mathrm{apx}})$ maintains $(\varepsilon, \delta)$-DP.

We consider the $\ell_1$ norm, i.e., $q = 1$. Let $a = (\frac{1}{2}, \ldots, \frac{1}{2})$. For unit cube $[0, 1]^d$, we have $a + \mathbb{B}_{\ell_q}^d(\frac{1}{2}) \subseteq [0, 1]^d \subseteq a + \mathbb{B}_{\ell_q}^d(\frac{d}{2})$. That is, $[0, 1]^d$ satisfies Assumption 1 with $R = \frac{d}{2}$, $r = \frac{1}{2}$, and $q = 1$. Therefore, by Theorem 1, $z_{\mathrm{pure}}$ satisfies $(2\varepsilon, 0)$-DP. After the post-processing, the output of Algorithm 2 maintains $(2\varepsilon, 0)$-DP.

For the utility, by $\delta < \frac{\varepsilon^d}{(2d)^{3d}}$ and Line 2 of Algorithm 1, we have $\Delta \leq \frac{\varepsilon}{4d}$.

Let $y_i^{\text{Laplace}} \stackrel{\text{i.i.d.}}{\sim} \text{Laplace}(2\Delta/\varepsilon')$, be the noise added in Line 4 in Algorithm 1. Then for any $i$,

$$\mathbb{P}(y_i^{\text{Laplace}} \geq t) = \frac{1}{2} e^{-\frac{t}{2\Delta/\varepsilon}} \leq \frac{1}{2} e^{-2dt}.$$

Thus,

$$\mathbb{P}\left(y_i^{\text{Laplace}} > 0.5, \forall i = 1, \ldots, d\right) = (1 - \frac{1}{2}e^{-d})^d \geq 1 - \frac{d}{2}e^{-d}, \tag{8}$$

where the last inequality is by Bernoulli's inequality.

Since the rounding function is defined as $\text{Round}_{\{0,1\}^d}(\mathbf{x}) = (\mathbf{1}(x_i \geq 0.5))_{i=1}^d$, we observe that $z_{\text{bin}} = z_{\text{pure}}$ if and only if the following conditions hold simultaneously:

1. $x \sim \text{Unif}(\Theta)$ is sampled in Line 3 of Algorithm 1.

2. For all $i \in [d]$, if $z_{\text{bin}}^{(i)} = 0$, then $y_i^{\text{Lap}} < 0.5$.

3. For all $i \in [d]$, if $z_{\text{bin}}^{(i)} = 1$, then $y_i^{\text{Lap}} > -0.5$.

By the symmetry of Laplace noise, applying Eq. (8), and using the union bound, we obtain

$$\mathbb{P}\left(z_{\text{bin}} = z_{\text{pure}}\right) \geq 1 - \omega - \frac{d}{2}e^{-d} = 1 - 2^{-d} - \frac{d}{2}e^{-d}.$$

∎

## G  Further Discussion of the Optimality of Our Purification Results

In this section, we examine the optimality of the utility guarantees achieved by our purified algorithm, as summarized in Table 1. We compare our results against known information-theoretic lower bounds and the best existing upper bounds, discussing each setting in turn.

For the *DP-ERM* setting (Rows 1 and 2), our bounds (Theorem 3) match the lower bounds reported in [BST14, Table 1, Rows 1 and 3], up to logarithmic factors.

In the *DP-Frank-Wolfe* setting (Row 3), our guarantee matches with the lower bound established in Lemma 30, again up to logarithmic terms.

For the *PTR-type* setting (Row 4), to the best of our knowledge, no pure-DP mechanism has previously been developed in this regime, and thus no direct baseline is available for comparison.

For the *Mode Release* task (Row 5), our result (Theorem 6) matches the lower bound from [CHS14, Proposition 1], up to logarithmic factors.

For *Regression* (Row 6), assuming bounded data and parameter domains, our result (Theorem 7) agrees with the lower bounds in [BST14, Table 1] with respect to $n, d, \varepsilon$, and $\lambda$, up to logarithmic factors and constants depending on the data and parameter radii.

Finally, for *Query Release* (Row 7), our utility guarantee (Theorem 8) matches that of the SmallDB and Private Multiplicative Weights algorithms up to logarithmic factors [DR$^+$14, BLR13], which represent the current state of the art for pure-DP query release. Whether this rate can be further improved remains an open question [NS21].

## H  Deferred Proofs for DP-SGD

The study of DP-ERM is extensive; for other notable contributions, see, e.g., [RBHT12, FTS17, INS$^+$19, SSTT21, MBST22, GTU22, GLL22, RKW23, KLT24].

### H.1  Algorithms and Notations

Let $f(\theta; x)$ represent the individual loss function. $\mathcal{L}(\theta) := \frac{1}{n}\sum_{i=1}^n f(\theta; x_i)$ and $\mathcal{L}^* = \min_{\theta \in \mathcal{C}} \mathcal{L}(\theta)$. We also denote $F(\theta) := \sum_{i=1}^n f(\theta; x_i)$ and $F^* = \min_{\theta \in \mathcal{C}} F(\theta)$. $L$-Lipschitz, $\beta$-smooth, or $\lambda$-strongly convex are all w.r.t. individual the loss function $f$. The parameter space $\Theta$ in Algorithm 4 is selected as the $\ell_2$-ball with diameter $C$ that contains $\mathcal{C}$.

**Algorithm 3:** Differentially Private SGD [ACG⁺16]

---

**2 Input:** Dataset $D = \{x_1, \ldots, x_n\}$, loss function $f : \mathcal{C} \times D \to \mathbb{R}$, parameters: learning rate $\eta_t$, noise scale $\sigma$, subsampling rate $\gamma$, Lipschitz constant $L$, parameter space $\mathcal{C} \in \mathbb{R}^d$

**4** Initialize $\theta_0 \in \mathcal{C}$ randomly

**6 for** $t \in [T]$ **do**

**8**     Sample a subset $S_t$ by selecting a $\gamma$ fraction of the dataset without replacement

**10**     **Compute gradient: for** *each* $i \in S_t$ **do**

**12**       $g_t(x_i) \leftarrow \nabla_\theta f(\theta_t; x_i)$

**14**     **Aggregate and add noise:** $\hat{g}_t \leftarrow \frac{1}{\gamma} \left( \sum_{i \in S_t} g_t(x_i) + \sigma \mathcal{N}(0, \mathbf{I}_d) \right)$

**16**     **Descent:** $\theta_{t+1} \leftarrow \text{Proj}_{\mathcal{C}} (\theta_t - \eta_t \tilde{g}_t)$

**18 Output:** $\theta_{\text{out}} = \frac{1}{T} \sum_{t=1}^{T} \theta_t$ if $f$ is convex; $\theta_{\text{out}} = \frac{2}{T(T+1)} \sum_{t=1}^{T} t\theta_t$ if $f$ is strongly convex.

---

**Algorithm 4:** Pure DP SGD

---

**2 Input:** Output from DP-SGD Algorithm 3 $\theta_{apx}$, parameter space $\Theta$, privacy parameter $\varepsilon'$ and $\delta$, mixture level $\omega$

**4** $\theta_{\text{pure}} \leftarrow \mathcal{A}_{\text{pure}}(\theta_{apx}, \Theta, \varepsilon', \delta, \omega)$          ▷ Algorithm 1

**6 Output:** $\theta_{\text{pure}}$

---

### H.2 Noisy Gradient Descent Using Laplace Mechanism

**Lemma 19 (Laplace Noisy Gradient Descent)** *Let the loss function* $f : \mathcal{X} \to \mathbb{R}^d$ *be convex and* $L$-*Lipschitz with respect to* $\|\cdot\|_2$ *and* $\max_{X \sim X'} \|\nabla f(X) - \nabla f(X')\|_1 \leq \Delta_1$. *Suppose the parameter space* $\mathcal{C} \subset \mathbb{R}^d$ *is convex with an* $\ell_2$ *diameter of at most* $C$. *Running **full-batch** noisy gradient descent with learning rate* $\eta = \frac{C}{\sqrt{T(n^2 L^2 + 2d\sigma^2)}}$, *number of iterations* $T = \frac{\varepsilon n L}{\Delta_1 \sqrt{d}}$, *and Laplace noise with parameter* $\sigma = \frac{\Delta_1 T}{\varepsilon}$, *satisfies* $\varepsilon$-*DP. Moreover,*

$$\mathbb{E}_\mathcal{A} \left( \mathcal{L}(\bar{\theta}) - \mathcal{L}^* \right) \leq \mathcal{O} \left( \frac{C \Delta_1^{1/2} L^{1/2} d^{1/4}}{n^{1/2} \varepsilon^{1/2}} \right).$$

*and the total number of gradient calculation is* $\frac{n^2 \varepsilon L \sqrt{d}}{\Delta_1}$. *Without further assumptions on* $\nabla f$, *we have:*

$$\mathbb{E} \left( \mathcal{L}(\bar{\theta}) - \mathcal{L}^* \right) \leq \mathcal{O} \left( \frac{C L d^{1/2}}{n^{1/2} \varepsilon^{1/2}} \right)$$

**Proof** Suppose we run $T$ iterations and the final privacy budget is $\varepsilon$. Then, the privacy budget per iteration is $\varepsilon_0 = \frac{\varepsilon}{T}$, and the parameter of the additive Laplace noise is $\sigma = \Delta_1/\varepsilon_0 = \Delta_1 T/\varepsilon$. By [GG23, Theorem 9.6], we have

$$\mathbb{E} \left( F \left( \frac{1}{T} \sum_{t=1}^{T} \theta_t \right) - F^* \right) \leq \frac{C^2}{T\eta} + \eta(n^2 L^2 + 2d\sigma^2)$$

$$\leq \mathcal{O} \left( \frac{CnL}{\sqrt{T}} + \frac{C\sigma\sqrt{d}}{\sqrt{T}} \right)$$

$$= \mathcal{O} \left( \frac{CnL}{\sqrt{T}} + \frac{C\Delta_1\sqrt{dT}}{\varepsilon} \right)$$

$$\leq \mathcal{O} \left( \frac{C(nL\Delta_1)^{1/2} d^{1/4}}{\varepsilon^{1/2}} \right)$$

where the second inequality is obtained by choosing learning rate $\eta = \frac{C}{\sqrt{T(n^2 L^2 + 2d\sigma^2)}}$ and the fact $\sqrt{a+b} < \sqrt{a} + \sqrt{b}$ for any positive $a$ and $b$. The last inequality is by setting $T = \frac{\varepsilon n L}{\Delta_1 \sqrt{d}}$. Divide

both sides by $n$, we have:

$$\mathbb{E}\left(\mathcal{L}(\bar{\theta}) - \mathcal{L}^*\right) \leq \mathcal{O}\left(\frac{C\Delta_1^{1/2}L^{1/2}d^{1/4}}{n^{1/2}\varepsilon^{1/2}}\right) \tag{9}$$

Without additional information, the best upper bound for $\Delta_1$ is $\sqrt{d}\Delta_2 = \sqrt{d}L$. Plugging this bound to Eq. (9) yields:

$$\mathbb{E}\left(\mathcal{L}(\bar{\theta}) - \mathcal{L}^*\right) \leq \mathcal{O}\left(\frac{CLd^{1/2}}{n^{1/2}\varepsilon^{1/2}}\right)$$

∎

## H.3 Analysis of DP-SGD

### H.3.1 Privacy Accounting Results

Our privacy accounting for DP-SGD is based on Rényi differential privacy [Mir17]. Before stating the privacy accounting result (Corollary 24), we define Rényi Differential privacy and its variant, zero concentrated Differential Privacy [BS16].

**Definition 20 (Rényi differential privacy)** *A randomized mechanism $\mathcal{M}$ satisfies $(\alpha, \varepsilon(\alpha))$-Rényi Differential Privacy (RDP) if for all neighboring datasets $D, D'$ and for all $\alpha \geq 1$,*

$$D_\alpha(\mathcal{M}(D)\|\mathcal{M}(D')) \leq \varepsilon(\alpha),$$

*where $D_\alpha(P\|Q)$ denotes the $\alpha$-Rényi divergence when $\alpha > 1$; Kullback-Leibler divergence when $\alpha = 1$; max-divergence when $\alpha = \infty$. We refer the readers to [Mir17, Definition 3] for a complete description.*

Zero Concentrated Differential Privacy is a special case of Rényi differential privacy when Rényi divergence grows linearly with $\alpha$, e.g., Gaussian Mechanism.

**Definition 21 (Zero Concentrated Differential Privacy (zCDP))** *A randomized mechanism $\mathcal{M}$ satisfies $\rho$-zCDP if for all neighboring datasets $D, D'$ and for all $\alpha > 1$,*

$$D_\alpha(\mathcal{M}(D)\|\mathcal{M}(D')) \leq \rho\alpha,$$

*where $D_\alpha(P\|Q)$ is the Rényi divergence of order $\alpha$.*

**Lemma 22 ($\rho$-zCDP to $(\varepsilon, \delta)$-DP)** *If mechanism $\mathcal{M}$ satisfies $\rho$-zCDP, then for any $\delta \in (0, 1)$, $\mathcal{M}$ satisfies $(\rho + 2\sqrt{\rho \log(1/\delta)}, \delta)$-DP.*

**Proof** Since $\mathcal{M}$ satisfies $\rho$-zCDP, $\mathcal{M}$ also satisfies $(\alpha, \rho\alpha)$-RDP for any $\alpha > 1$. By [Mir17, Proposition 3], for any $\delta \in (0, 1)$, $\mathcal{M}$ satisfies $(\rho\alpha + \frac{\log(1/\delta)}{\alpha-1}, \delta)$-DP. The remaining proof is by minimizing $f(\alpha) := \rho\alpha + \frac{\log(1/\delta)}{\alpha-1}$ for $\alpha > 1$. The minimum is $\rho + 2\sqrt{\rho \log(1/\delta)}$, by choosing minimizer $\alpha^* = 1 + \sqrt{\frac{\log(1/\delta)}{\rho}}$. ∎

We now introduce RDP accounting results for DP-SGD. We first demonstrate RDP guarantee for one-step DP-SGD (i.e. sub-sampled Gaussian mechanism, Lemma 23) then show the privacy guarantee for multi-step DP-SGD through RDP composition (Corollary 24).

**Lemma 23 (RDP guarantee for subsampled Gaussian Mechanism, Theorem 11 in [BDRS18])** *Let $\mathcal{M}$ be a $\rho$-zCDP Gaussian mechanism, and $\mathcal{S}_\gamma$ be a subsampling procedure on the dataset with subsampling rate $\gamma$, then the subsampled Gaussian mechanism $\mathcal{M} \circ \mathcal{S}_\gamma$ satisfies $(\alpha, \varepsilon(\alpha))$-RDP with:*

$$\varepsilon(\alpha) \geq 13\gamma^2\rho\alpha, \quad \text{for any } \alpha \leq \frac{\log(1/\gamma)}{4\rho} \tag{10}$$

**Corollary 24 (RDP guarantee for DP-SGD)** *Let $\mathcal{M}$ be a Gaussian mechanism satisfying $\rho_0$-zCDP and $\mathcal{S}_\gamma$ be a subsampling procedure on the dataset with subsampling rate $\gamma$, then $T$-fold (adaptive) composition of subsampled Gaussian mechanism, $\mathcal{M}_T := \underbrace{(\mathcal{M} \circ \mathcal{S}_\gamma) \circ \ldots \circ (\mathcal{M} \circ \mathcal{S}_\gamma)}_{T \text{ times}}$, satisfies $(\alpha, \varepsilon(\alpha))$-RDP with*

$$\varepsilon(\alpha) \geq 13\gamma^2\rho_0 T\alpha, \quad \text{for any } \alpha \leq \frac{\log(1/\gamma)}{4\rho_0}. \tag{11}$$

*Denote $\rho := 13\gamma^2\rho_0 T$ for a shorthand, if further $\rho_0 \leq \frac{\log(1/\gamma)}{4(1+\sqrt{\frac{\log(1/\delta)}{\rho}})}$, the composed mechanism $\mathcal{M}_T$ satisfies $(\rho + 2\sqrt{\rho\log(1/\delta)}, \delta)$-DP for any $\delta \in (0,1)$.*

**Proof** By RDP accounting of subsampled gaussian mechanisms Lemma 23 and the composition theorem for RDP ([Mir17, Proposition 1]), we have $\mathcal{M}_T$ satisfies $(\alpha, 13\gamma^2\rho_0 T\alpha)$-RDP for any $\alpha \in (1, \frac{\log(1/\gamma)}{4\rho_0})$. Denote $\rho := 13\gamma^2\rho_0 T$. By Lemma 22, if $1 + \sqrt{\frac{\log(1/\delta)}{\rho}} \leq \frac{\log(1/\gamma)}{4\rho_0}$ (i.e., the optimal $\alpha^* \leq \frac{\log(1/\gamma)}{4\rho_0}$), we have $\mathcal{M}_T$ satisfies $(\rho + 2\sqrt{\rho\log(1/\delta)}, \delta)$-DP. ∎

### H.3.2 Convex and Lipschitz case

In this section, we analyze the convergence of DP-SGD in convex and Lipschitz settings.

**Lemma 25 (Convergence of DP-SGD in convex and $L$-Lipschitz setting)** *Assume that the individual loss function $f$ is convex and L-Lipschitz. Running DP-SGD with parameters $\gamma = \frac{2\sqrt{d\log(1/\delta)}}{n\sqrt{\varepsilon}}$, $\sigma^2 = \frac{416L^2\log(1/\delta)}{\varepsilon}$, $T = \frac{n^2\varepsilon^2}{d\log(1/\delta)}$, $\eta = \sqrt{\frac{C^2}{T(n^2L^2 + d\sigma^2/\gamma^2 + nL^2/\gamma)}}$ satisfies $(\varepsilon, \delta)$-DP for any $\varepsilon \leq (d \wedge 8)\log(1/\delta)$. Moreover,*

$$\mathbb{E}\left[F(\bar{\theta}_T)\right] - F^* \leq \mathcal{O}\left(\frac{CLd^{1/2}\log^{1/2}(1/\delta)}{\varepsilon}\right),$$

*with $\bar{\theta}_T$ being the averaged estimator. In addition, the number of incremental gradient calls is*

$$\mathcal{G} = \frac{2n^2\varepsilon^{3/2}}{\sqrt{d\log(1/\delta)}}.$$

**Proof** We first state the privacy guarantee. Since each gaussian mechanism satisfies $L^2/2\sigma^2$-zCDP, by Corollary 24, the composed mechanism satisfies $\rho$-zCDP with $\rho = \frac{13L^2\gamma^2 T}{2\sigma^2} = \frac{\varepsilon^2}{16\log(1/\delta)}$, and thus $(\varepsilon, \delta)$-approximate DP.

Let $\hat{g}_t$ be the output from noisy gradient oracle with variance $\sigma^2$ and subsampling rate $\gamma$ (line 7 of Algorithm 3). The variance of the gradient estimator can be upper bounded by:

$$\mathbb{E}[\|\hat{g}_t - \mathbb{E}(\hat{g}_t)\|_2^2] \leq \frac{d\sigma^2}{\gamma^2} + \frac{nL^2}{\gamma}.$$

By [GG23, Theorem 9.6], we have:

$$\mathbb{E}\left[\frac{1}{T}\sum_t (F(\theta_t) - F^*)\right] \leq \frac{C^2}{T\eta} + \eta\mathbb{E}\left[\frac{1}{T}\sum_{t=1}^T \|\nabla F(\theta_t)\|_2^2\right] + \eta\left(\frac{d\sigma^2}{\gamma^2} + \frac{nL^2}{\gamma}\right)$$

$$\leq \frac{C^2}{T\eta} + \eta\left(n^2L^2 + \frac{d\sigma^2}{\gamma^2} + \frac{nL^2}{\gamma}\right)$$

$$\leq \sqrt{\frac{C^2}{T}\left(n^2L^2 + \frac{d\sigma^2}{\gamma^2} + \frac{nL^2}{\gamma}\right)}$$

$$= \mathcal{O}\left(\sqrt{C^2L^2\left(\frac{n^2}{T} + \frac{d}{\rho} + \frac{n}{T\gamma}\right)}\right),$$

where the third inequality is by choosing $\eta = \sqrt{\frac{C^2}{T(n^2L^2 + \frac{d\sigma^2}{\gamma^2} + \frac{nL^2}{\gamma})}}$, and the forth line is by $\rho = 13T\gamma^2L^2/2\sigma^2$. By the choice of $T$ and $\gamma$, we have $\frac{d}{\rho} \geq \max\{\frac{n^2}{T}, \frac{n}{T\gamma}\}$. This implies:

$$\mathbb{E}\left[\frac{1}{T}\sum_t (F(\theta_t) - F^*)\right] \leq \mathcal{O}\left(\sqrt{C^2L^2\left(\frac{n^2}{T} + \frac{d}{\rho} + \frac{n}{T\gamma}\right)}\right) \leq \mathcal{O}\left(\frac{CLd^{1/2}}{\rho^{1/2}}\right).$$

Since the target approximate DP privacy budget $\varepsilon = 4\sqrt{\rho \log(1/\delta)}$, we have $\sqrt{\rho} = \frac{\varepsilon}{4\sqrt{\log(1/\delta)}}$. Plugging this into the bound above, we have:

$$\mathbb{E}\left[F(\bar{\theta}_T)\right] - F^* \leq \mathbb{E}\left[\frac{1}{T}\sum_t (F(\theta_t) - F^*)\right] \leq \mathcal{O}\left(\frac{CLd^{1/2}\log^{1/2}(1/\delta)}{\varepsilon}\right).$$

For the number of incremental gradient calls (denoted as $\mathcal{G}$), we have

$$\mathcal{G} = n\gamma T = \frac{2n^2\varepsilon^{3/2}}{\sqrt{d\log(1/\delta)}}.$$

$\blacksquare$

### H.3.3 Strongly Convex and Lipschitz case

**Lemma 26 (Convergence of DP-SGD in strongly convex and $L$-Lipschitz setting)** *Assume individual loss function $f$ is $\lambda$-strongly convex and $L$-Lipschitz. Running DP-SGD with parameters $\gamma = \frac{2\sqrt{d\log(1/\delta)}}{n\sqrt{\varepsilon}}$, $\sigma^2 = \frac{416L^2\log(1/\delta)}{\varepsilon}$, $T = \frac{n^2\varepsilon^2}{d\log(1/\delta)}$, $\eta_t = \frac{2}{n\lambda(t+1)}$ satisfies $(\varepsilon, \delta)$-DP for any $\varepsilon \leq (d \wedge 8)\log(1/\delta)$. Moreover,*

$$\mathbb{E}\left[F\left(\frac{2}{T(T+1)}\sum_{t=1}^T t\theta_t\right)\right] - F^* \leq \mathcal{O}\left(\frac{dL^2\log(1/\delta)}{n\lambda\varepsilon^2}\right),$$

*and the number of incremental gradient calls is*

$$\mathcal{G} = \frac{2n^2\varepsilon^{3/2}}{\sqrt{d\log(1/\delta)}}.$$

**Proof** The derivation of privacy guarantee follows the same procedure as Lemma 25. By Corollary 24, the total zCDP guarantee $\rho = \frac{13L^2\gamma^2T^2}{2\sigma^2}$. Choosing learning rate $\eta_t = \frac{2}{\Lambda(t+1)}$ and apply convergence result from [LJSB12], where $\Lambda = n\lambda$ be strong convex parameter of $F$, we have:

$$\mathbb{E}\left[F\left(\frac{2}{T(T+1)}\sum_{t=1}^T tx_t\right)\right] - F^* \leq \frac{2\mathbb{E}[\|\hat{g}_t\|_2^2]}{\Lambda(T+1)}$$

$$\leq \mathcal{O}\left(\frac{n^2L^2}{\Lambda T} + \frac{d\sigma^2}{\Lambda\gamma^2 T} + \frac{nL^2}{\Lambda\gamma T}\right)$$

$$= \mathcal{O}\left(\frac{L^2}{\Lambda}\left(\frac{n^2}{T} + \frac{d}{\rho} + \frac{n}{\gamma T}\right)\right).$$

where in the last line we use the fact that $\rho = \frac{13T\gamma^2L^2}{2\sigma^2}$. By the choice of $T$ and $\gamma$, we have $\frac{d}{\rho} \geq \max\{\frac{n^2}{T}, \frac{n}{T\gamma}\}$. This implies:

$$\mathbb{E}\left[F\left(\frac{2}{T(T+1)}\sum_{t=1}^T tx_t\right)\right] - F^* \leq \mathcal{O}\left(\frac{dL^2}{\Lambda\rho}\right) = \mathcal{O}\left(\frac{dL^2\log(1/\delta)}{n\lambda\varepsilon^2}\right),$$

where the last equality is using the fact that the target privacy budget $\varepsilon = 4\sqrt{\rho\log(1/\delta)}$.

For the number of incremental gradient calls (denote as $\mathcal{G}$), we have the same result as in Lemma 25:

$$\mathcal{G} = \frac{2n^2\varepsilon^{3/2}}{\sqrt{d\log(1/\delta)}}.$$

$\blacksquare$

## H.4 Analysis of Purified DP-SGD

**Lemma 27 (Error from Laplace perturbation)** *Suppose $x \in \mathbb{R}^d$ and $\tilde{x} = x + Lap^{\otimes d}(b)$, then*

$$\mathbb{E}[\|x - \tilde{x}\|_2] \leq \sqrt{2d}b.$$

**Proof**

$$\mathbb{E}[\|x - \tilde{x}\|_2] = \mathbb{E}\left[\sqrt{\|x - \tilde{x}\|_2^2}\right] \leq \sqrt{\mathbb{E}[\|x - \tilde{x}\|_2^2]} = \sqrt{2db^2}$$

∎

### H.4.1 Proof of Theorem 3

**Theorem 10 (Restatement of Theorem 3)** *Let the domain $\mathcal{C} \subset \mathbb{R}^d$ be a convex set with $\ell_2$ diameter $C$, and let $f(\cdot; x)$ be L-Lipschitz for all $x \in \mathcal{X}$. Algorithm 4 satisfies $2\varepsilon$-pure DP and with $\tilde{\mathcal{O}}(n^2\varepsilon^{3/2}d^{-1})$ incremental gradient calls, the output $\theta_{\text{out}}$ satisfies:*

1. *If $f(\cdot; x)$ is convex for all $x \in \mathcal{X}$, then $\mathbb{E}\left[\mathcal{L}(\theta_{\text{out}})\right] - \mathcal{L}^* \leq \tilde{\mathcal{O}}\left(CLd/n\varepsilon\right).$*

2. *If $f(\cdot; x)$ is $\lambda$-strongly convex for all $x \in \mathcal{X}$, then $\mathbb{E}\left[\mathcal{L}\left(\theta_{\text{out}}\right)\right] - \mathcal{L}(x^*) \leq \tilde{\mathcal{O}}\left(d^2L^2/n^2\lambda\varepsilon^2\right).$*

**Proof** Setting $\omega = \frac{1}{n^2}, \delta = \frac{2\omega}{2^{4d}d^dn^{2d}C^d} = 2^{1-4d}d^{-d}n^{-2d-2}C^{-d}$, we have

$$\log(1/\delta) = \mathcal{O}\left(d\log(2) + d\log(n) + d\log(d) + d\log(C)\right) = \tilde{\mathcal{O}}(d) \tag{12}$$

Applying Corollary 4 with $\omega, \delta$ defined above and choose $\varepsilon' = \varepsilon$, the additional error from purification can be upper bounded by $\frac{1}{n^2\varepsilon} + \frac{C}{n^2}$.

*(When $f$ is Convex and L-Lipschitz):* By Lemma 25 and dividing both sides by $n$:

$$\mathbb{E}\left[\mathcal{L}(\theta_{\text{out}})\right] - \mathcal{L}^* \leq \frac{L}{n^2\varepsilon} + \frac{CL}{n^2} + \mathcal{O}\left(\frac{CLd^{1/2}\log^{1/2}(1/\delta)}{n\varepsilon}\right)$$

$$= \tilde{\mathcal{O}}\left(\frac{L}{n^2\varepsilon} + \frac{CL}{n^2} + \frac{CLd}{n\varepsilon}\right)$$

*(When $f$ is $\lambda$-strongly Convex and L-Lipschitz):* By Lemma 26 and dividing both sides by $n$:

$$\mathbb{E}\left[\mathcal{L}(\theta_{\text{out}})\right] - \mathcal{L}^* \leq \frac{L}{n^2\varepsilon} + \frac{CL}{n^2} + \mathcal{O}\left(\frac{dL^2\log(1/\delta)}{n^2\lambda\varepsilon^2}\right)$$

$$= \tilde{\mathcal{O}}\left(\frac{L}{n^2\varepsilon} + \frac{CL}{n^2} + \frac{d^2L^2}{n^2\lambda\varepsilon^2}\right)$$

The number of incremental gradient calls for both cases:

$$\mathcal{G} = \frac{2n^2\varepsilon^{3/2}}{\sqrt{d\log(1/\delta)}} = \tilde{\mathcal{O}}\left(n^2\varepsilon^{3/2}d^{-1}\right)$$

∎

# I  Deferred Proofs for DP-Frank-Wolfe

## I.1  Approximate DP Frank-Wolfe Algorithm

Given a dataset $D = x_1, \ldots, x_n$ and a parameter space $\mathcal{C}$, we denote the individual loss function by $f : \mathcal{C} \times \mathcal{X} \to \mathbb{R}$. We define the average empirical loss as follows:

$$\mathcal{L}(\theta) := \frac{1}{n}\sum_{i=1}^{n} f(\theta; x_i) \tag{13}$$

For completeness, we state the Frank-Wolfe algorithm [FW+56] as follows.

The differential private version of Algorithm 5 is modified by using the exponential mechanism to select coordinates in each update. We follow the setting in [TGTZ15] with initialization at point zero.

---
**Algorithm 5:** Frank-Wolfe algorithm (Non-Private)
---
**2 Input:** $\mathcal{C} \subseteq \mathbb{R}^d$, loss function $\mathcal{L} : \mathcal{C} \to \mathbb{R}$, number of iterations $T$, stepsizes $\eta_t$.
**4** Choose an arbitrary $\theta_1$ from $\mathcal{C}$
**6 for** $t = 1$ *to* $\tilde{T} - 1$ **do**
**8**     Compute $\tilde{\theta}_t = \arg\min_{\theta \in \mathcal{C}} \langle \nabla \mathcal{L}(\theta_t), \theta - \theta_t \rangle$
**10**    Set $\theta_{t+1} = \theta_t + \eta_t(\tilde{\theta}_t - \theta_t)$
**12** Output $\theta_T$
---

---
**Algorithm 6:** Approximate DP Frank-Wolfe Algorithm [TGTZ15]
---
**2 Input:** Dataset $D = \{d_1, \ldots, d_n\}$, loss function $f$ defined in Eq. (13) with $\ell_1$-Lipschitz constant $L_1$, privacy parameters $(\varepsilon, \delta)$, convex set $\mathcal{C} = \text{conv}(S)$, $\|\mathcal{C}\|_1 = \max_{s \in S} \|s\|_1$.
**4** Initialize $\theta_0 \leftarrow \mathbf{0} \in \mathcal{C}$.
**6 for** $t = 0$ *to* $T - 1$ **do**
**8**    $\forall s \in S, \alpha_s \leftarrow \langle s, \nabla \mathcal{L}(\theta_t; \mathcal{D}) \rangle + \text{Lap}\left( \frac{L_1 \|\mathcal{C}\|_1 \sqrt{8T \log(1/\delta)}}{n\varepsilon} \right)$, where

     $\text{Lap}(\lambda) \sim \frac{1}{2\lambda} e^{-|x|/\lambda}$
**10**    $\tilde{\theta}_t \leftarrow \arg\min_{s \in S} \alpha_s$
**12**    $\theta_{t+1} \leftarrow (1 - \eta_t)\theta_t + \eta_t\tilde{\theta}_t$, where $\eta_t = \frac{2}{t+2}$
**14** Output $\theta_{\text{priv}} = \theta_T$
---

**Lemma 28 (Equation 21 of [TTZ14])** *Running Algorithm 6 for $T$ iterations yields:*

$$\mathbb{E}\left[ \mathcal{L}(\theta_T; D) - \min_{\theta \in \mathcal{C}} \mathcal{L}(\theta; D) \right] = O\left( \frac{\Gamma_{\mathcal{L}}}{T} + \frac{L_1 \|\mathcal{C}\|_1 \sqrt{8T \log(1/\delta)} \log(T L_1 \|\mathcal{C}\|_1 \cdot |S|)}{n\varepsilon} \right),$$

*where $\Gamma_{\mathcal{L}}$ is the curvature parameter [TGTZ15, Definition 2.1], which can be upper bounded by $\beta \|\mathcal{C}\|_1^2$ if the loss function $f$ is $\beta$-smooth [TTZ14].*

The sparsity of $\theta_T$ is given in the following lemma.

**Lemma 29 (Sparsity of DP Frank-Wolfe)** *Suppose $S \subset \{x \in \mathbb{R}^d \mid \text{nnz}(x) \leq s\}$. After running Algorithm 6 for $T$ iterates, the output $\theta_T$ is $Ts \wedge d$-sparse.*

**Proof** From Line 10 and 12 of Algorithm 6, we know $\text{nnz}(\theta_{t+1}) \leq \text{nnz}(\theta_t) + s$. Since $\text{nnz}(\theta_0) = 0$, we have $\text{nnz}(\theta_T) \leq Ts$. Since $\theta_T \in \mathbb{R}^d$, we have $\text{nnz}(\theta_T) \leq d$. Therefore, $\text{nnz}(\theta_T) \leq Ts \wedge d$. $\blacksquare$

### I.2 Pure DP Frank-Wolfe Algorithm

---
**Algorithm 7:** Pure DP Frank-Wolfe Algorithm
---
**2 Input:** Dataset $D$, loss function $\mathcal{L}$: defined in Eq. (13), DP parameter $\varepsilon$, a convex polytope $\mathcal{C} = \text{conv}(S)$, where $S$ is the vertices set, number of iterations $T$, a Gaussian random matrix $\Phi \in \mathbb{R}^{k \times d}$ constructed by Lemma 44 satisfying $(1/4, 4T)$-RWC with high probability.
**4** Set parameters $T = \tilde{\Theta}\left( \sqrt{\frac{\beta \|\mathcal{C}\|_1 n\varepsilon}{L_1}} \right)$, $k = \Theta\left( T \log\left(\frac{d}{T}\right) + \log n \right)$, $\omega = \frac{1}{n}$, $\delta = \frac{2\omega}{(nk)^k}$.
**6** $\theta_{\text{FW}} \leftarrow$ Algorithm 6                            $\triangleright (\varepsilon, \delta)$-DP FW
**8** $\theta_{\text{apx-}k} \leftarrow \Phi\theta_{\text{FW}} \in \mathbb{R}^k$                   $\triangleright$ Dimension reduction
**10** $\theta_{\text{pure-}k} \leftarrow \mathcal{A}_{\text{pure}}\left( x_{\text{apx}} = \text{Clip}_{2\|\mathcal{C}\|_1}^{\ell_2}(\theta_{\text{apx-}k}), \Theta = \mathcal{B}_{\ell_2}^k(2\|\mathcal{C}\|_1), \varepsilon' = \varepsilon, \delta, \omega \right)$    $\triangleright$ Algorithm 1
**12** $\theta_{\text{pure-}d} \leftarrow \mathcal{M}_{\text{rec}}(\theta_{\text{pure-}k}, \Phi, \xi)$         $\triangleright$ Algorithm 8, with $\xi$ defined in Eq. (14)
**14** $\theta_{\text{out}} = \text{Clip}_{\|\mathcal{C}\|_1}^{\ell_1}(\theta_{\text{pure-}d})$
**16 Output:** $\theta_{\text{out}}$
---

**Algorithm 8:** Sparse Vector Recovery $\mathcal{M}_{\mathrm{rec}}(b, \Phi, \xi)$

---

**2 Input:** Noisy measurement $b$, design matrix $\Phi$, noise tolerant magnitude $\xi$

**4** Define the feasible set $\mathcal{U} := \{\theta \in \mathbb{R}^d \mid \|\Phi\theta - b\|_1 \le \xi\}$

**6** Solve $\hat{\theta} = \arg\min_{\theta \in \mathcal{U}} \|\theta\|_1$

**8 Output:** $\hat{\theta}$

---

### I.3  Proof of Theorem 4

**Proof of Theorem 4** The privacy analysis follows by Theorem 1 and the post-possessing property of DP. The utility analysis follows by bounding the following $(a)$ and $(b)$:

$$\underbrace{\mathbb{E}[\mathcal{L}(\theta_{\mathrm{out}}; D) - \mathcal{L}(\theta_{FW}; D)]}_{(a)} + \underbrace{\mathbb{E}[\mathcal{L}(\theta_{FW}; D) - \mathcal{L}(\theta^*; D)]}_{(b)}$$

**For (a):** Since $\mathcal{C}$ is an $\ell_1$-ball with center $\mathbf{0}$ and $\ell_1$ radius $\|\mathcal{C}\|_1$, the vertices set is $S = \{x \mid \|x\|_1 = \|\mathcal{C}\|_1, \mathrm{nnz}(x) = 1\}$. We note that $\theta_{FW}$, the output of Algorithm 6 is $T$-sparse by Lemma 29.

Denote the "failure" events as follows: $F_1 := \{u \le \omega$ in Line 3 of Algorithm 1$\}$, where the uniform sample is accepted in Algorithm 1; $F_2 := \{\Phi$ is not $(e, 4T)$-RWC$\}$, where the randomly sampled $\Phi$ is not $(e, 4T)$-restricted well-conditioned (RWC); and $F_3 := \{\|\mathbf{z}\|_1 > \xi\}$, where the Laplace noise added in Line 4 of Algorithm 1 exceeds the tolerance threshold. We first bound the failure probability $\mathbb{P}(F_1 \cup F_2 \cup F_3)$, and then analyze the utility under the "success" event $F_1^c \cap F_2^c \cap F_3^c$, followed by an expected overall utility bound.

Since $\omega = \frac{1}{n}$, we have $\mathbb{P}(F_1) = \frac{1}{n}$.

Set the distortion rate as $e = \frac{1}{4}$, and $k = \Theta\left(T\log\left(\frac{d}{T}\right) + \log n\right)$. Constructing $\Phi$ following Lemma 44, and by Lemma 45, $\Phi \in \mathbb{R}^{k \times d}$ is $(4T, e)$-RWC with probability at least $1 - \frac{1}{n}$, i.e., $\mathbb{P}(F_2) \le \frac{1}{n}$.

Set

$$\xi = \frac{4\Delta}{\varepsilon}(k + \log n), \tag{14}$$

where $\Delta = 8\sqrt{k}\|\mathcal{C}\|_1 \left(\frac{\delta}{2\omega}\right)^{1/k} = \frac{8\|\mathcal{C}\|_1}{n\sqrt{k}}$. Then by Lemma 48 with taking $b = \frac{2\Delta}{\varepsilon}$ and $t = b(k + 2\log n)$, we have $\mathbb{P}(F_3) \le \frac{1}{n}$.

Under the "success" event $F_1^c \cap F_2^c \cap F_3^c$, consider the variables in Algorithm 7, we have

$$\theta_{\text{pure-}k} = \mathcal{A}_{\mathrm{pure}}\left(x_{\mathrm{apx}} = \mathsf{Clip}_{2\|\mathcal{C}\|_1}^{\ell_2}(\theta_{\text{apx-}k}), \Theta = \mathcal{B}_{\ell_2}^k(2\|\mathcal{C}\|_1), \varepsilon' = \varepsilon, \delta, \omega\right)$$

(Line 10 of Algorithm 7)

$$= \mathcal{A}_{\mathrm{pure}}\left(\theta_{\text{apx-}k}, \Theta = \mathcal{B}_{\ell_2}^k(2\|\mathcal{C}\|_1), \varepsilon' = \varepsilon, \delta, \omega\right) \qquad (\text{Under } F_2^c, \|\theta_{\text{apx-}k}\|_2 \le (1+e)\|\mathcal{C}\|_1)$$

$$= \theta_{\text{apx-}k} + \mathrm{Laplace}^{\otimes \mathrm{d}}\left(\frac{2\Delta}{\varepsilon}\right) \qquad (\text{Under } F_1^c)$$

$$= \Phi\theta_{\mathrm{FW}} + \tilde{w}, \text{ where } \|\tilde{w}\|_1 \le \xi. \qquad (\text{Under } F_3^c)$$

By Lemma 46 and $\theta_{\text{pure-}d} = \mathcal{M}_{\mathrm{rec}}(\theta_{\text{pure-}k}, \Phi, \xi)$, since $\theta_{\mathrm{FW}}$ is $T$-sparse, we have

$$\|\theta_{\text{pure-}d} - \theta_{\mathrm{FW}}\|_1 \le \frac{4\sqrt{T}}{\sqrt{1-e}} \cdot \xi.$$

Since $\|\theta_{\text{pure-}k} - \Phi\theta_{\mathrm{FW}}\|_1 = \|\tilde{w}\|_1 \le \xi$, i.e., $\theta_{\mathrm{FW}}$ is in the feasible set, we have $\|\theta_{\text{pure-}d}\|_1 \le \|\theta_{\mathrm{FW}}\|_1 \le \|\mathcal{C}\|_1$, therefore, by Line 14 in Algorithm 7, $\theta_{\mathrm{out}} = \mathsf{Clip}_{\|\mathcal{C}\|_1}^{\ell_1}(\theta_{\text{pure-}d}) = \theta_{\text{pure-}d}$ (under event $F_3^c$.)

Since $\mathcal{L}$ is $L_1$-Lipschitz with respect to $\ell_1$ norm, we get an upper bound on $(a)$:

$$\mathbb{E}[\mathcal{L}(\theta_{\text{out}}; D) - \mathcal{L}(\theta_{FW}; D) \mid F_1^c \cap F_2^c \cap F_3^c] \leq L_1 \|\theta_{\text{out}} - \theta_{FW}\|_1$$

$$\leq L_1 \frac{4\sqrt{T}}{\sqrt{1-e}} \cdot \xi \tag{15}$$

$$\leq \frac{64}{\sqrt{3}} \frac{L_1 \|\mathcal{C}\|_1 (k + \log n)}{n\varepsilon}$$

**For (b)**: by Lemma 28, we have:

$$\mathbb{E}\left[\mathcal{L}(\theta_{\text{FW}}; D) - \min_{\theta \in \mathcal{C}} \mathcal{L}(\theta; D) \mid F_1^c \cap F_2^c \cap F_3^c\right] = \mathcal{O}\left(\frac{\Gamma_{\mathcal{L}}}{T} + \frac{L_1 \|\mathcal{C}\|_1 \sqrt{8T \log(1/\delta)} \log(T L_1 \|\mathcal{C}\|_1 \cdot |S|)}{n\varepsilon}\right)$$

$$\leq \mathcal{O}\left(\frac{\beta \|\mathcal{C}\|_1^2}{T} + \frac{L_1 \|\mathcal{C}\|_1 \sqrt{8T \log(1/\delta)} \log(T L_1 \|\mathcal{C}\|_1 \cdot |S|)}{n\varepsilon}\right).$$

Therefore,

$$\mathbb{E}\left[\mathcal{L}(\theta_{\text{out}}; D) - \min_{\theta \in \mathcal{C}} \mathcal{L}(\theta; D) \mid F_1^c \cap F_2^c \cap F_3^c\right]$$

$$\leq \mathcal{O}\left(\frac{L_1 \|\mathcal{C}\|_1 (k + \log n)}{n\varepsilon} + \frac{\beta \|\mathcal{C}\|_1^2}{T} + \frac{L_1 \|\mathcal{C}\|_1 \sqrt{8T \log(1/\delta)} \log(T L_1 \|\mathcal{C}\|_1 \cdot |S|)}{n\varepsilon}\right)$$

$$= \tilde{\mathcal{O}}\left(\frac{L_1 \|\mathcal{C}\|_1 T}{n\varepsilon} + \frac{\beta \|\mathcal{C}\|_1^2}{T} + \frac{L_1 \|\mathcal{C}\|_1 T}{n\varepsilon}\right) \qquad (\delta = \frac{2\omega}{(nk)^k})$$

$$= \tilde{\mathcal{O}}\left(\frac{\beta \|\mathcal{C}\|_1^2}{T} + \frac{L_1 \|\mathcal{C}\|_1 T}{n\varepsilon}\right).$$

By setting $T = \tilde{\Theta}(\sqrt{\frac{n\varepsilon\beta\|\mathcal{C}\|_1}{L_1}})$, we have

$$\mathbb{E}\left[\mathcal{L}(\theta_{\text{out}}; D) - \min_{\theta \in \mathcal{C}} \mathcal{L}(\theta; D) \mid F_1^c \cap F_2^c \cap F_3^c\right] \leq \tilde{\mathcal{O}}\left(\frac{(\beta L_1 \|\mathcal{C}\|_1^3)^{1/2}}{(n\varepsilon)^{1/2}}\right). \tag{16}$$

By Line 14 in Algorithm 7, we have $\|\theta_{\text{out}}\|_1 \leq \|\mathcal{C}\|_1$. Therefore,

$$\mathbb{E}\left[\mathcal{L}(\theta_{\text{out}}; D) - \min_{\theta \in \mathcal{C}} \mathcal{L}(\theta; D) \mid F_1 \cup F_2 \cup F_3\right] \leq L_1 \|\theta_{\text{out}} - \theta^*\|_1 \leq 2L_1 \|\mathcal{C}\|_1.$$

$$\mathbb{E}\left[\mathcal{L}(\theta_{\text{out}}; D) - \min_{\theta \in \mathcal{C}} \mathcal{L}(\theta; D)\right] \leq \tilde{\mathcal{O}}\left(\frac{(\beta L_1 \|\mathcal{C}\|_1^3)^{1/2}}{(n\varepsilon)^{1/2}} + \frac{3}{n} 2L_1 \|\mathcal{C}\|_1\right)$$

$$\leq \tilde{\mathcal{O}}\left(\frac{(\beta L_1 \|\mathcal{C}\|_1^3)^{1/2}}{(n\varepsilon)^{1/2}}\right)$$

For the computation cost, in each iteration, the full-batch gradient is calculated; therefore, the cost for calculating $\theta_{\text{FW}}$ is $Tnd = \tilde{\mathcal{O}}(n^{3/2}d)$. The computation cost for Algorithm 1 is $\mathcal{O}(d)$. Therefore, the computation cost is $\tilde{\mathcal{O}}(n^{3/2}d)$, plus one call of the LASSO solver.

### I.4 Proof of Lemma 30

**Lemma 30** *Let $\mathcal{A}$ be any $\varepsilon$-DP ERM algorithm. For every parameter $n, d, \varepsilon$. There is a DP-ERM problem with a convex, $1$-Lipschitz, $1$-smooth loss function, a constrained parameter space $\Theta = \{\theta \in \mathbb{R}^d | \|\theta\|_1 \leq 1\}$ and a dataset $\text{Data} := \{x_1, ..., x_n\} \in \mathcal{X}^n$ that gives rise to the empirical risk $\mathcal{L}(\theta) = \frac{1}{n}\sum_{i=1}^{n} \ell(\theta; x_i)$, such that with probability at least $0.5$, the excess empirical risk*

$$\mathcal{L}(\mathcal{A}(\text{Data})) - \min_{\theta \in \Theta} \mathcal{L}(\theta) \geq \sqrt{\frac{\log(d+1)}{n\varepsilon + \log(4)}} \wedge 1.$$

**Definition 31 ($(\alpha, \beta)$-accurate ERM algorithm)** *Given parameter space $\Theta$, dataspace $\mathcal{X}$, and risk function $R$, we say an ERM algorithm $\mathcal{M}: \mathcal{X}^n \to \Theta$ is $(\alpha, \beta)$-accurate if for any dataset $D \in \mathcal{X}^n$, with probability at least $1 - \beta$ over the randomness of algorithm:*

$$R(\mathcal{M}(D); D) - \min_{\theta \in \Theta} R(\theta; D) \leq \alpha$$

**Lemma 32 (Restatement of Lemma 30)** *There exists a hard instance with $n$ samples over $\mathcal{B}_1^d$ and a 1-Lipschitz loss function $\mathcal{L}$ such that any $\varepsilon$-pure differential private $(\alpha, 1/2)$-accurate ERM algorithm $\mathcal{M}$ must have:*

$$\alpha \geq \sqrt{\frac{\log(d+1)}{n\varepsilon + \log(4)}} \wedge 1$$

**Proof of Lemma 30** We proof by the standard packing argument. Consider $\mathcal{B}_1^d$ and an $\alpha$-packing over it: $\{\theta_i\}_{i \in [K]}$, with $K$ being packing number $M(\alpha, \mathcal{B}_1^d, \|\cdot\|_2)$. For any $i \in [K]$, let $E_i = \{\theta \in \Theta \mid \|\theta - \theta_i\|_2 \leq \alpha\}$ and $X_i = \underbrace{\{\theta_i, \ldots, \theta_i\}}_{n \text{ copies}}$.

We define the error function is $\mathcal{L}(\theta; X_i) = \frac{1}{n} \sum_{j=1}^n \|\theta - X_i(j)\|_2$. Notice that:

$$
\begin{aligned}
1 \geq \mathbb{P}(\mathcal{M}(X_i) \notin E_i) &\geq \sum_{j \in [K] \setminus i} \mathbb{P}(\mathcal{M}(X_i) \in E_j) \\
&\geq \sum_{j \in [K] \setminus i} \exp(-n\varepsilon) \times \mathbb{P}(\mathcal{M}(X_j) \in E_j) \\
&\geq \frac{K \exp(-n\varepsilon)}{4}
\end{aligned}
\tag{17}
$$

Taking log of both sides, we have

$$n\varepsilon + \log(4) \geq \log(K) \tag{18}$$

It remains to calculate the packing number $K$. Notice that $M(\alpha, \mathcal{B}_1^d, \|\cdot\|_2) \asymp N(\alpha, \mathcal{B}_1^d, \|\cdot\|_2) \asymp \exp\left(\frac{1}{\alpha^2} \log(\alpha^2 d)\right)$[4], where we assume $\alpha \gtrsim \frac{1}{\sqrt{d}}$ [Wu16, Equation 15.4]. This implies:

$$\alpha \geq \tilde{\Omega}\left(\frac{1}{\sqrt{n\varepsilon + \log(4)}} \vee \frac{1}{\sqrt{d}}\right) \tag{19}$$

where $\tilde{\Omega}$ hides universal constant and logarithmic terms w.r.t. $d$. We conclude the proof by observing that $\mathcal{L}(\theta; X_i) \leq 1$, which implies that $\alpha \leq 1$. ∎

## J  Deferred Proofs for Data-dependent DP mechanism Design

### J.1  Pure DP Propose Test Release

**Definition 33 (Local Sensitivity)** *The local sensitivity of a query function $q$ on a dataset $X$ is defined as*

$$\Delta_{\text{Local}}^q(X) := \max_{X' \simeq X} \|q(X) - q(X')\|_2,$$

*where $X' \simeq X$ denotes that $X'$ is a neighboring dataset of $X$.*

We first present the original version of PTR in Algorithm 9. The pure DP version, obtained via the purification trick, is given in Algorithm 10. Their privacy guarantees are stated as follows:

**Lemma 34** *Algorithm 9 satisfies $(2\varepsilon, \delta)$-DP and its purified version, Algorithm 10 satisfies $(2\varepsilon + \varepsilon', 0)$-DP*

**Proof** The privacy guarantee for Algorithm 9 is based on [Vad17, Proposition 7.3.2]. For Algorithm 10, the privacy guarantee follows from the post-processing property of differential privacy and the privacy guarantee of Algorithm 1. ∎

---

[4] $N$ denotes the covering number.

---

**Algorithm 9:** $\mathcal{M}_{\mathrm{PTR}}(X, q, \varepsilon, \delta, \beta)$: Propose-Test-Release [DL09]

---

**2 Input:** Dataset $X$; privacy parameters $\varepsilon, \delta$; proposed bound $\beta$; query function $q : \mathcal{X} \to \Theta$

**4 Compute:** $\mathcal{D}_\beta^q(X) = \min\limits_{X'}\{d_{\mathrm{Hamming}}(X, X') \mid \Delta_{\mathrm{Local}}^q(X') > \beta\}$

**6 if** $\mathcal{D}_\beta^q(X) + \mathrm{Lap}\left(\frac{1}{\varepsilon}\right) \leq \frac{\log(1/\delta)}{\varepsilon}$ **then**

**8** $\quad$ Output $\perp$

**10 else**

**12** $\quad$ Release $f(X) + \mathrm{Lap}\left(\frac{\beta}{\varepsilon}\right)$

---

---

**Algorithm 10:** Pure DP Propose-Test-Release

---

**2 Input:** Dataset $X$; privacy parameters $\varepsilon, \varepsilon', \delta$; proposed bound $\beta$; query function $q : \mathcal{X} \to \Theta$, level of uniform smoothing $\omega$

**4** $\theta \leftarrow \mathcal{M}_{\mathrm{PTR}}(X, q, \varepsilon, \delta, \beta)$

**6 if** $\theta == \perp$ **then**

**8** $\quad u \sim \mathrm{Unif}(\Theta)$

**10** $\quad \theta \leftarrow u$

**12** $\theta_{\mathrm{out}} \leftarrow \mathcal{A}_{\mathrm{pure}}(\theta, \Theta, \varepsilon', \varepsilon, \delta, \omega)$ $\qquad\qquad\qquad\qquad\qquad$ ▷ Algorithm 1

**14 Output:** $\theta_{\mathrm{out}}$

---

## J.2 Privately Bounding Local Sensitivity

We assume query function $q : \mathcal{X}^* \to \Theta$ with $\Theta \subset \mathbb{R}^d$ being a convex set and $\mathrm{Diam}_2(\Theta) = R$. Assume the global sensitivity of local sensitivity is upper bounded by 1: $\max\limits_{X \simeq X'} \|\Delta_{\mathrm{Local}}^q(X) - \Delta_{\mathrm{Local}}^q(X')\|_2 \leq 1$. The purified version of PTR based on privately releasing local sensitivity is stated in Algorithm 11 and its utility guarantee is included in Theorem 11.

---

**Algorithm 11:**

---

**2 Input:** Dataset $D$; privacy parameters $\varepsilon, \varepsilon', \delta$; proposed bound $\beta$; query function
$\quad q : \mathcal{X}^* \to \Theta \subset \mathbb{R}^d$ with $\mathrm{Diam}_2(\Theta) = R$, level of uniform smoothing $\omega$

**4** $\hat{\beta} = \Delta_{\mathrm{Local}}^q(D) + \mathrm{Lap}\left(1/\varepsilon\right) + \log(2/\delta)/\varepsilon$

**6** $q_{\mathrm{apx}} \leftarrow \mathrm{Proj}_\Theta(q(D) + \mathrm{Lap}^{\otimes d}(\hat{\beta}/\varepsilon))$

**8** $q_{\mathrm{pure}} \leftarrow \mathcal{A}_{\mathrm{pure}}(\Theta, \varepsilon', \omega, R)$ $\qquad\qquad\qquad\qquad\qquad\qquad$ ▷ Algorithm 1

**10 Output:** $q_{\mathrm{pure}}$

---

**Theorem 11 (Restatement of theorem 5)** *Algorithm 11 satisfies* $(3\varepsilon, 0)$*-DP . Moreover,*

$$\mathbb{E}\left[\|q_{\mathrm{out}}(D) - q(D)\|_2\right] \leq \tilde{\mathcal{O}}\left(\frac{d^{1/2}\Delta_{\mathrm{Local}}^q(D)}{\varepsilon} + \frac{d^{3/2}}{\varepsilon^2}\right).$$

**Proof** First notice that $\hat{\beta}$ satisfies $\varepsilon$-DP by the privacy guarantee from Laplace mechanism and the assumption that global sensitivity of $\Delta_{\mathrm{Local}}^q(D)$ is upper bounded by 1. Second, we notice that $\mathbb{P}(\hat{\beta} > \Delta_{\mathrm{Local}}^q(D)) = \mathbb{P}(\mathrm{Lap}(1/\varepsilon) \geq \log(\delta/2)/\varepsilon) = 1 - \delta$. This implies $q(D) + \mathrm{Lap}^{\otimes d}(\hat{\beta}/\varepsilon)$ satisfies $(\varepsilon, \delta)$-probabilistic DP ([DRV10, Definition 2.2]), thus satisfies $(\varepsilon, \delta)$-DP. By post-processing and simple composition, $q_{apx}$ satisfies $(2\varepsilon, \delta)$-DP. Finally, using $\mathcal{A}_{\mathrm{pure}}$ under appropriate choice of $\delta$, we have $q_{pure}$ satisfies $(3\varepsilon, 0)$-DP by Theorem 1.

We now prove the utility guarantee. For notational convenience, we denote $z_0 \sim \mathrm{Lap}\left(1/\varepsilon\right)$, $Z_1 \sim \mathrm{Lap}^{\otimes d}(\hat{\beta}/\varepsilon)$ and $Z_2 \sim \mathrm{Lap}^{\otimes d}(\Delta/\varepsilon)$. By definition of $q_{\mathrm{pure}}$, we have:

$$\begin{aligned}
\mathbb{E}\left[\|q_{\mathrm{pure}} - q(D)\|_2\right] &= \mathbb{E}\left[\|q_{apx} - q(D) + Z_2\|_2\right] + \omega C \\
&= \mathbb{E}\left[\|\mathrm{Proj}_\Theta\left(q(D) + Z_1\right) - q(D) + Z_2\|_2\right] + \omega C \qquad (20) \\
&\leq \mathbb{E}\left[\|Z_1\|_2\right] + \mathbb{E}\left[\|Z_2\|_2\right] + \omega C
\end{aligned}$$

Notice that:

$$\mathbb{E}\left[\|Z_1\|_2\right] \leq \sqrt{\mathbb{E}\left[\|Z_1\|_2^2\right]}$$

$$= \sqrt{d\mathbb{E}[Z_{11}^2]} \qquad (Z_{11} \text{ denotes first element of } Z_1)$$

$$\leq \frac{\Delta_{\text{Local}}^q(D) + \log(2/\delta)/\varepsilon}{\varepsilon} + \frac{1}{\varepsilon}\mathbb{E}\left[|z_0|\right]$$

$$= \mathcal{O}\left(\frac{\sqrt{d}(\Delta_{\text{Local}}^q(D) + 1 + \log(2/\delta)/\varepsilon)}{\varepsilon}\right)$$

Now, set $\omega = \frac{1}{100} \wedge \frac{1}{C\varepsilon^2}$ and $\delta = \frac{2\omega}{(16dC\varepsilon)^d}$. By Corollary 4, we have :

$$\mathbb{E}[\|Z_2\|_2] + \omega C \leq \frac{2}{\varepsilon^2}.$$

Also notice that $\log(2/\delta) = \tilde{\mathcal{O}}(d)$. Thus,

$$\mathbb{E}\left[\|q_{\text{pure}} - q(D)\|_2\right] \leq \tilde{\mathcal{O}}\left(\frac{\sqrt{d}\Delta_{\text{Local}}^q(D)}{\varepsilon} + \frac{d^{3/2}}{\varepsilon^2}\right)$$

■

## J.3 Private Mode Release

The mode release algorithm discussed in Section 5.2 is provided in Algorithm 12.

---

**Algorithm 12:** Pure DP Mode Release

---

**2 Input:** Dataset $D$, pure DP parameter $\varepsilon$

**4 Set:** $\log(1/\delta) = d\log\left(2d^3/\varepsilon\right)$, where $d = \log_2|\mathcal{X}|$.

**6** Compute the mode $f(D)$ and its frequency $\text{occ}_1$, as well as the frequency $\text{occ}_2$ of the second most frequent item.

**8** Compute the gap: $\mathcal{D}_0^f(D) \leftarrow \left\lceil \frac{\text{occ}_1 - \text{occ}_2}{2} \right\rceil$.

**10 if** $\mathcal{D}_0^f(D) - 1 + \text{Lap}\left(\frac{1}{\varepsilon}\right) \leq \frac{\log(1/\delta)}{\varepsilon}$ **then**

**12** $\quad u_{\text{apx}} \leftarrow \bot$

**14 else**

**16** $\quad u_{\text{apx}} \leftarrow f(D)$

**18** $u_{\text{pure}} \leftarrow$ Algorithm 2 with inputs $\varepsilon, \delta, u_{\text{apx}}, \mathcal{Y} = \mathcal{X}$, and $\text{Index} = \text{id}$, the identity map.

**20 Output:** $u_{\text{pure}}$

---

**Proof of Theorem 6** By [Vad17, Proposition 3.3] and that $dist(D, \{D' : f(D') \neq f(D)\}) = \left\lceil \frac{\text{occ}_1 - \text{occ}_2}{2} \right\rceil$, we know $u_{\text{apx}}$ satisfies $(\varepsilon, \delta)$-DP. By [Vad17, Proposition 3.4], when $\text{occ}_1 - \text{occ}_2 \geq 4\lceil \ln(1/\delta)/\varepsilon \rceil$, $u_{\text{apx}}$ is the exact mode, i.e., $u_{\text{apx}} = f(D)$, with probability at least $1 - \delta$. Choosing $\delta < \frac{\varepsilon^d}{(2d)^{3d}}$, by Theorem 2 and the union bound, we have $\mathbb{P}\left(u_{\text{pure}} = f(D)\right) \geq 1 - \delta - 2^{-d} - \frac{d}{2}e^{-d} \geq 1 - \frac{3}{|\mathcal{X}|}$. ■

## J.4 Private Linear Regression Through Adaptive Sufficient Statistics Perturbation

We investigate the problem of differentially private linear regression. Specifically, we consider a fixed design matrix $X \in \mathbb{R}^{n \times d}$ and a response variable $Y \in \mathbb{R}^n$, which represent a collection of data points $(x_i, y_i)_{i=1}^n$, where $x_i \in \mathbb{R}^d$ and $y_i \in \mathbb{R}$. Assuming that there exists $\theta^* \in \Theta$ such that $Y = X\theta^*$, our goal is to find a differentially private estimator $\theta$ that minimizes the mean squared error:

$$\text{MSE}(\theta) = \frac{1}{2n}\|Y - X\theta\|_2^2$$

We assume prior knowledge of the magnitude of the dataspace: $\|\mathcal{X}\| := \sup_{x \in \mathcal{X}} \|x\|_2$ and $\|\mathcal{Y}\| := \sup_{y \in \mathcal{Y}} \|y\|_2$. Our algorithm operates under the same parameter settings as [Wan18, Algorithm 2]. To enable our purification procedure, we first derive a high-probability upper bound on $\|\tilde{\theta}\|_2$, where

---

**Algorithm 13:** $\mathcal{M}_{\mathrm{pure-AdaSSP}}$

---

**2 Input:** Data $X, y$; privacy parameters $\epsilon, \varepsilon', \delta$, Bounds: $\|\mathcal{X}\|, \|\mathcal{Y}\|$, level of smoothing $\omega$

**4** $\theta_{apx} \leftarrow \mathrm{AdaSSP}(X, y, \varepsilon, \delta, \|\mathcal{X}\|, \|\mathcal{Y}\|)$ ▷ [Wan18, Algorithm 2]

**6** Propose a high probability upper bound $\tilde{R} = \tilde{\mathcal{O}}\left((1+n\varepsilon/d)\|\mathcal{Y}\|/\|\mathcal{X}\|\right)$

**8** Construct trust region $\Theta := \mathcal{B}_{\ell_2}^d(\tilde{R})$

**10** Norm clipping $\theta_{apx} \leftarrow \mathrm{Proj}_\Theta(\theta_{apx})$

**12** $\theta_{\mathrm{pure}} \leftarrow \mathcal{A}_{\mathrm{pure}}(\theta_{apx}, \Theta, \varepsilon, \varepsilon', \delta)$

**14 Output:** $\theta_{\mathrm{pure}}$

---

$\tilde{\theta}$ is the output of AdaSSP. Subsequently, we clip the output of AdaSSP and apply the purification procedure. The implementation details are provided in Algorithm 13.

**Theorem 12 (Restatement of theorem 7)** *Assume $X^\top X$ is positive definite and $\|\mathcal{Y}\| \lesssim \|\mathcal{X}\|\|\theta^*\|$. With probability $1 - \zeta - 1/n^2$, the output $\theta_{pure}$ from Algorithm 13 satisfies:*

$$\mathrm{MSE}(\theta_{pure}) - \mathrm{MSE}(\theta^*) \leq \tilde{\mathcal{O}}\left(\frac{\|\mathcal{X}\|^2}{\varepsilon^2 n^4} + \frac{d\|\mathcal{X}\|^2\|\theta^*\|^2}{n\epsilon} \wedge \frac{d^2\|\mathcal{X}\|^4\|\theta^*\|^2}{\epsilon^2 n^2 \lambda_{\min}}\right) \tag{21}$$

*where $\lambda_{\min} := \lambda_{\min}(X^\top X/n)$.*

**Proof** First, we introduce a utility Lemma from [Wan18]

**Lemma 35 (Theorem 3 from [Wan18])** *Under the setting of [Wan18, Algorithm 2], AdaSSP satisfies $(\varepsilon, \delta)$-DP. Assume $\|\mathcal{Y}\| \lesssim \|\mathcal{X}\|\|\theta^*\|$, then with probability $1 - \zeta$,*

$$MSE(\tilde{\theta}) - MSE(\theta^*) \leq \mathcal{O}\left(\frac{\sqrt{d\log\left(\frac{d^2}{\zeta}\right)}\|\mathcal{X}\|^2\|\theta^*\|^2}{n\epsilon/\sqrt{\log\left(\frac{6}{\delta}\right)}} \wedge \frac{\|\mathcal{X}\|^4\|\theta^*\|^2 \mathrm{tr}[(X^T X)^{-1}]}{n\epsilon^2/[\log\left(\frac{6}{\delta}\right)\log\left(\frac{d^2}{\zeta}\right)]}\right) \tag{22}$$

Now, we prove the utility guarantee for our results. In order to operate the purification technique, we need to estimate the range $\|\tilde{\theta}\|_2$ in order to apply the uniform smoothing technique. Notice that

$$\|\tilde{\theta}\|_2 \leq \underbrace{\|(X^\top X + \lambda I_d + E_1)^{-1}\|_2}_{(a)} \underbrace{\|X^\top y + E_2\|_2}_{(b)} \tag{23}$$

with $E_2 \sim \frac{\sqrt{\log(6/\delta)}\|\mathcal{X}\|\|\mathcal{Y}\|}{\varepsilon/3}\mathcal{N}(0, I_d)$ and $E_1 \sim \frac{\sqrt{\log(6/\delta)}\|\mathcal{X}\|^2}{\varepsilon/3}Z$, where $Z \in \mathbb{R}^{d \times d}$ is a symmetric matrix, and each entry in its upper-triangular part (including the diagonal) is independently sampled from $\mathcal{N}(0, 1)$. *Under the high probability event in Lemma 35, we upper bound $(a)$ and $(b)$ separately:*

**For (a):**

$$\|(X^\top X + \lambda I_d + E_1)^{-1}\|_2 = \frac{1}{\lambda_{\min}(X^\top X + \lambda I_d + E_1)}$$

By the choice of $\lambda$ and concentration of $\|E_1\|_2$, we have $(X^\top X + \lambda I_d + E_1) \succ \frac{1}{2}(X^\top X + \lambda I_d)$, which allows a lower bound on $\lambda_{\min}(X^\top X + \lambda I_d + E_1)$:

$$\begin{aligned}
2\lambda_{\min}(X^\top X + \lambda I_d + E_1) &\geq \lambda_{\min}(X^\top X + \lambda I_d) \\
&\geq \lambda_{\min}(X^\top X) + \lambda \\
&\geq \lambda_{\min}(X^\top X) + \frac{\sqrt{d\log(6/\delta)\log(2d^2/\zeta)}\|\mathcal{X}\|^2}{\varepsilon/3} - \lambda_{\min}^* \\
&\geq \frac{\sqrt{d\log(6/\delta)\log(2d^2/\zeta)}\|\mathcal{X}\|^2}{\varepsilon/3}
\end{aligned}$$

where the third inequality is by setting $\lambda = \max\left\{0, \frac{\sqrt{d\log(6/\delta)\log(2d^2/\zeta)}\|\mathcal{X}\|^2}{\varepsilon/3} - \lambda_{\min}^*\right\}$, where $\lambda_{\min}^*$ is a high probability lower bound on $\lambda_{\min}(X^\top X)$. Thus,

$$
\begin{aligned}
\|X^\top X + \lambda I_d + E_1\|_2 &= \frac{1}{\lambda_{\min}(X^\top X + \lambda I_d + E_1)} \\
&\leq \frac{2}{\lambda_{\min}(X^\top X + \lambda I_d)} \\
&\leq \frac{2\varepsilon}{3\sqrt{d\log(6/\delta)\log(2d^2/\zeta)}\|\mathcal{X}\|^2} \\
&= \tilde{\mathcal{O}}\left(\frac{\varepsilon}{\sqrt{d\log(6/\delta)}\|\mathcal{X}\|^2}\right)
\end{aligned}
$$

For (b), by triangle inequality

$$(b) = \|X^\top y + E_2\|_2 \leq \|X^\top y\|_2 + \|E_2\|_2 \leq n\|\mathcal{X}\|\|\mathcal{Y}\| + \|E_2\|_2$$

Apply [Wan18, Lemma 6], we have w.p. at least $1 - \beta$:

$$\|E_2\|_2 = \mathcal{O}\left(\frac{\sqrt{d}\|\mathcal{X}\|\|\mathcal{Y}\|\sqrt{\log(6/\delta)\log(d/\beta)}}{\varepsilon}\right)$$

Thus, w.p. at least $1 - \beta$ over the randomness of $E_2$,

$$
\begin{aligned}
(b) &\leq \mathcal{O}\left(n\|\mathcal{X}\|\|\mathcal{Y}\| + \frac{\sqrt{d}\|\mathcal{X}\|\|\mathcal{Y}\|\sqrt{\log(6/\delta)\log(d/\beta)}}{\varepsilon}\right) \\
&= \tilde{\mathcal{O}}\left(n\|\mathcal{X}\|\|\mathcal{Y}\| + \frac{\sqrt{d}\|\mathcal{X}\|\|\mathcal{Y}\|\sqrt{\log(6/\delta)}}{\varepsilon}\right)
\end{aligned}
$$

Putting everything together under the high probability event:

$$
\begin{aligned}
\|\tilde{\theta}\|_2 &\leq \tilde{\mathcal{O}}\left(\frac{\|\mathcal{Y}\|}{\|\mathcal{X}\|}\left(1 + \frac{n\varepsilon}{\sqrt{d\log(6/\delta)}}\right)\right) \\
&\leq \tilde{\mathcal{O}}\left(\frac{\|\mathcal{Y}\|}{\|\mathcal{X}\|}\left(1 + \frac{n\varepsilon}{d}\right)\right) := \tilde{r}
\end{aligned}
$$

Now, we apply purification Algorithm 1 with $\Theta = \mathcal{B}_{\ell_2}^d(\tilde{r})$, $\omega = \frac{1}{n^2}$, $\delta = \frac{2\omega}{(16d^{3/2}\tilde{r}n^2)^d}$. This parameter configuration implies $\log(1/\delta) = \tilde{\mathcal{O}}(d)$ and Wasserstein-$\infty$ distance $\Delta = \frac{1}{4n^2d}$ (Line 2 of Algorithm 1)

Finally, it remains to bound the estimation error for $\theta_{pure}$. Let's denote the additive Laplace noise from purification by $Z_2 \sim \mathrm{Lap}^{\otimes d}(\Delta/\varepsilon)$, and the purified estimator $\theta_{pure} := \tilde{\theta} + Z_2$. Under the event that the purification algorithm doesn't output uniform noise, which happens w.p. at least $1 - \omega$:

$$
\begin{aligned}
\mathrm{MSE}(\theta_{pure}) - \mathrm{MSE}(\tilde{\theta}) &= \frac{1}{2n}\|y - X\theta_{pure}\|_2^2 - \frac{1}{2n}\|y - X\tilde{\theta}\|_2^2 \\
&= \frac{1}{n}Z_2^\top X^\top X Z_2 \\
&\leq \frac{1}{n}\lambda_{\max}(X^\top X)\|Z_2\|_2^2 \\
&\leq \|\mathcal{X}\|^2\|Z_2\|_1^2 \\
&\leq \frac{\|\mathcal{X}\|^2}{\varepsilon^2 n^4}
\end{aligned}
$$

where the last inequality holds w.p. $1 - 1/n^2$ by Lemma 48, as we derived below:

$$\|Z_2\|_1 \leq \frac{2d\Delta}{\varepsilon} + \frac{2\Delta}{\varepsilon} \log(n^2)$$
$$\leq \tilde{\mathcal{O}}\left(\frac{1}{\varepsilon n^2}\right)$$

Under the high probability event of Lemma 35 and Algorithm 1, which happens with probability at least $1 - \zeta - 1/n^2$:

$$\text{MSE}(\theta_{pure}) - \text{MSE}(\theta^*) = \text{MSE}(\theta_{pure}) - \text{MSE}(\tilde{\theta}) + \text{MSE}(\tilde{\theta}) - \text{MSE}(\theta^*)$$
$$\leq \tilde{\mathcal{O}}\left(\frac{\|\mathcal{X}\|^2}{\varepsilon^2 n^4} + \frac{d\|\mathcal{X}\|^2\|\theta^*\|^2}{n\epsilon} \wedge \frac{d\|\mathcal{X}\|^4\|\theta^*\|^2 \operatorname{tr}[(X^T X)^{-1}]}{n\epsilon^2}\right) \quad (24)$$
$$\leq \tilde{\mathcal{O}}\left(\frac{\|\mathcal{X}\|^2}{\varepsilon^2 n^4} + \frac{d\|\mathcal{X}\|^2\|\theta^*\|^2}{n\epsilon} \wedge \frac{d^2\|\mathcal{X}\|^4\|\theta^*\|^2}{n\epsilon^2 \lambda_{\min}(X^\top X)}\right)$$

By denoting $\lambda_{\min} = \lambda_{\min}\left(\frac{X^\top X}{n}\right)$, we have:

$$\text{MSE}(\theta_{pure}) - \text{MSE}(\theta^*) \leq \tilde{\mathcal{O}}\left(\frac{\|\mathcal{X}\|^2}{\varepsilon^2 n^4} + \frac{d\|\mathcal{X}\|^2\|\theta^*\|^2}{n\epsilon} \wedge \frac{d^2\|\mathcal{X}\|^4\|\theta^*\|^2}{\epsilon^2 n^2 \lambda_{\min}}\right) \quad (25)$$

■

# K  Deferred Proofs for Private Query Release

## K.1  Problem Setting

Let data universe $\mathcal{X} = \{0,1\}^d$ and denote $N := |\mathcal{X}|$. The dataset, $D \in \mathcal{X}^n$ is represented as a histogram $D \in \mathbb{N}^{|\mathcal{X}|}$ with $\|D\|_1 = n$. We consider bounded linear query function $q : \mathcal{X} \to [0,1]$ and workload $Q$ with size $K$. For a shorthand, we denote:

$$Q(D) := (q_1(D), \dots, q_k(D))^\top := \left(\frac{1}{n}\sum_{i\in[n]} q_1(d_i), \dots, \frac{1}{n}\sum_{i\in[n]} q_k(d_i)\right)^\top$$

## K.2  Private Multiplicative Weight Exponential Mechanism

We first introduce private multiplicative weight exponential algorithm (MWEM):

---

**Algorithm 14:** Multipliative Weight Exponential Mechanism $\texttt{MWEM}(D, Q, T, \rho)$ [HLM12]

---

2 **Input:** Data set $D \in \mathbb{N}^{|\mathcal{X}|}$, set $Q$ of linear queries; Number of iterations $T \in \mathbb{N}$; zCDP Privacy parameter $\rho > 0$.

4 **Set:** number of data points $n \leftarrow \|D\|_1$, initial distribution $p_0 \leftarrow \frac{1_{|\mathcal{X}|}}{|\mathcal{X}|}$, privacy budget for each mechanism $\varepsilon_0 \leftarrow \sqrt{\rho/T}$

6 **Define:** $\text{Score}(\cdot; \hat{p}, p) = |\langle \cdot, \hat{p}\rangle - \langle \cdot, p\rangle|$

8 **for** $t = 1$ *to* $T$ **do**

10     $q_t \leftarrow \texttt{ExpoMech}(Q, \varepsilon_0, \text{Score}(\cdot; p_{t-1}, p))$

12     $m_t \leftarrow \langle q_t, X\rangle + \text{Laplace}(1/n\varepsilon_0)$

14     *Multiplicative weights update:* let $p_{t+1}$ be the distribution over $\mathcal{X}$ with entries satisfy:

$$q_t(x) \propto q_{t-1}(x) \cdot \exp(q_t(x) \cdot (m_t - q_t(p_{t-1}))/2), \ \forall x \in \mathcal{X}$$

16 **Output:** $D_{\text{out}} \leftarrow \frac{n}{T}\sum_{t=0}^{T-1} p_t$.

---

---

**Algorithm 15:** `ProportionalSampling`$(\mathcal{X}, p, m)$

---

**2 Input:** Dataspace $\mathcal{X}$, Probability vector $p \in \mathbb{R}^{|\mathcal{X}|}$, sample size $m$
**4 for** $i = 1$ *to* $m$ **do**
**6** $\quad \lfloor \; s_i \leftarrow$ `UnSortedProportionalSampling`$(p, \mathcal{X})$ $\qquad \triangleright$ e.g., Alias method [Wal74]
**8 Output:** $\{s_1, ... s_m\}$

---

---

**Algorithm 16:** Pure DP Multiplicative Weight Exponential Mechanism

---

**2 Input:** Dataset $D \in \mathbb{N}^{|\mathcal{X}|}$ with size $\|D\|_1 = n$, Query set $Q$, privacy parameters $\epsilon, \delta$, accuracy parameter $\alpha$
**4 Set:** Number of iterations $T = \tilde{\mathcal{O}}(\varepsilon^{2/3} n^{2/3} d^{1/3})$, size of subsampled dataset
$\quad m = n^{2/3} \varepsilon^{2/3} d^{-2/3}$, zCDP budget $\rho = \varepsilon^2/16 \log(1/\delta)$,
$\quad \text{Score}(q; \hat{p}, p) = |\langle q, \hat{p} \rangle - \langle q, p \rangle|, \; \forall q \in Q$
**6 Initialize:** $p_1 \leftarrow \frac{\mathbf{1}_{|\mathcal{X}|}}{|\mathcal{X}|}, \; p \leftarrow \frac{D}{\|D\|_1}$
**8** $\hat{D} \leftarrow$ `MWEM`$(D, Q, T, \rho)$ $\qquad\qquad\qquad\qquad\qquad \triangleright$ Algorithm 14
**10** $Y \leftarrow$ `ProportionalSampling`$(\mathcal{X}, \hat{D}/n, m)$ $\qquad\qquad \triangleright$ Algorithm 16
**12** $\hat{Y} \leftarrow \mathcal{A}_{\text{pure-discrete}}(\varepsilon, \delta, Y, \mathcal{X}^m)$
**14 Output:** $\hat{Y}$

---

**Lemma 36 (Privacy and Utility of MWEM [HLM12])** *Algorithm 14 instantiated as*

`MWEM`$(D, Q, T, \varepsilon^2/16 \log(1/\delta))$ *satisfies* $(\varepsilon, \delta)$-*DP. With probability* $1 - 2\beta T$, *PMW and the output* $\hat{D}$ *such that:*

$$\|Q(\hat{D}) - Q(D)\|_\infty \leq \mathcal{O}\left( \sqrt{\frac{d}{T}} + \frac{\sqrt{T \log(1/\delta)} \log(K/\beta)}{n\varepsilon} \right) \qquad (26)$$

**Proof** We first state the privacy guarantee. Since each iteration satisfies zCDP guarantee $\rho/T$-zCDP, the total zCDP guarantee for $T$ iterations is $\rho$. By Lemma 22, the whole algorithm satisfies $(4\sqrt{\rho \log(1/\delta)}, \delta)$-DP. Plugging in the choice of $\rho = \varepsilon^2/16 \log(1/\delta)$, we have `MWEM`$(D, Q, T, \varepsilon^2/16 \log(1/\delta))$ satisfies $(\varepsilon, \delta)$-DP. The utility guarantee follows [HLM12, Theorem 2.2]. Specifically, we choose $\text{adderr} = 2\sqrt{T/\rho} \log(K/\beta)$, this yields the utility guarantee stated in Theorem with probability $1 - 2\beta T$. $\blacksquare$

**Lemma 37 (Sampling bound, Lemma 4.3 in [DR$^+$14])** *Let data* $X = (a_1, ..., a_N)$ *with* $\sum_{i=1}^N a_i = 1$ *and* $a_i \geq 0$. $Y \sim \text{Multinomial}(m, X)$. *Then we have:*

$$\mathbb{P}[\|Q(Y) - Q(X)\|_\infty \geq \alpha] \leq 2|Q| \exp(-2m\alpha^2)$$

**Proof** The proof follows the proof of [DR$^+$14, Lemma 4.3]. Since we have $Y = (Y_1, ..., Y_m)$ with $Y_i \overset{\text{iid}}{\sim} \text{Multinomial}(1, X)$, for any $q \in Q$, we have $q(Y) = \frac{1}{m} \sum_{i=1}^m q(Y_i)$ and $\mathbb{E}[q(Y)] = q(X)$. By the Chernoff bound and a union bound over all queries in $Q$, we have $\mathbb{P}[\|Q(Y) - Q(X)\|_\infty \geq \alpha] \leq 2|Q| \exp(-2m\alpha^2)$. $\blacksquare$

**Theorem 13 (Restatement of Theorem 8)** *Algorithm 16 satisfies* $2\varepsilon$-*DP. Moreover, the output* $\hat{Y}$ *satisfies*

$$\|Q(D) - Q(\hat{Y})\|_\infty \leq \tilde{\mathcal{O}}\left( \frac{d^{1/3}}{n^{1/3} \varepsilon^{1/3}} \right),$$

*and the runtime is* $\tilde{\mathcal{O}}(nK + \varepsilon^{2/3} n^{2/3} d^{1/3} NK + N)$.

**Proof** We first state the privacy guarantee. By Lemma 36, the output from multiplicative weight exponential mechanism, $\hat{D}$, satisfies $(\varepsilon, \delta)$-DP. By post-processing, $Y$ is also $(\varepsilon, \delta)$-DP. Thus, apply Theorem 2, the purified $\hat{Y}$ satisfies $2\varepsilon$-DP.

The utility guarantee is via bounding the following terms:

$$\|Q(D) - Q(\hat{Y})\|_\infty \leq \underbrace{\|Q(D) - Q(\hat{D})\|_\infty}_{(a)} + \underbrace{\|Q(\hat{D}) - Q(Y)\|_\infty}_{(b)} + \underbrace{\|Q(Y) - Q(\hat{Y})\|_\infty}_{(c)}$$

For $(c)$: Since the output space $\mathcal{Y} = \mathcal{X}^m$, if use binary encoding, the length of code is $\log_2(|\mathcal{Y}|) = md := \tilde{d}$. Thus, by Theorem 2, we choose $\delta = \frac{\varepsilon^{\tilde{d}}}{(2\tilde{d})^{3\tilde{d}}}$, this implies $\log(1/\delta) = \mathcal{O}(md\log(2md/\varepsilon))$ and failure probability $\beta_0 = \frac{1}{2^{md}} + \frac{md}{\exp(md)}$.

For $(a)$: In order to minimize upper bound in Equation 26, we choose $T = \frac{d^{1/2}n\varepsilon}{\log^{1/2}(1/\delta)\log(K/\beta)}$, this implies $\|Q(D) - Q(\hat{D})\|_\infty \leq \frac{d^{1/4}\log^{1/4}(1/\delta)\log^{1/2}(K/\beta)}{(n\varepsilon)^{1/2}} = \mathcal{O}\left(\frac{d^{1/2}m^{1/4}\log^{1/4}(2md/\varepsilon)\log^{1/2}(K/\beta)}{(n\varepsilon)^{1/2}}\right)$

For $(b)$: using Sampling bound (Lemma 37) and setting failure probability $\beta_1 = 2K\exp(-2m\alpha^2)$, we have $\|Q(\hat{D}) - Q(Y)\|_\infty \leq \mathcal{O}\left(\frac{\log^{1/2}(2K/\beta_1)}{m^{1/2}}\right)$

Finally, we choose $m = (n\varepsilon/d)^{2/3}$ to balance the error between $(a)$ and $(b)$. This implies:

$$\|Q(\hat{D}) - Q(Y)\|_\infty \leq \mathcal{O}\left(\frac{d^{1/3}}{(n\varepsilon)^{1/3}}\left(\log^{1/2}(2K/\beta_1) + \log^{1/2}(K/\beta)\log^{1/4}(2d^{1/3}n^{2/3}\varepsilon^{-1/3})\right)\right)$$

Set $\beta = \frac{1}{2Tn}$ and $\beta_1 = \frac{1}{n}$, and by $\beta_0 \leq \frac{2(n\varepsilon)^{2/3}d^{1/3}}{2^{(n\varepsilon)^{2/3}d^{1/3}}} = o(\frac{1}{n})$, we have

$$\mathbb{E}[\|Q(\hat{D}) - Q(Y)\|_\infty] \leq \mathcal{O}\left(\frac{d^{1/3}}{(n\varepsilon)^{1/3}}\left(\log^{1/2}(2nK) + \log^{1/2}(Kd^{1/3}n^{5/3}\varepsilon^{1/3})\log^{1/4}(2d^{1/3}n^{2/3}\varepsilon^{-1/3})\right)\right)$$
$$+ (T\beta + \beta_1 + \beta_0)$$
$$= \tilde{\mathcal{O}}\left(\frac{d^{1/3}}{(n\varepsilon)^{1/3}} + \frac{1}{n}\right)$$

The computational guarantee follows: (1) Since $T = \frac{d^{1/2}n\varepsilon}{\log^{1/2}(1/\delta)\log(K/\beta)} = \tilde{\mathcal{O}}\left(\varepsilon^{2/3}n^{2/3}d^{1/3}\right)$. The runtime for MWEM is $\tilde{\mathcal{O}}(nK + \varepsilon^{2/3}n^{2/3}d^{1/3}NK)$ [HLM12]; (2) For subsampling, by runtim analysis of Alias method [Wal74], the preprocessing time is $\mathcal{O}(N)$ and the query time is $\mathcal{O}(m)$ for generating $m$ samples. Thus, total runtime is $\mathcal{O}(N + (n\varepsilon)^{2/3}d^{-2/3})$; (3) Finally, note that the query time for Algorithm 2 is $\mathcal{O}(\tilde{d}) = \mathcal{O}(d^{1/3}(n\varepsilon)^{2/3})$. We conclude that the runtime for Algorithm 16 is $\tilde{\mathcal{O}}(nK + \varepsilon^{2/3}n^{2/3}d^{1/3}NK + N)$. ∎

## L  Deferred Proofs for Lower Bounds

### L.1  Proof of Theorem 9

**Lemma 38 (Lemma 5.1 in [BST14])** *Let* $n, d \in \mathbb{N}$ *and* $\epsilon > 0$. *There is a number* $M = \Omega(\min(n, d/\epsilon))$ *such that for every* $\epsilon$-*differentially private algorithm* $\mathcal{A}$, *there is a dataset* $D = \{x_1, \ldots, x_n\} \subset \{-1/\sqrt{d}, 1/\sqrt{d}\}^d$ *with* $\|\sum_{i=1}^n x_i\|_2 \in [M-1, M+1]$ *such that, with probability at least* $1/2$ *(taken over the algorithm random coins), we have*

$$\|\mathcal{A}(D) - \bar{D}\|_2 = \Omega\left(\min\left(1, \frac{d}{\epsilon n}\right)\right)$$

*where* $\bar{D} = \frac{1}{n}\sum_{i=1}^n x_i$.

**Theorem 14 (Restatement of Theorem 9)** *Denote* $\mathcal{D} := \{-1/\sqrt{d}, 1/\sqrt{d}\}^d$. *Let* $\varepsilon \leq \mathcal{O}(1)$, *and* $\delta \in \left(\frac{1}{\exp(4d\log(d)\log^2(nd))}, \frac{1}{4n^d\log^{2d}(8d)}\right)$. *For any* $(\varepsilon, \delta)$-*DP mechanism* $\mathcal{M}$, *there exist a dataset* $D \in \mathcal{D}^n$ *such that w.p. at least* $1/4$ *over the randomness of* $\mathcal{M}$:

$$\|\mathcal{M}(D) - \bar{D}\|_2 \geq \tilde{\Omega}\left(\frac{\sqrt{d\log(1/\delta)}}{\varepsilon n}\right)$$

*Here,* $\tilde{\Omega}(\cdot)$ *hides all polylogarithmic factors, except those with respect to* $\delta$.

**Proof** Suppose there exists an $(\varepsilon, \delta)$-differentially private mechanism $\mathcal{M}$ such that with probability at least $3/4$ over the randomness of $\mathcal{M}$, for any $D \in \mathcal{D}$,

$$\|\mathcal{M}(D) - \bar{D}\|_2 \leq \frac{\sqrt{d \log(1/\delta)}}{n\varepsilon a}$$

where $a$ is a term involving $n$ and $d$, to be specified later. Let $\frac{\sqrt{d \log(1/\delta)}}{n\varepsilon a} \leq \frac{d}{n\varepsilon \log^{1/2} d}$ implies:

$$\delta > \exp(-a^2 d / \log(d)) \tag{27}$$

We execute Algorithm 1 to purify $\mathcal{M}$ directly over the output space $[-1/\sqrt{a}, 1/\sqrt{a}]^d$. Let $Y$ denote the output of Algorithm 1 and $U \sim \mathrm{Unif}([-1/\sqrt{a}, 1/\sqrt{a}]^d)$. The remainder of the proof involves bounding the additional errors introduced during the purification process. By triangle inequality we have

$$\|\bar{D} - Y\|_2 \leq \underbrace{\|\bar{D} - \mathcal{M}(D)\|_2}_{(a)} + \underbrace{\|\mathcal{M}(D) - Y\|_2}_{(b)}.$$

Notice that under the event that Line 3 of Algorithm 1 doesn't return the uniform random variable, which happens with probability $1 - \omega$, we have $Y = \mathcal{M}(X) + \mathrm{Laplace}^{\otimes d}(2\Delta/\varepsilon)$, so term (b) equals the 2-norm of the Laplace perturbation.

For the remaining proofs, we choose the mixing level $\omega = 1/8$ in Algorithm 1. We now justify the choice of $\delta$:

Observe that since $Y = \mathcal{M}(X) + \mathrm{Laplace}^{\otimes d}(2\Delta/\varepsilon)$, term (b), which accounts for the error introduced by Laplace noise. With probability at least $7/8$ by the concentration of the $L_2$ norm of Laplace vector:

$$(b) \leq \frac{2\sqrt{d}\Delta \log(8d)}{\varepsilon}$$

Thus, without loss of generality, we require $\frac{2\sqrt{d}\Delta \log(8d)}{\varepsilon} \leq \frac{16d}{n\varepsilon \log(8d)}$, this implies

$$\Delta \leq \frac{8\sqrt{d}}{n \log^2(8d)}$$

Notice that:

$$\Delta = d^{1-\frac{1}{q}} \cdot \frac{2R^2}{r} \left(\frac{\delta}{2\omega}\right)^{1/d}$$

Choosing $q = \infty$ (corresponding to the use of $\ell_\infty$ norm in $W$-$\infty$ distance), and noticing $R = 2/\sqrt{d}$ and $r = 1/\sqrt{d}$, we obtain the condition:

$$\Delta = 8\sqrt{d}\left(\frac{\delta}{2\omega}\right)^{1/d} \leq \frac{8\sqrt{d}}{n \log^2(8d)}$$

which further implies:

$$\delta \leq \frac{1}{4n^d \log^{2d}(8d)} \tag{28}$$

By Eq. (32) and Eq. (33), we have:

$$\delta \in \left(\exp(-a^2 d / \log(d)), \quad \frac{1}{4n^d \log^{2d}(8d)}\right) \tag{29}$$

The constrained above yields a lower bound on $a^2$, after some relaxation for simplicity (and assume $d \geq \log(8d)$ and $d \log(n) > \log(4)$):

$$a^2 \geq 2 \log(d) \log(nd) \tag{30}$$

Thus, we set $a = 2\log(d)\log(nd)$ to satisfy the constraint stated in Eq. (35), and now

$$\delta \in \left( \frac{1}{\exp(4d\log(d)\log^2(nd))}, \quad \frac{1}{4n^d\log^{2d}(8d)} \right) \tag{31}$$

When $\delta$ is within the range above, we have:

$$\log(1/\delta) < 4d\log(d)\log^2(nd)$$

This implies that, w.p. at least $1/2$ over the randomness of $\mathcal{M}$ and purification algorithm:

$$\|\bar{D} - Y\|_2 \leq \frac{d}{n\varepsilon\log^{1/2}(d)} + \frac{16d}{n\varepsilon\log(8d)},$$

which violates the lower bound stated in Lemma 38. Thus, for any $(\varepsilon, \delta)$-DP mechanism $\mathcal{M}$ with $\delta$ being in the range of Eq. (36), there exists a dataset $D \in \mathcal{D}$, such that with probability greater than $1/4$ over the randomness of $\mathcal{M}$:

$$\|\mathcal{M}(D) - \bar{D}\|_2 \geq \left( \frac{\sqrt{d\log(1/\delta)}}{2n\varepsilon\log(d)\log(nd)} \right) = \tilde{\Omega}\left( \frac{\sqrt{d\log(1/\delta)}}{n\varepsilon} \right)$$

Here, $\tilde{\Omega}(\cdot)$ hides all polylogarithmic factors, except those with respect to $\delta$. ∎

## L.2 More Examples of Lower Bounds via the Purification Trick

In this section, we present an extended result for Theorem 9. Additionally, we establish a lower bound for the discrete setting, as stated in Theorem 16, thereby demonstrating that the purifying recipe for proving lower bound remains applicable in the discrete case.

### L.2.1 One-Way Marginal Release

We establish a stronger version of Theorem 9, as stated in Theorem 15, which strengthens Theorem 9 by establishing that the lower bound holds for any $c \in (0, 1)$, rather than being restricted to $c = 1/2$.

**Theorem 15 (Restatement of Theorem 9)** *Denote $\mathcal{D} := \{-1/\sqrt{d}, 1/\sqrt{d}\}^d$. Let $\varepsilon \leq \mathcal{O}(1)$, for any $c \in (0, 1)$, and $\delta \in \left( \frac{1}{n^{2d}d^{2d}}, \frac{1}{4n^d\log^{2d}(8d)} \right)$. For any $(\varepsilon, \delta)$-DP mechanism $\mathcal{M}$, there exist a dataset $D \in \mathcal{D}^n$ such that with probability at least $1/4$ over the randomness of $\mathcal{M}$:*

$$\|\mathcal{M}(D) - \bar{D}\|_2 \geq \tilde{\Omega}\left( \max_{c \in (0,1)} \frac{d^c\log^{1-c}(1/\delta)}{\varepsilon n} \right).$$

*Here, $\tilde{\Omega}(\cdot)$ hides all polylogarithmic factors, except those with respect to $\delta$.*

**Proof** Suppose for some $c \in (0, 1)$, there exists an $(\varepsilon, \delta)$-differentially private mechanism $\mathcal{M}$ such that with probability at least $3/4$ over the randomness of $\mathcal{M}$, for any $D \in \mathcal{D}$,

$$\|\mathcal{M}(D) - \bar{D}\|_2 \leq \frac{d^c\log^{1-c}(1/\delta)}{n\varepsilon a}$$

where $a$ is a term involving $n$ and $d$, to be specified later. For the purpose of causing contradiction, we let:

$$\frac{d^c\log^{1-c}(1/\delta)}{n\varepsilon a} \leq \frac{d}{n\varepsilon\log^{1-c}(d)}$$

This implies:

$$\delta > \exp(-a^{\frac{1}{1-c}}d/\log(d)) \tag{32}$$

We execute Algorithm 1 to purify $\mathcal{M}$ directly over the output space $[-1/\sqrt{d}, 1/\sqrt{d}]^d$. Let $Y$ denote the output of Algorithm 1 and $U \sim \text{Unif}([-1/\sqrt{d}, 1/\sqrt{d}]^d)$. The remainder of the proof involves bounding the additional errors introduced during the purification process. By triangle inequality we have

$$\|\bar{D} - Y\|_2 \leq \underbrace{\|\bar{D} - \mathcal{M}(D)\|_2}_{(a)} + \underbrace{\|\mathcal{M}(D) - Y\|_2}_{(b)}.$$

Notice that under the event that Line 3 of Algorithm 1 doesn't return the uniform random variable, which happens with probability $1 - \omega$, we have $Y = \mathcal{M}(X) + \text{Laplace}^{\otimes d}(2\Delta/\varepsilon)$, so term (b) equals the 2-norm of the Laplace perturbation.

For the remaining proofs, we choose the mixing level $\omega = 1/8$ in Algorithm 1. We now justify the choice of $\delta$:

Observe that since $Y = \mathcal{M}(X) + \text{Laplace}^{\otimes d}(2\Delta/\varepsilon)$, term (b), which accounts for the error introduced by Laplace noise. With probability at least $7/8$ by the concentration of the $L_2$ norm of Laplace vector:

$$\text{(b)} \leq \frac{2\sqrt{d}\Delta \log(8d)}{\varepsilon}$$

Thus, without loss of generality, we will require

$$\frac{2\sqrt{d}\Delta \log(8d)}{\varepsilon} \leq \frac{16d}{n\varepsilon \log(8d)}$$

This implies:

$$\Delta \leq \frac{8\sqrt{d}}{n \log^2(8d)}$$

Notice that:

$$\Delta = d^{1-\frac{1}{q}} \cdot \frac{2R^2}{r} \left(\frac{\delta}{2\omega}\right)^{1/d}$$

Choosing $q = \infty$ (corresponding to the use of $\ell_\infty$ norm in the Wassertain-$\infty$ distance), and noticing $R = 2/\sqrt{d}$ and $r = 1/\sqrt{d}$, we obtain the condition:

$$\Delta = 8\sqrt{d} \left(\frac{\delta}{2\omega}\right)^{1/d} \leq \frac{8\sqrt{d}}{n \log^2(8d)}$$

which further implies:

$$\delta \leq \frac{1}{4n^d \log^{2d}(8d)} \tag{33}$$

By Eq. (32) and Eq. (33), we have:

$$\delta \in \left(\exp\left(-a^{\frac{1}{1-c}}d/\log(d)\right), \quad \frac{1}{4n^d \log^{2d}(8d)}\right) \tag{34}$$

The constrained above yields a lower bound on $a$, after some relaxation for simplicity (and assume $d \geq \log(8d)$ and $d\log(n) > \log(4)$):

$$a^{\frac{1}{1-c}} \geq 2\log(d)\log(nd) \tag{35}$$

Thus, we set $a = (2\log(d)\log(nd))^{1-c}$ to satisfy the constraint stated in Eq. (35), and now

$$\delta \in \left(\frac{1}{\exp(2d\log(nd))}, \quad \frac{1}{4n^d \log^{2d}(8d)}\right) = \left(\frac{1}{n^{2d}d^{2d}}, \quad \frac{1}{4n^d \log^{2d}(8d)}\right) \tag{36}$$

When $\delta$ is within the range above, we have:

$$\log(1/\delta) < 2d\log(nd)$$

This implies that, w.p. at least $1/2$ over the randomness of $\mathcal{M}$ and purification algorithm:

$$\|\bar{D} - Y\|_2 \leq \frac{d}{n\varepsilon \log^{1-c}(d)} + \frac{16d}{n\varepsilon \log(8d)},$$

which violates the lower bound stated in Lemma 38. Thus, for any $(\varepsilon, \delta)$-DP mechanism $\mathcal{M}$ with $\delta$ being in the range of Eq. (36), there exists a dataset $D \in \mathcal{D}$, such that with probability greater than $1/4$ over the randomness of $\mathcal{M}$:

$$\|\mathcal{M}(D) - \bar{D}\|_2 \geq \left( \frac{d^c \log^{1-c}(1/\delta)}{n\varepsilon \left( 2 \log(d) \log(nd) \right)^{1-c}} \right) = \tilde{\Omega} \left( \frac{d^c \log^{1-c}(1/\delta)}{n\varepsilon} \right)$$

Here, $\tilde{\Omega}(\cdot)$ hides all polylogarithmic factors, except those with respect to $\delta$. Since the above derivation holds for arbitraty $c \in (0, 1)$, this implies with probability at least $1/4$ over the randomness of the algorithm $\mathcal{M}$, we have:

$$\|\mathcal{M}(D) - \bar{D}\|_2 \geq \tilde{\Omega} \left( \max_{c \in (0,1)} \frac{d^c \log^{1-c}(1/\delta)}{n\varepsilon} \right)$$

■

### L.2.2 Private Selection

We begin by stating a lower bound for pure differential privacy in the selection setting, as established in [CHS14].

**Lemma 39 (Proposition 1 in [CHS14])** *Let $\varepsilon \in (0, 1)$, $n \geq 2$ and denote item set to be $\mathcal{U}$. For any $\varepsilon$-DP mechanism $\mathcal{A}$, there exist a domain $\mathcal{X}$ and a function $f(i, \cdot)$ which is $(1/n)$-Lipschitz for all item $i \in \mathcal{U}$ such that the following holds with probability at least $1/2$ over the randomness of the algorithm:*

$$\max_{i \in \mathcal{U}} f(i; D) - f(\mathcal{A}(D); D) \geq \Omega \left( \frac{\log(K)}{\varepsilon n} \right).$$

**Theorem 16 (Lower bound for private selection)** *Let $\varepsilon \in (0, 1)$, $\delta \in \left( \frac{\varepsilon^{3d}}{(2d)^{3d}}, \frac{\varepsilon^d}{(2d)^{3d}} \right)$, $n \geq 2$, and $K := |\mathcal{U}| \geq 7$ where $\mathcal{U}$ is the item set. For any $(\varepsilon, \delta)$-DP mechanism $\mathcal{A}$, there exist a domain $\mathcal{X}$ and a function $f(i, \cdot)$ which is $(1/n)$-Lipschitz for all item $i \in \mathcal{U}$ such that the following holds with probability at least $1/2$ over the randomness of the algorithm:*

$$\max_{i \in \mathcal{U}} f(i; D) - f(\mathcal{A}(D); D) \geq \Omega \left( \max_{c \in (0,1)} \frac{\log^c K \log^{1-c}(1/\delta)}{\varepsilon n} \right).$$

**Proof** Without loss of generality, we set $d = \lceil \log_2 K \rceil$, we have that $\log K = \Theta(d)$. For any $c \in (0, 1)$, assume there exists an $(\varepsilon, \delta)$-DP algorithm such that with probability at least $\frac{1}{2} + 2^{-d} + \frac{d}{2\exp(d)}$, for any $D \in \mathcal{X}^n$, we have $\max_{i \in \mathcal{U}} f(i, D) - f(\mathcal{A}(D), D) = \Omega \left( \frac{d^c \log^{1-c}(1/\delta)}{\varepsilon n a} \right)$, with $a$ being some term involved with $n, d$ which will be specified later.

First, to ensure the quality of purification, we need to set $\delta \leq \varepsilon^d (2d)^{-3d}$, this ensures with probability at least $1 - 2^{-d} - \frac{d}{2} \exp(-d)$ over the randomness of purification algorithm, we have $\mathcal{A}^{\text{purified}}(D) = \mathcal{A}(D)$.

Further, in order to fulfill contrast argument, without loss of generality, we require

$$\frac{d^c \log^{1-c}(1/\delta)}{\varepsilon n a} \leq \frac{d}{\varepsilon n \log^{1-c}(d)}$$

which implies:

$$\delta > \exp(-a^{\frac{1}{1-c}} d / \log(d))$$

Thus, to ensure the lower bound of $\delta$ doesn't exceed the upper bound of $\delta$, we require:

$$a^{\frac{1}{1-c}} \geq \log(d) \log \left( \frac{8d^3}{\varepsilon} \right)$$

So, we set $a^{\frac{1}{1-c}} = 3 \log(d) \log \left( \frac{2d}{\varepsilon} \right)$, this implies

$$\delta \in \left( \frac{1}{\exp(3d \log(2d/\varepsilon))}, \quad \frac{\varepsilon^d}{(2d)^{3d}} \right)$$

This implies with probability at least $1/2$,

$$\max_{i \in \mathcal{U}} f(i; D) - f(\mathcal{A}^{\text{purified}}(D); D) \leq \mathcal{O}\left(\frac{d}{\varepsilon n \log^{1-c}(d)}\right)$$

Observe that, under the assumption of $K := |\mathcal{U}| \geq 7$ which implies $d \geq \exp(1)$, the inequality above contradicts Lemma 39. Since $c \in (0,1)$ was chosen arbitrarily, this completes the proof of the stated theorem. ∎

### L.3 An alternative proof for Theorem 9

A $\tilde{\Omega}(d)$ lower bound for mean estimation under $(\varepsilon, \delta)$-DP can also be proved using a packing argument when $\delta$ is exponentially small, as detailed in the theorem below.

**Theorem 17 (Packing lower bound for $(\varepsilon, \delta)$-DP mean estimation)** *Fix constants $\varepsilon > 0$ and $\alpha \in (0,1]$. Let the data domain be $\mathcal{X} = [-1,1]^d$, and let $\mu(D) = \frac{1}{n}\sum_{i=1}^{n} x_i$ for $D \in \mathcal{X}^n$. Suppose a mechanism $\mathcal{M} : \mathcal{X}^n \to \mathbb{R}^d$ is $(\varepsilon, \delta)$-DP under the replace-one neighboring relation and, for every dataset $D$, satisfies*

$$\mathbb{P}\big[\|\mathcal{M}(D) - \mu(D)\|_2 \leq \alpha\big] \geq 2/3.$$

*If $\delta \leq \frac{1}{6} e^{-n\varepsilon}$, then*

$$n \geq \frac{\log 2}{\varepsilon}(d-1).$$

**Proof** For each $v \in \{\pm 1\}^d$, set $\mu_v := v \in [-1,1]^d$ and define $D^{(v)} = (\mu_v, \ldots, \mu_v) \in \mathcal{D}^n$. Then $\mu(D^{(v)}) = \mu_v$. For $u \neq v$, $\|\mu_u - \mu_v\|_2 = 2\sqrt{|\{j : u_j \neq v_j\}|} \geq 2$, so the $\alpha$-balls

$$A_v := \{y \in \mathbb{R}^d : \|y - \mu_v\|_2 \leq \alpha\}$$

are pairwise disjoint for any $\alpha \leq 1$. By the accuracy assumption, for all $v$, we have

$$\mathbb{P}\big[\mathcal{M}(D^{(v)}) \in A_v\big] \geq 2/3. \tag{37}$$

If two datasets differ in at most $h$ positions, then for any measurable $S$, by the group privacy, we have

$$\mathbb{P}[\mathcal{M}(D) \in S] \geq e^{-h\varepsilon}\Big(\mathbb{P}[\mathcal{M}(D') \in S] - \delta_h\Big), \qquad \delta_h \leq \delta \sum_{i=0}^{h-1} e^{i\varepsilon} \leq \delta \frac{e^{h\varepsilon} - 1}{e^{\varepsilon} - 1} \leq \delta e^{h\varepsilon}.$$

Notice that in our setting, $\text{dist}(D^{(u)}, D^{(v)}) = n$ (when every coordinate is substituted), hence for $S = A_v$,

$$\mathbb{P}\big[\mathcal{M}(D^{(u)}) \in A_v\big] \geq e^{-n\varepsilon}\Big(\mathbb{P}\big[\mathcal{M}(D^{(v)}) \in A_v\big] - \delta_n\Big) \geq e^{-n\varepsilon} \cdot \frac{2}{3} - \delta, \tag{38}$$

where we used $\delta_n \leq \delta e^{n\varepsilon}$ so $e^{-n\varepsilon}\delta_n \leq \delta$.

For fixed $u$, by the disjointness of the $A_v$'s, we have:

$$1 \geq \sum_v \mathbb{P}[\mathcal{M}(D^{(u)}) \in A_v] = \underbrace{\mathbb{P}\big[\mathcal{M}(D^{(u)}) \in A_u\big]}_{\geq 2/3 \text{ by (37)}} + \sum_{v \neq u} \underbrace{\mathbb{P}\big[\mathcal{M}(D^{(u)}) \in A_v\big]}_{\geq e^{-n\varepsilon}\frac{2}{3} - \delta \text{ by (38)}}.$$

which implies:

$$1 \geq \frac{2}{3} + (2^d - 1)\Big(\frac{2}{3}e^{-n\varepsilon} - \delta\Big) \quad \Rightarrow \quad 2^d \leq 1 + \frac{1/3}{\frac{2}{3}e^{-n\varepsilon} - \delta}.$$

If $\delta \leq \frac{1}{6}e^{-n\varepsilon}$ then the denominator is at least $\frac{1}{2}e^{-n\varepsilon}$, and so

$$2^d \leq 1 + \frac{1/3}{\frac{1}{2}e^{-n\varepsilon}} \leq 2e^{n\varepsilon}.$$

Taking logrithm on both sides yields $d \log 2 \leq n\varepsilon + \log 2$, i.e., $n \geq \frac{\log 2}{\varepsilon}(d-1)$. ∎

# M  Technical Lemmas

## M.1  Supporting Results on Sparse Recovery

For completeness, we introduce the results from sparse recovery [Tia24] that is used in Section 4.2 and Appendix I.

**Definition 40 (Numerical sparsity)** *A vector $x$ is $s$-numerically sparse if $\frac{\|x\|_1^2}{\|x\|_2^2} \leq s$.*

Numerical sparsity extends the traditional notion of sparsity. By definition, an $s$-sparse vector is also $s$-numerically sparse. A notable property of numerical sparsity is that the difference between a sparse vector and a numerically sparse vector remains numerically sparse, as stated in the following lemma.

**Lemma 41 (Difference of numerically sparse vectors)** *Let $x \in \mathbb{R}^d$ be an $s$-sparse vector. For any vector $x' \in \mathbb{R}^d$ satisfying $\|x'\|_1 \leq \|x\|_1$, the difference $x' - x$ is $4s$-numerically sparse.*

**Proof** Let $S := \{i \in [d] \mid x[i] \neq 0\}$ and denote $v := x' - x$. We have

$$\|x'\|_1 = \|x + v_S + v_{S^c}\|_1 = \|x + v_S\|_1 + \|v_{S^c}\|_1 \geq \|x\|_1 - \|v_S\|_1 + \|v_{S^c}\|_1 \geq \|x'\|_1 - \|v_S\|_1 + \|v_{S^c}\|_1,$$

which implies $\|v_S\|_1 \geq \|v_{S^c}\|_1$. Therefore,

$$\begin{aligned}
\|v\|_1 &= \|v_S\|_1 + \|v_{S^c}\|_1 \\
&\leq 2\|v_S\|_1 \\
&\leq 2\sqrt{s}\|v_S\|_2 \\
&\leq 2\sqrt{s}\|v\|_2
\end{aligned}$$

which implies $\|v\|_1^2 \leq 4s\|v\|_2^2$. Thus, by Definition 40, $v$ satisfies $4s$-numerically sparse. ∎

If the vector $x$ is $s$-sparse, we can reduce its dimension while preserving the $\ell_2$ norm using matrices that satisfy the Restricted Isometry Property.

**Definition 42 ($(e, s)$-Restricted isometry property (RIP))** *A matrix $\Phi \in \mathbb{R}^{k \times d}$ satisfies the $(e, s)$-Restricted Isometry Property (RIP) if, for any $s$-sparse vector $x \in \mathbb{R}^d$ and some $e \in (0, 1)$, the following holds:*

$$(1 - e)\|x\|_2^2 \leq \|\Phi x\|_2^2 \leq (1 + e)\|x\|_2^2.$$

For numerically sparse vectors, we can reduce the dimension while preserving utility by matrices satisfying a related condition – the Restricted well-conditioned (RWC).

**Definition 43 ($(e, s)$-Restricted well-conditioned (RWC) ([Tia24], Definition 4))** *A matrix $\Phi \in \mathbb{R}^{k \times d}$ is $(e, s)$-Restricted well-conditioned (RWC) if, for any $s$-numerically sparse vector $x \in \mathbb{R}^d$ and some $e \in (0, 1)$, we have*

$$(1 - e)\|x\|_2^2 \leq \|\Phi x\|_2^2 \leq (1 + e)\|x\|_2^2.$$

**Lemma 44 ([Tia24], Lemma 2; [CT05], Theorem 1.4)** *Let $\Phi \in \mathbb{R}^{k \times d}$ whose entries are independent and identically distributed Gaussian with mean zero and variance $\mathcal{N}(0, \frac{1}{k})$. For $e, \zeta \in (0, 1)$, if*

$$k \geq C \cdot \frac{s \log\left(\frac{d}{s}\right) + \log\left(\frac{1}{\zeta}\right)}{e^2},$$

*for an appropriate constant $C$, then $\Phi$ satisfies $(e, s)$-RIP with probability $\geq 1 - \zeta$.*

There is a connection between RIP and RWC matrices:

**Lemma 45 ([Tia24], Lemma 5)** *For $\Phi \in \mathbb{R}^{m \times n}$ and $e \in (0, 1)$, if $\Phi$ is $(\frac{e}{5}, \frac{25s}{e^2})$-RIP, then $\Phi$ is also $(e, s)$-RWC.*

Finally, we provide the following guarantee for Algorithm 8.

**Lemma 46 ($\ell_1$ error guarantee from sparse recovery)** *Let $\Phi \in \mathbb{R}^{m \times n}$, and $\theta_* \in \mathbb{R}^n$. Given noisy observation $b = \Phi\theta_* + \tilde{w}$ with bounded $\ell_1$ norm of noise, i.e. $\|\tilde{w}\|_1 \leq \xi$, consider the following noisy sparse recovery problem:*

$$\hat{\theta} = \arg\min_{\theta} \|\theta\|_1$$

$$\text{s.t. } \|\Phi\theta - b\|_1 \leq \xi$$

*where $\xi > 0$ is the constraint of the noise magnitude. Suppose that $\theta$ is an $s$-sparse vector in $\mathbb{R}^n$ and that $\Phi$ is a $(4s, e)$-RWC matrix. Then, the following $\ell_1$ estimation error bound holds:*

$$\|\hat{\theta} - \theta_*\|_1 \leq \frac{4\sqrt{s}}{\sqrt{1-e}} \cdot \xi$$

*Moreover, the problem can be solved in $\mathcal{O}((3m + 4n + 1)^{1.5}(2n + m)\texttt{prec})$ arithmetic operations in the worst case, with each operation being performed to a precision of $\mathcal{O}(\texttt{prec})$ bits.*

**Proof** For utility guarantee: By $\|\tilde{w}\|_1 \leq \xi$, $\theta_*$ is a feasible solution. Thus, we have $\|\hat{\theta}\|_1 \leq \|\theta_*\|_1$, which implies $h := \hat{\theta} - \theta_*$ is $4s$-numerically sparse by Lemma 41. Since $\Phi$ is $(e, 4s)$-RWC, we have:

$$(1 - e)\|h\|_2^2 \leq \|\Phi h\|_2^2 \leq (1 + e)\|h\|_2^2 \tag{39}$$

Now it remains to bound $\|h\|_1$:

$$\begin{aligned}
\|h\|_1 &\leq \sqrt{4s}\|h\|_2 \\
&\leq \sqrt{4s} \cdot \frac{\|\Phi h\|_2}{\sqrt{1-e}} \\
&\leq \frac{2\sqrt{s}}{\sqrt{1-e}}(\|\Phi\hat{\theta} - b\|_1 + \|\Phi\theta_* - b\|_1) \\
&\leq \frac{4\sqrt{s}}{\sqrt{1-e}} \cdot \xi
\end{aligned} \tag{40}$$

where the last inequality is by feasibility of $\hat{\theta}$ and the structure of $b$.

Now we prove the runtime guarantee. We first reformulate this problem to Linear Programming:

$$(P) \qquad \min_{\theta, u^+, u^-, v} \sum_{i=1}^n (u_i^+ + u_i^-)$$

subject to:

$$\begin{aligned}
\theta_i &= u_i^+ - u_i^-, \quad u_i^+, u_i^- \geq 0, \quad \forall i = 1, \ldots, n, \\
\Phi_j \theta - b_j &\leq v_j, \quad \forall j = 1, \ldots, m, \\
-(\Phi_j \theta - b_j) &\leq v_j, \quad \forall j = 1, \ldots, m, \\
\sum_{j=1}^m v_j &\leq \xi, \\
v_j &\geq 0, \quad \forall j = 1, \ldots, m.
\end{aligned}$$

The problem $(P)$ has $2n + m$ variables and $2m + 2n + 1$ constraints. By [Vai89], this can be solved in $\mathcal{O}((3m + 4n + 1)^{1.5}(2n + m)B))$ arithmetic operations in the worst case, with each operation being performed to a precision of $\mathcal{O}(B)$ bits. ∎

### M.2  A Concentration Inequality for Laplace Random Variables

**Definition 47 (Laplace Distribution)** $X \sim Lap(b)$ *if its probability density function satisfies* $f_X(t) = \frac{1}{2b}\exp\left(-\frac{|x|}{b}\right)$.

**Lemma 48 (Concentration of the $\ell_1$ norm of Laplace vector)** *Let $X = (x_1, \ldots, x_k)$ with each $x_i$ independently identically distributed as $Lap(b)$. Then, with probability at least $1 - \zeta$,*

$$\|X\|_1 \leq 2kb + 2b\log(1/\zeta).$$

**Proof** $\|X\|_1 = \sum_{i=1}^k |x_i|$ follows the Gamma distribution $\Gamma(k, b)$, with probability density function $f(x) = \frac{1}{\Gamma(k)b^k}x^{k-1}e^{-\frac{x}{b}}$. Applying the Chernoff's tail bound of Gamma distribution $\Gamma(k, b)$, we have

$$\mathbb{P}(\|X\|_1 \geq t) \leq \left(\frac{t}{kb}\right)^k e^{k - \frac{t}{b}}, \text{ for } t > kb.$$

Taking $t = 2kb + 2b\log(1/\zeta)$, we have

$$\mathbb{P}(\|X\|_1 \geq 2kb + 2b\log(1/\zeta)) \leq \left(\frac{2kb + 2b\log(1/\zeta)}{kb}\right)^k e^{k - \frac{2kb + 2b\log(1/\zeta)}{b}}$$

$$\leq 2^k \left(1 + \frac{\log(1/\zeta)}{k}\right)^k e^{-k}\zeta^2$$

$$\leq \frac{1}{\zeta}\zeta^2 \left(\frac{2}{e}\right)^k$$

$$\leq \zeta.$$

■

