# OpenReview forum: "Purifying Approximate Differential Privacy with Randomized Post-processing"
_NeurIPS.cc/2025/Conference — NeurIPS 2025 spotlight_

### Official Review · Reviewer_XywD · 2025-06-24

**Clarity:** 4
**Significance:** 2
**Originality:** 3
**Rating:** 4
**Confidence:** 4

**Summary:**

Differential privacy (DP) can be viewed as a bound on the difference between mechanism output distributions induced by neighboring datasets. This difference is measured in terms of the log probability ratio (privacy loss) between the two distributions. The two main variants of DP are pure-DP (the privacy loss is bounded by $\epsilon$ a.s.), and approximate-DP (roughly speaking, the privacy loss exceeds $\epsilon$ w.p. $\approx\le\delta$).

It is considered folklore that approx-DP mechanisms $\mathcal{M}$ with finite range can be turned into pure-DP by a simple random post-process, which w.p. $\omega$ samples uniformly at random some output and w.p. $1-\omega$ returns the output of $\mathcal{M}$. Scaling $\omega$ one can balance the degradation in privacy ($\epsilon$) and utility. Intuitively speaking, this process trades the low probability event of a large privacy loss for the low probability event of large utility loss.

This works moves beyond finite range, and prove that if in the $1-\omega$ case the post processing adds a small amount of Laplace noise, this result extends to any range contained in a $q$-norm ball in $\mathbb{R}^{d}$. This is achieved by converting TV distance to a variant of Wasserstein distance, thus using the known result which converts $(\epsilon, \delta)$-approx-DP to "$\delta$-TV distance away from $\epsilon$-pure-DP".

**Questions:**

Please address my two concerns.

**Ethical Concerns:**

["NO or VERY MINOR ethics concerns only"]

**Final Justification:**

The authors have provided detailed response to my concerns.

The existence of a relatively simple alternative algorithm with a simple analysis somewhat reduces the importance of this work, but it's contribution remains solid and important.

**Quality:**

2

**Strengths And Weaknesses:**

**Strengths**
1. While approx-DP is typically treated as sufficiently private for most use cases, pure-DP is a much cleaner and natural definition, so the ability to convert nearly any approx-DP mechanism to pure-DP is an important contribution.
2. From the theoretical perspective, this work is even more valuable, since lower bounds for pure-DP are generally tighter than those known for approx-DP. Using the proposed conversion, these lower bounds can be extended in some cases.
3. The authors survey a large range of classical approx-DP mechanisms, and analyze how purifying them affects their utility.

**Weaknesses**

I have two concerns regarding this work, which - if satisfactory addressed can affect my score.

*Main concern*

The authors do not consider a natural baseline, which - at first glance - seems to provide similar utility guarantees in slightly more general setting, without the need for complex proofs. It might very well be the case that this baseline has some limitations or is inferior in some other way, but without a direct comparison it is hard for me to assess the contribution of this work.

The proposed alternative is simple. Cover the range by a $\Delta$-net, round the output of the approx-DP mechanism to the nearest grid point, then purify using the finite range method. The utility error will be the sum of the low probability event of random output and the rounding error. Using this technique, one can purify a $(\epsilon, \delta)$-DP mechanism into a $2 \epsilon$-DP mechanism by bounding the number of grid points by $e^{\epsilon} (e^{\epsilon}-1)\frac{\omega}{\delta}$. The number of $\Delta$-net points required to cover a $q$-norm $d$ ball is $\approx \left(\frac{R}{\Delta}\right)^{d}\cdot\frac{1}{\Gamma(1+d/q)}$ (its volume divided by $\Delta^{d}$). Combining the two we get $\Delta \approx R \left(\Gamma(1+d/q) e^{\epsilon} (e^{\epsilon}-1)\frac{\omega}{\delta} \right)^{1/d}$ which can be further simplified using Stirling's approximation.

This expression shares many similarities with the one proposed in this work (Algorithm 1), in particular the relation of $\delta$ and $d$, while differing in others (which might be lower order effects). Besides it simplicity, this method does not depend on the inner radius $r$, which makes this result more general. In fact, it does not depend on the diameter either, only on the range's volume.

*Minor gaps*

The list of utility comparisons for the various purifying techniques mostly lacks the results for the SOTA pure-DP mechanisms. The method proposed in this work is of significant importance, even if the approx-DP -> purify route is sub-optimal relative to the directly pure algorithm in some cases, but this comparison is crucial for a honest analysis of this method.

---

> ### Author Rebuttal · Authors · 2025-07-31
>
> Thank you for reviewing our paper and for raising the questions!
>
> > Major concern
>
> Thank you for raising this thoughtful question. We will add a detailed discussion of this baseline in the revision.
>
> The baseline you described—purifying a $(\epsilon, \delta)$-DP mechanism via discretization using a $\Delta$-net—is indeed interesting and shares some similarities with our method (e.g., dependence on $\delta$ and $d$). However, implementing this baseline efficiently is not straightforward in general domains.
>
> A key difficulty is the explicit construction of the $\Delta$-net and efficient sampling from it. If the domain is a hypercube, then one can construct a uniform grid and sample efficiently; in that case, the utility of this baseline can be comparable to our method. However, when the domain is an $\ell_q$ ball with $q<\infty$ (e.g., $q=1,2$), explicitly constructing a $\Delta$-net and uniformly sampling from it with polynomial computation complexity in $d$ becomes more challenging.
>
> One clue to the difficulty comes from the geometry of $\ell_q$ balls in high dimensions: most of the volume concentrates near the boundary. As a result, any sufficiently fine covering net must include many points near the surface. Furthermore, although uniform sampling is not strictly necessary—purification only requires a distribution with a lower-bounded PMF—this geometric property suggests that using an “enlarged” $\ell_q$ ball to simplify sampling would not help, since the ratio of the volumes of the original and enlarged balls becomes exponentially small in $d$. Alternatively, if one uses an enclosing hypercube ($\ell_\infty$ ball) to simplify the $\Delta$-net construction and sampling, this introduces a mismatch between the $\ell_q$ and $\ell_\infty$ norms. As a result, the expected $\ell_q$ error will carry an additional factor of $d^{1/q}$ due to the conversion between norms.
>
> In contrast, our method is efficiently implementable for general domains with diameter $R$. In Algorithm 1, we purify over a set $\Theta$ that contains $\text{Range}(\mathcal{M})$ (specified in the input line of Algorithm 1). As discussed in Remark 3 (lines 134–135), $\Theta$ can be chosen as an $\ell_q$ ball that contains the range of the approximate DP algorithm. Its construction is simple: if $\text{Range}(\mathcal{M})$ has $\ell_q$ diameter $R/2$, we can take $\Theta$ to be an $\ell_q$ ball of diameter $R$ centered at any point in $\text{Range}(\mathcal{M})$. Uniform sampling from $\ell_1$, $\ell_2$, and $\ell_\infty$ balls can all be done with $O(d)$ time.
>
> Regarding the parameters $r$ and $R$: if $\Theta$ is an $\ell_q$ ball of diameter $R$, then $r = R/2$. We include $r$ for flexibility—for example, $\Theta$ could be a thin rectangular band of width $2r$ and length $R$—but $r$ can be eliminated in the standard $\ell_q$-ball case.
>
>
> Finally, we would like to emphasize that even if the $\Delta$-net method can be implemented efficiently on general domains, our paper remains significant, as the adaptation of approximate DP methods to achieve optimal utility under pure DP in various settings has not been previously studied. The $\Delta$-net discretization-based approach should be viewed as an alternative to the algorithm we proposed.
>
> Thanks again for raising this question, and we will add a section in the appendix to discuss this approach thoroughly.
>
>
>
> > Minor concern
>
> Thank you for your question. We did compare our results to baselines and lower bounds in Section 4-6 when we formally state the results for each problem in Table 1. We will make the comparisons with information-theoretic lower bounds more clearly tabulated in the final version.  A quick summary below for your convenience.
>
> Rows 1-2 (DP-ERM): Our results match the lower bounds from Table 1 in [BST14] (up to logarithmic factors).
>
> Row 3 (DP-Frank Wolfe): our result matches a lower bound proved in Lemma 28 (up to logarithmic factors).
>
> Row 4 (PTR type): To our knowledge, there isn’t a pure DP mechanism that works in this setting, so there isn’t a baseline. We can try constructing a packing lower bound and include it in the final version.
>
> Row 5 (Margin-based Mode Release): It matches the lower bound in Proposition 1 of [CHS14] (up to logarithmic factors)
>
> Row 6 (Linear Regression): Under the assumption of bounded data and parameter space, our result matches the lower bounds from Table 1 in [BST14].  To see this,    For example,  in the Convex Lipschitz case,  the [BST-14] lower bound is $\Omega(L\|\Theta\|d/\epsilon n)$ when we choose $L$ to be an upper bound of the gradient norm $\|x(x^T\theta - y)\|$ (assume $|y| \leq \|\mathcal{X}\| \|\mathcal{\Theta}\|)$  we can take $L = \|\mathcal{X}\|^2 \|\mathcal{\Theta}\|$) this matches our upper bound.  Similarly, in the strongly convex case, the [BST-14] lower bound is $\Omega(L^2d^2/\epsilon^2 n^2 \lambda)$, which also matches our upper bound when substituting the $L$ parameter above and choosing $\lambda = \lambda_{\min}$.
>
> Row 7 (Query Release): Our result matches the utility upper bound of the SmallDB algorithm (and private multiplicative weights) up to logarithmic factors, which represents the state-of-the-art for pure DP query release. Whether this rate can be further improved is a well-known open question in the DP theory community.

---

> > ### Comment · Reviewer_XywD · 2025-08-01
> >
> > I thank the authors for their response, though I must admit the answer did not sufficiently clarify the issue.
> >
> > To the best of my knowledge given a ball of radius $R$, a point $x$ in the ball (including its surface), and some $r \ll R$, the volume of the intersection of the ball of radius $r$ around $x$ and the first ball is at least half of the volume of ball of radius $r$ up to lower order terms in $r/R$. This is true in any norm. If this is indeed the case, sampling uniformly from the ball and rounding to the nearest grid point contained in that ball, should provide a lower bound on the sampling probability of each output with at most a constant factor, so why can't it be sampled efficiently?
> >
> > The list of comparisons to lower bounds helps clarify the story. I would recommend adding (perhaps to the appendix) an extended version of Table 1 including both lower bounds as well as upper bounds from SOTA algorithms under pure-DP to complete the picture.

---

> > > ### Author Response · Authors · 2025-08-04
> > >
> > > Thank you for the reviewer's discussion.  We clarify that the following statement is not correct in general:
> > >
> > > > To the best of my knowledge given a ball of radius $R$, a point $x$ in the ball (including its surface), and some  $r<<R$, the volume of the intersection of the ball of radius $r$ around $x$ and the first ball is at least half of the volume of ball of radius $r$ up to lower order terms in $r/R$. This is true in any norm.
> > >
> > > **This is false for $\ell_1$ norm.** We provide the following counter example: consider the $\ell_1$ ball of radius $R$ and take the point $x=(R,0,\dots,0)$ on its surface. Then the volume of the intersection: $$\\{z:\|z\|_1\leq R, \ \|z-x\|_1\leq r\\},$$
> > > **is only a $\frac{1}{2^d}$ fraction of the ball $\\{z:\|z-x\|_1\leq r\\}$.**
> > >
> > > We now provide the derivation. Writing in coordinates: $$\sum_{i=1}^d |z_i|\leq R, \ |z_1-R|+\sum_{i=2}^d |z_i|\leq r,$$
> > >
> > > we get $R-r\le z_1\le R$, and for such $z_1$, the other coordinates must satisfy
> > >
> > > $$\sum_{i=2}^d |z_i|\leq \min\\{R-z_1, r-R+z_1\\}$$
> > >
> > > Therefore, the intersection volume is
> > > $\int_{R-r\le z_1\le R}\int_{\sum_{i=2}^d |z_i|\leq \min\{R-z_1, r-R+z_1\}}\mathbf{1}d(z_2, \dots, z_d)d z_1$,
> > > $=\int_{R-r\le z_1\le R}\frac{2^{d-1}}{(d-1)!}(\min\\{R-z_1, r-R+z_1\\})^{d-1}d z_1$
> > > $=\frac{2^{d-1}}{(d-1)!}(\int_{R-r\le z_1\le R-r/2}(r-R+z_1)^{d-1}dz_1+\int_{R-r/2\le z_1\le R}(R-z_1)^{d-1})dz_1$
> > > $=\frac{2^{d-1}}{(d-1)!}\cdot 2 \frac{1}{d}(\frac{r}{2})^d=\frac{1}{d!}r^d$,
> > > which is $\frac{1}{2^d}$ portion of the volume of the ball $\\{z: \|z-x\|_1\leq r\\}$.
> > >
> > > We also note that the reviewer's claim does hold for $\ell_2$ norm, see Lemma 24 in Ref. [50] of the manuscript.
> > >
> > > Finally, we emphasize that our method is simple to implement, though the theoretical analysis is more involved.
> > >
> > > We sincerely thank the reviewer for bringing up this insightful and interesting point, and will include a section to discuss this baseline in the revised manuscript.

---

> > > > ### Comment · Reviewer_XywD · 2025-08-04
> > > >
> > > > Indeed, the argument I provided was only a hand-wavy outline, and the actual analysis will require dealing with the various edge cases, but I don't expect it to require an involved algorithm or analysis.
> > > >
> > > > In any case, the authors have clarified the picture. I will keep my score and look forward to see the final version.

---

> > > > > ### Author Response · Authors · 2025-08-05
> > > > >
> > > > > Thank you for reviewing our paper and for your insightful suggestions. We will incorporate them in the next revision.

---

### Official Review · Reviewer_nLKU · 2025-06-30

**Clarity:** 3
**Significance:** 2
**Originality:** 3
**Rating:** 4
**Confidence:** 2

**Summary:**

The paper presents a novel "purification" framework built on randomized post-processing techniques. This framework transforms mechanisms with $(\epsilon, \delta)$-approximate differential privacy into $(\epsilon + \epsilon', 0)$-pure differential privacy mechanisms, albeit with additional utility costs. The authors demonstrate the framework's versatility by applying it to differentially private empirical risk minimization (DP-ERM), stability-based data release mechanisms, and query processing tasks.

Another theoretical contribution of this framework is its use of purification to derive lower bounds for approximate differential privacy that explicitly depend on $\delta$. This provides a novel proof methodology that serves as an alternative to traditional fingerprinting approaches.

**Questions:**

1. **Mechanism Design**: Doesn't designing approximate DP mechanisms typically present greater challenges than pure DP mechanisms? Given that smaller $\delta$ values bring us closer to pure DP guarantees, does the purification algorithm consistently select $\log(1/\delta) \approx d$ as its operating parameter?

2. **Rate Discrepancies**: In Table 1, why do Theorems 4 and 7 yield different convergence rates with respect to $n\epsilon$ ? What accounts for this theoretical difference?

3. **Norms**: The domain assumptions appear to be tailored for $\ell_p$ norms. What represents the primary obstacle in extending this framework to more general norm spaces?

4. **Notation**: In the introduction, you use the notation $\mathcal{O}_p$. Should we interpret this as $p = q$ throughout the analysis?

**Ethical Concerns:**

["NO or VERY MINOR ethics concerns only"]

**Final Justification:**

The authors answered my qestions by underlying the theoretical contribution and the practical relevence of their work.

**Limitations:**

yes

**Paper Formatting Concerns:**

No major issue found

**Quality:**

3

**Strengths And Weaknesses:**

The paper is well written and presents an intellectually compelling approach to bridging approximate and pure differential privacy paradigms. The quest for a unified theoretical framework that can transform approximate DP solutions into pure DP guarantees represents an ambitious and potentially valuable research direction.

However, my main concern is about the practical relevance of this work. General theoretical frameworks of this nature often suffer from being too abstract to yield meaningful practical benefits. To better evaluate the contribution, I would appreciate a comparative analysis showing how existing pure DP solutions from the literature (as referenced in Table 1) perform relative to their purified approximate DP counterparts. A clearer articulation of when purification offers advantages versus when direct pure DP methods are preferable would substantially strengthen the paper.

---

> ### Author Rebuttal · Authors · 2025-07-31
>
> Thank you for reviewing our paper and for raising the questions!
>
> > Q1: Mechanism Design
>
> Designing pure DP mechanisms is more challenging than designing approximate DP mechanisms.
>
> * While every $\varepsilon$-DP mechanism trivially satisfies $(\varepsilon, \delta)$-DP for any $\delta > 0$, the reverse is not true. Approximate DP mechanisms do not necessarily satisfy pure DP. This strict containment makes the design space for pure DP strictly smaller and harder to work with.
>
> * From an algorithmic perspective, the set of known techniques for constructing pure DP mechanisms remains limited (we provide a more detailed discussion in the introduction). A key contribution of our paper is the purification technique, which offers a computationally efficient reduction from the design of pure DP mechanisms to that of approximate DP mechanisms. As shown in Section 4.1, for DP-ERM problems, existing pure DP mechanisms either suffer from computational inefficiency (e.g., the exponential mechanism; Table 2, Row 3) or suboptimal utility due to relying on basic composition (e.g., the Laplace mechanism; Table 2, Row 1). In contrast, our purification method achieves both computational efficiency and near-optimal utility under pure DP guarantees, through purification of DP-SGD using the Gaussian noise.
>
> * Regarding the use of $\log(1/\delta) \approx d$, this approximation is used only to preserve utility guarantees when comparing with state-of-the-art pure DP mechanisms. Our purification algorithm can be applied to mechanisms with larger $\delta$, but in such cases, the resulting utility may no longer be (nearly) optimal.
>
> > Q2: Rate Discrepancies
>
> In Table 1, Theorem 4 provides the utility guarantee for the DP Frank-Wolfe algorithm, which is designed to solve the DP-ERM problem under an $\ell_1$-ball constraint on parameter space. In contrast, Theorem 7 addresses DP linear regression under unconstrained parameter space (as described in Algorithm 13). These two problems have different constraint sets, objective functions, and are solved by different classes of algorithms, so it is not surprising that their convergence rates with respect to $n\varepsilon$ are not the same.
>
> >Q3: Norms
>
> Thank you for this interesting question.
> To extend the conversion from TV distance to infinity-Wasserstein distance (Lemma 6) for a more general norm $\Vert \cdot \Vert_{gen}$, the key is that the Lebesgue measure of the unit ball $\\{x:\Vert x \Vert_{gen}<1\\}$ can be explicitly calculated or lower bounded. For a general norm, this may not be guaranteed. Please refer to the response to Reviewer tAS6 for the proof sketch: The Lebesgue measure of the unit ball, together with density (i.e., Radon–Nikodym derivative with respect to the Lebesgue measure) lower bound $p_{min}$ gives the lower bound of the $\nu$-measure of the small ball constructed in the expansion band.
>
> We would also like to point out that the $\ell_q$ norm is already quite general. In fact, for obtaining pure DP mechanisms through adding Laplace noise, the $\ell_1$ version $W_{\infty}^{\ell_1} (\mu, \nu)$ is sufficient. In addition, for obtaining pure Gaussian DP mechanisms through adding Gaussian noise, the $\ell_2$ version $W_{\infty}^{\ell_2} (\mu, \nu)$ is sufficient.
>
> > Q4: Notation
>
> Thank you for pointing this out. We will clarify the notation $O_p(\cdot)$ in the next version. It denotes stochastic boundedness—roughly speaking, the quantity is bounded with high probability. For a formal definition, please refer to Section 2.2 of [Va00]. In our context, $O_p(k / n\varepsilon)$ indicates that the Laplace query error is bounded by $k / n\varepsilon$ with high probability. The  $p$ in $O_p ( \cdot )$ is unrelated to the $p$ used in the $L_p$ norm; it stands for "in probability".
>
> Reference:
>
> [Va00] Van der Vaart, Aad W. Asymptotic statistics. Vol. 3. Cambridge university press, 2000.
>
>
> >  Regarding your comments on weakness
>
> For many of the settings we consider, there are no known pure-DP mechanisms that achieve comparable utility or computational guarantees. For example: (1) there is no pure DP counterpart to the propose-test-release framework; and (2) existing pure DP methods for DP-ERM are either suboptimal in utility or incur computational overhead (see Table 2).
>
> Regarding the concern that our “general theoretical framework is too abstract to yield meaningful practical benefits”: We would like to emphasize that our purification algorithm is easy-to-implement. In most cases, it involves adding properly calibrated Laplace noise to the output of an approximate DP algorithm. For discrete outputs, it may additionally include a straightforward binary encoding step. Both are computationally efficient. In addition, it leads to pure DP algorithms with strong utility guarantees across a variety of applications, as demonstrated by the near-optimal utility guarantee shown in Table 1. (Please also refer to our response to Reviewer XywD for further details.)

---

> > ### Author Response · Authors · 2025-08-07
> >
> > Dear Reviewer nLKU,
> >
> > We sincerely appreciate your time and effort in reviewing our paper. We have carefully addressed your concerns in the rebuttal and would greatly appreciate it if you could review our response and consider updating the rating and final justification based on our rebuttal. Please don’t hesitate to let us know if there’s anything further we can clarify. Thanks again for your valuable review.
> >
> > Best,
> > Submission23564 Authors

---

> > ### Comment · Reviewer_nLKU · 2025-08-07
> >
> > I would like to thank the authors for their answers to my questions and other reviewers' questions. Consider the score up to 4.

---

### Official Review · Reviewer_tAS6 · 2025-07-01

**Clarity:** 3
**Significance:** 4
**Originality:** 4
**Rating:** 5
**Confidence:** 3

**Summary:**

This paper offers an efficient generic method to take an arbitrary approximate DP algorithm $M \colon X \rightarrow Y$ (i.e., $(\varepsilon,\delta)$-DP with $\delta>0$) with bounded $Y \subseteq R^d$ (Namely, that $Y$ is contained by an $\ell_q$ ball of radius $R$ and contains an $\ell_q$ ball of radius $r$), and transform it into a pure $\varepsilon’$-DP algorithm, where the resulting $\ell_q$ error is a function of all parameters (and more additional parameters like the internal mixture parameter $\omega \in [0,1]$). For finite output spaces, this was already known (mentioned in Appendix C): Given a dataset $D$ and an approximate DP algorithm $M \colon X \rightarrow Y$, output $M(D)$ with probability $1-\omega$ (for some parameter $\omega$), and otherwise output a uniform element in the range $Y$. But when $Y$ is infinite, the authors show that this method simply does not work. The first step of their solution is essentially the same, but this alone is not enough in the infinite $Y$ case and requires an additional step: They show that under some conditions on the parameters, for any neighboring datasets $D,D’$, an $\omega$-mixure of $M(D)$ with $Unif(Y)$ (denote by $\mu$) and an $\omega$-mixure of $M(D’)$ with $Unif(Y)$ (denote by $\mu’$) has a small enough TV distance that enables coupling in a way the ensures that if we draw $(x,x’) \sim (\mu,\mu’)$ then almost surely the $\ell_q$ distance between $x$ and $x’$ is bounded (this is called the $\infty$-Wasserstain distance). Given that, they simply add an additional step of Laplace noise per coordinate to guarantee that the final outputs are purely indistinguishable.

They illustrate the applicability of their method by presenting new pure-DP algorithms for Stochastic Gradient Descent, Frank-Wolfe Algorithm, Propose-Test-Release, Linear Regression, and Query Release. In all (or most of) these examples, they match state-of-the-art accuracy results, but sometimes with improved running time, like in Stochastic Gradient Descent and Query Release.

In addition to all of that, they show that their method is even useful for proving lower bounds for approximate DP algorithms, because for some problems, it is easier to provide pure DP lower bound, so if an accuracy guarantee for approximate DP is translated (using their method) to an pure DP accuracy that violates the lower bound, then we conclude that the approximate DP accuracy guanatee is impossible. They apply this method to the problem of estimating the average of points in the $d$-dimensional cube, and prove, using their method, a lower bound that is almost as the fingerprinting lower bound (but without the need to use the fingerprint machinery).

**Questions:**

I’d appreciate it if you could provide a high-level intuition for the proof of Lemma 6 (converting TV distance to $W_{\infty}$) under simplified assumptions like d=1 or 2 (and maybe using more relaxations). I just want to get the feeling of how the coupling between $\mu$ and $\nu$ looks.

**Ethical Concerns:**

["NO or VERY MINOR ethics concerns only"]

**Final Justification:**

The authors answered my question. I agree with the final justification of Reviewer JJ6eI.  I will keep my positive score.

**Limitations:**

yes

**Paper Formatting Concerns:**

No formatting concerns

**Quality:**

4

**Strengths And Weaknesses:**

The statements and proofs are not easy to read, and I believe it is possible to simplify them and make them more accessible. But besides that, this paper is one of the most interesting and significant papers that I have reviewed for NeurIPS, and therefore, I support acceptance.

In general, I’m not a fan of pure-DP. I think it is too stringent, and I don’t think that allowing a small $\delta$ loss should be considered as a real privacy concern. But I do appreciate the improved pure DP upper bounds, especially since they have been achieved using a simple and generic method. Furthermore, even for someone like me, this paper offers a new method to prove approximate DP lower bounds, and I really like that.

---

> ### Author Rebuttal · Authors · 2025-07-31
>
> Thank you for your support and your question!
>
> To prove Lemma 6, we use an equivalent definition of infinity-Wasserstein distance (see Lemma 12 in our manuscript for more details, or Proposition 5 of [GS84]):
> $$
>     W_{\infty}^{\ell_q} (\mu, \nu) = \inf\{ \alpha > 0 : \mu(U) \leq \nu(U^\alpha), \  \textrm{for all open subsets } \ U \subset \Theta\},
> $$
> where the $\alpha$-expansion of $U$ is denoted by $U^\alpha := \{x \in \Theta : \|x-U\|_q \leq \alpha\}$.
>
> This definition does not involve the coupling, and gives a geometric interpretation by comparing the probability measures on open sets and their expansion. This makes it compatible with the TV distance because TV distance compares the probability measures on the same measurable sets:
> $$TV(\mu, \nu) = \sup_{U \in \mathcal{F}} |\mu(U) - \nu(U)|.$$
>
> We summarize the key idea of the proof: Since $TV(\mu,\nu)<\xi$, by the definition of TV distance, we have $\mu(U)\leq \nu(U)+\xi$, for any measurable $U$.
> To prove $W_{\infty}^{\ell_q} (\mu, \nu)\leq \Delta$, by the above equivalent definition, we want to prove $\mu(U) \leq \nu(U^\Delta)$, for any open set $U$. Hence it suffices to prove $$\nu(U)+\xi\leq\nu(U^\Delta).$$
>
> Rewriting the above inequality gives $\nu(U^\Delta \setminus U)\geq \xi$.
>
> To prove this, we prove that there exists a small $l_q$ ball in the expansion band $U^\Delta \setminus U$ such that its $\nu$-measure is greater than $\xi$. We first prove that there exists a small $l_q$ ball of radius $O(\Delta)$ in the expansion band $U^\Delta \setminus U$ intersected with the domain $\Theta$. Then, we prove that its $\nu$-measure is lower bounded by the product of its Euclidean volume (Lebesgue measure) and the assumed $p_{min}$.
>
> The coupling is not explicitly used in this proof. However, we believe it is interesting to explore what the optimal coupling of the infinity-Wasserstein distance looks like. To the best of our knowledge, this problem is solved for 1-D cases for $\rho$-Wasserstein distance for general $\rho$, see Remark 2.28 and Remark 2.30 of [GM19]: For empirical measures, the optimal transport is given by matching the ordered values. Thanks again for your question, and we will incorporate the discussion in our revision.
>
>
> References:
>
> [GS84] Givens, Clark R., and Rae Michael Shortt. "A class of Wasserstein metrics for probability distributions." Michigan Mathematical Journal 31.2 (1984): 231-240.
>
> [GM19] Peyré, G., & Cuturi, M. (2019). Computational optimal transport: With applications to data science. Foundations and Trends® in Machine Learning, 11(5-6), 355-607.

---

> > ### Comment · Reviewer_tAS6 · 2025-08-05
> >
> > Thank you for your response and the proof overview, and thanks for agreeing to add a discussion about the coupling to the next version. I have no further questions.

---

### Official Review · Reviewer_JJ6e · 2025-07-01

**Clarity:** 4
**Significance:** 3
**Originality:** 4
**Rating:** 5
**Confidence:** 4

**Summary:**

The paper studies a question that sounds, when you hear it the first time, shocking. Is it possible to convert an algorithm with weak privacy guarantees to an algorithm with stronger privacy guarantees using some black-box approach.
The answer is surprisingly yes when the weak privacy guarantee means the mechanism is approx DP and the stronger prviacy guarantee means pure DP (with comparable epsilon).

The paper describes the transformation and uses it to give the best known mechanism for ERM and some other problems.

**Questions:**

1. I am not sure I am able to understand Figure 1: what is the dimension used?
2. I think there is a confusion and sometimes the term support is used instead of range.
3. In Remark 3, the transformation is applied to DP-SGD, but typically DP-SGD has unbounded range...
4. line 190: I don't think all of these papers need to be cited.

**Ethical Concerns:**

["NO or VERY MINOR ethics concerns only"]

**Final Justification:**

I agree that the main contribution here is the possibility of purification and it is fine if the approach is not giving optimal guarantees.

The possibility of a simpler algorithm is a bit concerning, but considering that it would require solving some (potentially non-trivial) issues, I think it doesn't lower the quality of this pape.

**Limitations:**

yes

**Paper Formatting Concerns:**

No concerns

**Quality:**

4

**Strengths And Weaknesses:**

Approx DP, while loved by specialists, has multiple issues in practice: e.g., it is not clear whether any delta is acceptable for real-life solutions operating with sensitive data.

Hence, the ability to construct pure DP mechanisms with performance similar to approx-DP counterparts is opening a lot of new directions. In addition, this method could simplify proofs of lower bounds for approx-DP algorithms since proving pure-DP bounds is usually not too hard.

---

> ### Author Rebuttal · Authors · 2025-07-31
>
> Thank you for your support and your questions!
>
> > Figure 1: what is the dimension used?
>
> The dimension (of the histogram) used is d = 100. The dashed line (labelled "1.0-DP w. Laplace Mech.") is given by fixing the variance at $2/\varepsilon^2=2$ for pure 1.0-DP, regardless of $\delta$. The solid line (labelled "(1.0, $\delta$)-DP w. Gaussian Mech.") is generated using the analytic Gaussian mechanism in [BW18]. The choice of $d$ does not matter and we omitted it in the main paper. Thank you for your question, and we will include more details in the revision to help readers better understand.
>
> > confusion: sometimes the term support is used instead of range.
>
> Thank you for pointing this out. Our purification algorithm operates over the range. More specifically, the purification operates on a set $\Theta$ that contains the (output) range of $\mathcal{M}$. We will clarify this to avoid any confusion in the next revision.
>
> > In Remark 3, the transformation is applied to DP-SGD, but typically DP-SGD has unbounded range...
>
> The bounded domain assumption is common in theoretical work. For example, in the paper by [BST14], with which we compare. We adopt the same assumption to ensure a fair comparison.
>
> > line 190: I don't think all of these papers need to be cited.
>
> Thank you for your suggestion. We originally included these citations to provide a more complete picture of the DP-ERM literature for interested readers. Nevertheless, we agree with your point and will revisit the citation list and cite those are relevant.
>
> References:
>
> [BST14] Bassily, Raef, Adam Smith, and Abhradeep Thakurta. "Differentially private empirical risk minimization: Efficient algorithms and tight error bounds." arXiv preprint arXiv:1405.7085 (2014).
>
> [BW18] Balle, Borja, and Yu-Xiang Wang. "Improving the Gaussian mechanism for differential privacy: Analytical calibration and optimal denoising." International conference on machine learning. PMLR, 2018.

---

> > ### Comment · Reviewer_JJ6e · 2025-08-05
> >
> > Thanks for answering my questions; I will keep my score and I am looking forward reading final version.

---

### Note · Authors · 2025-08-16

We thank the reviewers again for their thoughtful consideration of our paper. We are encouraged that the reviewers find our method valuable. Here, we highlight our main contribution: an algorithmic technique that converts approximate DP mechanisms into pure DP ones under certain conditions. This technique introduces a new design strategy for pure DP algorithms: first run an approximate DP algorithm under suitable conditions, and then apply a purification step. Applied across various settings, this approach yields pure-DP mechanisms that are either information-theoretically optimal or match the best-known pure DP results for the task. In addition, it provides a new way to establish approximate DP lower bounds under specific conditions.

---

### Decision · Program_Chairs · 2025-09-17

**Decision:**

Accept (spotlight)

**Comment:**

This paper develops a nice and simple method for converting any approximate differentially private algorithm into a pure DP algorithm. This is a nice contribution, both in terms of algorithms and upper bounds for pure DP, but also for the possibility of simpler lower bound proofs for the approximate DP setting via pure DP lower bounds.

All the reviewers are positive and found these contributions to be valuable. Therefore I recommend accepting this paper.

However, the authors should address the issues mentioned in the reviews, especially the following two:

1. Discussing the simpler approach of Reviewer XywD for purifying DP algorithms in the paper, and comparing it to their method.

2. Adding and discussing the rates they get in their applications compared to existing DP SOTA rates.